# RLCracker: Exposing the Vulnerability of LLM Watermarks with Adaptive RL Attacks

## Abstract

Large language model (LLM) watermarking has shown promise in detecting AI-generated content and mitigating misuse, with prior work claiming robustness against paraphrasing and text editing. In this paper, we argue that existing evaluations are not sufficiently adversarial, obscuring critical vulnerabilities and overstating the security. To address this, we introduce the *adaptive robustness radius*, a formal metric that quantifies the worst-case resilience of watermarks against adaptive adversaries. By lifting the paraphrase space into a KL-divergence ball, we approximate this radius and theoretically demonstrate that optimizing the attack context and model parameters can significantly reduce the approximated radius, making watermarks highly vulnerable to paraphrase attacks. Leveraging this insight, we propose RLCracker, a reinforcement learning (RL)–based adaptive attack that erases watermarks while preserving semantic fidelity. RLCracker requires only *limited* watermarked examples and *zero* access to the detector. Despite weak supervision, it empowers a 3B model to achieve 98.5% removal success with minimal semantic shift on 1,500-token Unigram-marked texts after training on only *100* short samples. This performance dramatically exceeds 6.75% by GPT-4o and generalizes across five model sizes over ten watermarking schemes. Our results confirm that adaptive attacks are broadly effective and pose a fundamental threat to current watermarking defenses.

## 1 Introduction

With the rapid advancement and increasing accessibility of large language models (LLMs), they are being widely applied across diverse applications, generating fluent, human-like content (Yang et al., 2025). However, this extensive adoption raises pressing concerns about misuse, ranging from misinformation and copyright violations to prompt injection and model theft (Liu et al., 2024a; Wei et al., 2023; Wang et al., 2024). As a safeguard, *text watermarking* has become a leading defense: by subtly embedding statistical signals into model outputs, watermarking allows reliable attribution while preserving the output quality (Kirchenbauer et al., 2023a; Liu et al., 2023b; Zhao et al., 2024).

Most existing watermarking schemes follow a generate-and-detect paradigm. A detection algorithm scans the model outputs for hidden patterns that distinguish AI-generated text from human-written text (Liu et al., 2023a). Prior evaluations show robustness to naive paraphrasing (Liu et al., 2023b), while translation can break a few standard watermarks unless strengthened designs are used (He et al., 2024). However, these evaluations focus on *average-case* prompting under fixed, handcrafted instructions, leaving their *worst-case robustness* under adaptive, high-capacity attacks largely unexplored.

To address this, researchers have investigated watermark removal attacks to more rigorously assess the watermark security. Nevertheless, existing approaches suffer from significant drawbacks, including ineffectiveness on challenging long-form text ($\geq$500 tokens) (Krishna et al., 2023; Cheng et al., 2025), poor generalization ability (Jovanović et al., 2024), and excessive data requirements (i.e., more than 100k samples) (Huang et al., 2024). As LLMs continue to advance, there is an urgent need for both a principled metric and an efficient, generalizable, and data-efficient methodology for evaluating the worst-case vulnerability of watermarking schemes.

In this paper, we systematically investigate the vulnerability of LLM watermarking schemes by introducing the **adaptive robustness radius**, a semantic margin within which watermark detection

Figure 1: The RLCracker algorithm for watermark removal. The model is trained on question and watermarked-text pairs, jointly optimizing a semantic reward to preserve meaning and a token-wise KL reward to shift the model's output $Q$ from the watermark distribution ($P_{wm}$) and towards a human-written distribution ($P_h$), thereby effectively removing the watermark in texts.

remains reliable under adversarial paraphrasing. Inspired by certified robustness in classification (Cohen et al., 2019; Zhu et al., 2021), this metric quantifies the minimum semantic shift required to erase a watermark. However, directly computing this radius is intractable: it requires optimizing over paraphrased outputs without access to the watermark key or detector. To address this, we relax the problem from instance-level attacks to distribution-level shifts. Specifically, we lift the paraphrase space into a KL-divergence ball centered on the watermark distribution and derive a surrogate metric, the *KL adaptive radius*. This formulation reveals that optimizing attack context and model parameters to steer the paraphrased distribution toward human-like text and away from the watermark distribution can significantly reduce the adaptive radius, enabling more effective watermark removal.

Leveraging this insight, we propose **RLCracker**, a reinforcement learning–based attacker that removes watermarks without accessing the detector. As shown in Figure 1, RLCracker jointly optimizes semantic fidelity and token-wise KL divergence to shift model outputs away from the watermark distribution and toward natural text. Despite using only **100 short watermarked samples**, RLCracker enables a compact 3B model to achieve a **98.5% success rate** against the Unigram watermark on *1,500-token texts*, surpassing GPT-4o by over 90 points. Extensive experiments across five model scales over ten distinct watermarking schemes validate our theoretical claims and the broad effectiveness of our attack, exposing critical vulnerabilities in current watermark defenses. Our contributions can be summarized as follows:

- We introduce the adaptive robustness radius, a theoretical framework to formally quantify a watermark's worst-case robustness against paraphrase attacks. We prove that this radius can be systematically minimized by optimizing the attack context and model parameters. (Section 4.1)
- We propose RLCracker, an efficient RL-based adaptive attack that removes watermarks while preserving high semantic fidelity, requiring only limited watermarked samples and zero access to watermark detectors. (Section 4.2)
- We conduct extensive experiments across five model scales and ten distinct watermarking schemes. The results validate our theoretical claims and demonstrate the broad effectiveness of RLCracker, exposing critical vulnerabilities in current watermarking methods. (Section 5)

## 2 RELATED WORKS

**LLM Watermarking.** Watermarking schemes for LLMs, which embed imperceptible but algorithmically detectable patterns in generated text, have become essential for content attribution, copyright protection, and misuse mitigation (Wu et al., 2025; Liu et al., 2024b; Sander et al., 2024). Existing methods can be broadly classified into two dominant paradigms. The first, *logit-based* methods (Hu et al., 2023; Wu et al., 2023; Takezawa et al., 2023), partition the vocabulary into "green" and "red" lists based on a secret hash key and bias the output logits of LLM toward greenlist tokens during generation. The detection is then based on the observation of a statistically significant frequency of these tokens (Kirchenbauer et al., 2023a). In contrast, the second paradigm, *sampling-based* methods (Christ et al., 2024; Zhao et al., 2024; Kuditipudi et al., 2023), embed watermarks by modifying the token selection process itself rather than applying a fixed logit bias, enabling finer-grained control. Recent advances in both paradigms have focused on minimizing detectable patterns, aiming to make the watermarked output indistinguishable from the natural text (Kuditipudi et al., 2023).

**LLM Watermark Removal Attacks.** To assess the robustness of watermarking schemes, researchers have explored various watermark removal attacks. A common approach involves straightforward text

manipulations using LLMs. For instance, GPT-4o and the specialized DIPPER (Krishna et al., 2023) paraphraser have been used to remove watermarks through semantic rewriting (Liu et al., 2023b). Similarly, SIRA (Cheng et al., 2025) performs watermark removal using training-free paraphrasing templates guided by semantically constrained reference texts. Other methods train large proxy models to approximate the watermarking process. WS (Jovanović et al., 2024) mimics the KGW detector scoring mechanism to infer watermark embedding patterns, while $B^4$ (Huang et al., 2024) learns the distribution of watermarked text to guide a paraphraser away from watermark-likely tokens. However, these methods suffer from key limitations. GPT, DIPPER, and SIRA exhibit sharp performance drops on longer texts ($\geq$500 tokens), failing to reveal the worst-case vulnerability of watermarking schemes. WS is tailored specifically to the KGW watermark and generalizes poorly, whereas $B^4$ requires impractically large datasets ($\sim$ 100k samples). These limitations underscore the urgent need for effective, generalizable, and practical attack strategies to rigorously evaluate the watermark security. More detailed related works on LLM watermarking and removal attacks is provided in Appendix B.

## 3 PRELIMINARIES

In this section, we develop a theoretical framework grounded in widely adopted logit-based watermarking schemes. We introduce the *adaptive robustness radius* to quantify the worst-case watermark robustness and formally characterize the optimization objective of adversaries.

**LLM Generation and Logit-based Watermark.** Let $\mathcal{V}$ be a vocabulary and $\mathcal{V}^*$ the set of all finite token sequences. A paraphraser $\pi$ with parameters $\theta$ defines a distribution over output sequences $\mathbf{X}' = (x'_1, x'_2, \dots)$ conditioned on an input $\mathbf{X}$ and a context $c$: $\pi_\theta(\mathbf{X}' \mid \mathbf{X}, c) = \prod_{t \geq 1} \pi_\theta(x'_t \mid x'_{<t}, \mathbf{X}, c)$, where its support is $\mathrm{supp}(\pi_\theta(\cdot \mid \mathbf{X}, c)) := \{\mathbf{X}' \in \mathcal{V}^* : \pi_\theta(\mathbf{X}' \mid \mathbf{X}, c) > 0\}$. Independently of the paraphraser, a watermark is embedded in a generated sequence $\mathbf{X} = (x_1, x_2, \dots)$ using a secret key $s$ and a target greenlist rate $q \in (0, 1)$, which define a keyed hash function $H_s : \mathcal{V}^* \times \mathcal{V} \to [0, 1]$. At each position $t$, the *greenlist* is $\mathcal{G}_t := \{x \in \mathcal{V} : H_s(x_{<t}, x) \leq q\}$, and tokens in $\mathcal{G}_t$ are softly upweighted during generation of $\mathbf{X}$. Given any sequence $\mathbf{X}$, the detector reconstructs each $\mathcal{G}_t$ and computes a score $f(\mathbf{X}, s)$ from the token–greenlist pairs $\{(x_t, \mathcal{G}_t)\}_{t \geq 1}$. A watermark is detected if $f(\mathbf{X}, s) > \delta$ for a fixed threshold $\delta$; otherwise, the sequence is deemed unmarked.

**Threat Model: Semantics-Preserving Paraphrasing.** We consider an adversary aiming to erase the watermark while preserving semantic meaning. The adversary is defined by three components: (1) *Objective*: Given a watermarked sequence $\mathbf{X}$, generate a paraphrased output $\mathbf{X}'$ that is semantically equivalent yet undetectable. (2) *Knowledge*: The adversary does not know the exact watermarking algorithm, the secret key $s$, detector function $f$, or threshold $\delta$, but may query the generator to observe watermarked outputs (Jovanović et al., 2024). (3) *Capability*: The adversary selects and tunes a paraphrasing model $\pi_\theta$, adjusts parameters $\theta$, and chooses any conditioning context $c$. We define a semantic distance function $d(\mathbf{X}, \mathbf{X}')$, where smaller values indicate higher similarity. An attack is considered successful if (i) $d(\mathbf{X}, \mathbf{X}')$ within a certain semantic distance and (ii) $f(\mathbf{X}', s) \leq \delta$.

**Limitations of Standard Robustness.** Prior works assess watermark robustness by testing against fixed paraphrasers or manually crafted prompts (Kirchenbauer et al., 2023b), reporting average-case detection success. However, such evaluations fail to reflect worst-case watermark vulnerability. In practice, attackers can adapt both the model and the attack prompts to suppress watermark signals. To bridge this gap, we propose *adaptive robustness*, a certified notion of integrity under adversarial control of both generation and paraphrasing, constrained by semantic drift.

**A Certified View of Watermark Integrity.** Our formulation draws inspiration from certified robustness in classification (Cohen et al., 2019; Zhu et al., 2021) and distributionally robust optimization (DRO) (Sinha et al., 2018; Duchi & Namkoong, 2019), where models are guaranteed to remain correct under bounded perturbations. Similarly, we define a semantic robustness radius within which watermark detection is verifiably preserved, even when the adversary adaptively chooses paraphrasing models and prompts. Unlike prior watermarking work focused on average-case robustness (Liu et al., 2023b), our framework yields worst-case semantic guarantees under adaptive attacks.

**Adaptive Robustness Radius** $r^\star(\mathbf{X})$**.** Let $\Theta$ be a constrained space of paraphrasing models $\pi_\theta$, and $\mathcal{C}$ a restricted prompt space. We define the *adaptive robustness radius* as:

$$r^\star(\mathbf{X}) := \sup \left\{\rho \geq 0 \mid \forall \theta \in \Theta, \ c \in \mathcal{C}, \ \forall \mathbf{X}' \in \mathrm{supp}(\pi_\theta(\cdot \mid \mathbf{X}, c)), \ d(\mathbf{X}, \mathbf{X}') \leq \rho \Rightarrow f(\mathbf{X}', s) > \delta\right\}.$$

This measures the worst-case semantic margin within which the watermark remains intact despite adaptive manipulation. It parallels certified prediction radii in adversarial NLP (Zhu et al., 2021), but applies to watermark detection rather than classification.

**Problem: How to Evaluate $r^\star(\mathbf{X})$?** Evaluating $r^\star(\mathbf{X})$ reduces to a constrained optimization: the adversary seeks the closest valid paraphrase that evades detection. This yields the equivalent form:

$$r^\star(\mathbf{X}) = \inf_{\theta \in \Theta, \, c \in \mathcal{C}, \, \mathbf{X}'} d(\mathbf{X}, \mathbf{X}') \quad \text{subject to} \quad \mathbf{X}' \in \mathrm{supp}(\pi_\theta(\cdot \mid \mathbf{X}, c)), \; f(\mathbf{X}', s) \leq \delta.$$

However, solving this problem is challenging: the adversary does not know the detector function $f$, the threshold $\delta$, or the secret key $s$, making the problem black-box and non-convex, and instance-sensitive. In this work, we develop efficient approximations to estimate $r^\star(\mathbf{X})$, and use it as a robustness metric for certified evaluation under constrained adaptive attacks.

**Adversarial Optimization Objective.** Given a watermark with hash key $s$, the attacker seeks to generate the closest paraphrase that evades detection. The adversary directly minimizes the semantic deviation needed to break the watermark by tuning model parameters $\theta$ and attack context $c$:

$$\inf_{\theta \in \Theta, \, c \in \mathcal{C}, \, \mathbf{X}'} d(\mathbf{X}, \mathbf{X}') \quad \text{s.t.} \quad \mathbf{X}' \in \mathrm{supp}(\pi_\theta(\cdot \mid \mathbf{X}, c)), \; f(\mathbf{X}', s) \leq \delta.$$

This formulation supports both empirical and certified evaluation of watermark robustness, analogous to robust training in adversarial learning (Madry et al., 2018), and lays a foundation for benchmarking watermarking schemes by worst-case guarantees, not just the average-case attack success.

# 4 METHODOLOGY

## 4.1 A COMPUTABLE CERTIFICATE VIA KL DIVERGENCE

**Direct instance-level computation of $r^\star(\mathbf{X})$ is intractable.** The adaptive robustness radius $r^\star(\mathbf{X})$ measures how far an adversarial paraphrase $\mathbf{X}'$ can drift semantically from the original input $\mathbf{X}$ while remaining detectable. But identifying this worst-case $\mathbf{X}'$ requires a combinatorial search over the entire paraphrase space of $\mathbf{X}$, a discrete, irregular domain with no gradients and no exploitable structure. This makes instance-level certification fundamentally infeasible.

**From deterministic instance to local distribution: a new robustness lens.** We model the watermarked input $\mathbf{X}$ as inducing a *local distribution* $P_{\mathbf{X}, c, \theta}$, formed by passing $\mathbf{X}$ through a lightweight paraphraser $\pi_\theta(\cdot \mid \mathbf{X}, c)$ with context $c$ and parameters $\theta$ that generates natural, meaning-preserving variants. This shift is motivated by a key observation: naive paraphrases of $\mathbf{X}$ often preserve the watermark, suggesting that $\mathbf{X}$ carries a stochastic watermark signature rather than a fixed pattern. This distributional view captures far more about watermark's behavior than any individual paraphrase.

**A smooth geometric view of the adaptive radius.** Rather than searching for a single elusive paraphrase, we evaluate how far an *adversarial paraphrasing distribution $Q$* moves from the local watermark distribution $P_{\mathbf{X}, c, \theta}$ in KL space. Here $Q$ represents an attacker's distribution over meaning-preserving rewrites that aim to reduce or erase the watermark signal. Measuring this distributional deviation yields the *KL adaptive radius*, a geometric robustness certificate quantifying the maximum permissible drift under which all such adversarial paraphrase distributions remain detectable.

**Definition 1** (KL Adaptive Radius). *For a given watermarked input $\mathbf{X}$, context $c$ and paraphraser parameters $\theta$, the KL adaptive radius is*

$$r_{\mathrm{KL}}(\mathbf{X}, c, \theta) := \sup \left\{ \rho \geq 0 \mid \forall Q : \mathrm{D}_{\mathrm{KL}}(Q \| P_{\mathbf{X}, c, \theta}) \leq \rho \Rightarrow \mathbb{E}_{\mathbf{X}'' \sim Q}[f(\mathbf{X}'', s)] > \delta \right\}.$$

The KL adaptive radius measures how much local distributional shift the watermark can withstand. It defines a safety region around the local watermark distribution: as long as an attack distribution stays within this region, its expected detection score remains above threshold. A small radius signals a fragile watermark; a large one indicates resilience to substantial distributional perturbations.

**Advantages over instance-level robustness definitions.** Prior robustness notions, including the $(\varepsilon, \delta)$-robustness (Diaa et al., 2024) and the instance-level radius in Section 3, measure deviation of a *single* paraphrase from $\mathbf{X}$, a brittle criterion in the discrete paraphrase space. By contrast, our definition compares *distributions*: the attack distribution $Q$ and the local watermark distribution

$P_{\mathbf{X},c,\theta}$ in KL space. This provides two key advantages. (1) *More informative:* $P_{\mathbf{X},c,\theta}$ reveals how the watermark behaves across natural variations of $\mathbf{X}$, exposing structure invisible to any individual paraphrase. (2) *Geometrically smoother:* KL divergence endows paraphrasing distributions with a differentiable geometry, enabling gradient-based reasoning and formal robustness certificates that the discrete instance space cannot support.

**A necessary condition for preserving watermark robustness.** This relaxation enables us to express robustness as a concrete requirement on the adversarial distribution: the watermark remains detectable only if the attacker's paraphrasing distribution stays sufficiently close, in KL divergence, to the local watermark distribution. To make this condition explicit and obtain a computable lower bound on the KL radius, we assume a mild concentration property of the detector score. This assumption is natural: green-list watermarking schemes produce z-scores that are empirically tightly concentrated and, as shown in Appendix C.1, provably sub-Gaussian under mild sequential assumptions.

**Assumption 1** (Sub-Gaussian Detector Score). *Under the local watermark distribution $P_{\mathbf{X},c,\theta}$ (i.e., $\mathbf{X}'' \sim P_{\mathbf{X},c,\theta}$), the detector score $f(\mathbf{X}'', s)$ concentrates around its mean $\mu(\mathbf{X}, s, c, \theta)$ with sub-Gaussian tails governed by variance proxy $\sigma^2(\mathbf{X}, s, c, \theta)$.*

**Theorem 1** (KL lower bound for watermark robustness). *Let $Q$ be any adversarial paraphrasing distribution. Under the sub-Gaussian assumption,*

$$\mathbb{E}_{\mathbf{X}'' \sim Q}[f(\mathbf{X}'', s)] \geq \mu(\mathbf{X}, s, c, \theta) - \sqrt{2\,\sigma^2(\mathbf{X}, s, c, \theta)\,\mathrm{D}_{\mathrm{KL}}(Q \| P_{\mathbf{X},c,\theta})}.$$

*This implies that the KL adaptive radius is lower-bounded by a computable certificate $\rho^*$:*

$$r_{\mathrm{KL}}(\mathbf{X}, c, \theta) \geq \rho^*(\mathbf{X}, s, c, \theta) := \frac{[(\mu(\mathbf{X}, s, c, \theta) - \delta)_+]^2}{2\,\sigma^2(\mathbf{X}, s, c, \theta)},$$

*where $(x)_+ = \max\{x, 0\}$.*

**Remark 1.** *A large KL shift is necessary to scrub the watermark. The theorem imposes a geometric constraint on the attacker: the expected detection score drops only when the adversarial distribution moves far away from the natural distribution of meaning-preserving rewrites $P_{\mathbf{X},c,\theta}$. Scrubbing the watermark therefore requires a* substantial *distributional shift, not a single cleverly crafted paraphrase. Because the score decreases only at the rate $\sqrt{\mathrm{KL}(Q\|P)}$, any successful attack must push the KL divergence beyond the certified radius. This yields a natural and differentiable attack objective: maximize $\mathrm{KL}(Q\|P_{\mathbf{X},c,\theta})$ to leave the detectable region. The proof is given in Appendix C.*

**Remark 2.** *Enlarging the paraphrasing space can shrink the adaptive radius. Our framework reveals that certified robustness is fundamentally tied to the attacker's search space. When the adversary is allowed to choose the context $c$ and parameters $\theta$ from a larger candidate set, the smallest attainable KL adaptive radius can be reduced. Thus, restricting evaluations to a narrow set of paraphrasers can substantially* overestimate *the true robustness of a watermark. Specifically, let $R_{\mathrm{KL}}(\mathbf{X}, \mathcal{C}, \Theta) = \min_{c \in \mathcal{C}, \theta \in \Theta} r_{\mathrm{KL}}(\mathbf{X}, c, \theta)$ denote the worst-case KL adaptive radius over context set $\mathcal{C}$ and parameter set $\Theta$. Now we present the following corollary.*

**Proposition 1** (Increased vulnerability under larger attack spaces). *If $\mathcal{C} \subseteq \mathcal{C}'$ and $\Theta \subseteq \Theta'$, then*

$$R_{\mathrm{KL}}(\mathbf{X}, \mathcal{C}', \Theta') \leq R_{\mathrm{KL}}(\mathbf{X}, \mathcal{C}, \Theta).$$

**Remark 3.** *The proof follows directly from the definition of $R_{KL}$. Proposition 1 states that enlarging the adversary's choice of contexts or paraphraser configurations can only reduce the certified radius. Expanding the attacker's search space with more prompts, broader parameters, or more expressive paraphrasers can reduce the worst-case robustness guarantee. This highlights a key vulnerability: robustness claims that do not account for adversarial choice over $c$ and $\theta$ may substantially overstate the watermark's true resilience.*

## 4.2 RLCRACKER: CRACK THE WATERMARKS WITH RL ATTACK

**From KL guidance to practical attack strategies.** The KL analysis in Section 4.1 shows that watermark removal requires increasing the divergence between the attack distribution $Q$ and the local watermark distribution $P_{\mathbf{X},c,\theta}$. In practical terms, a successful attack must place more probability mass on regions where the watermark signal is weak. Multi-sample attacks provide a simple mechanism for achieving this: by drawing $k$ candidates from $\pi_\theta$, the chance that *at least one* sample naturally falls into a low-watermark region increases with $k$.

To validate this effect, we run a Pass@20 attack on the EWD watermark Lu et al. (2024) using Qwen2.5-3B-Instruct. As shown in Figure 2, Pass@20 reaches an 89% success rate at a semantic threshold of 0.7, far above the 32% Pass@1 baseline. This sharp improvement shows that *multi-sample* attacks are highly effective: with enough samples, at least one candidate typically lands in a low-score region that scrubs the watermark. It also suggests a natural goal: *can we raise single-sample (Pass@1) success to approach multi-sample (Pass@k) performance?*

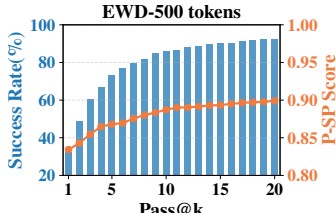

Figure 2: Pass@20 on EWD.

**From multi-sample attacks to single-sample attacks.** Although multi-sample attacks are powerful, deploying Pass@k in real black-box settings is infeasible because an attacker cannot query the detector to select the successful sample. This motivates an alternative: learn a policy whose *single* output already behaves like the best-of-$k$ candidates, achieving high success without an oracle detector. Prior work by Yue et al. (2025) shows that reinforcement learning can approximate multi-sample behavior by shaping a model's output distribution toward high-reward regions. Building on this insight, we reformulate watermark removal as a policy optimization problem: our aim is to train a model whose attack distribution $Q$ achieves Pass@k-level success in a single sample, eliminating the dependence on detector access.

**Algorithm Design.** Contemporary LLM watermarking techniques often induce detectable shifts in output distributions (Liu et al., 2024a). We exploit this property to propose RLCracker, an adaptive RL-based attack that reshapes the adversarial output distribution $Q$ to align with human text $P_h$, while diverging from the watermark distribution $P_{wm}$. Formally, the objective is defined as: $\max_{\theta \in \Theta} D_{\text{KL}}(Q\|P_{wm}) - D_{\text{KL}}(Q\|P_h)$. RLCracker employs token-wise optimization derived from GRPO (Shao et al., 2024), requiring only question–watermarked response pairs $(q, wr)$ to effectively learn watermark evasion. Concretely, the attack policy $\pi_\theta$ is iteratively updated by sampling a set of outputs $\{o_1, \ldots, o_G\}$ from the previous policy $\pi_{\theta_{\text{old}}}$, maximizing the following training objective:

$$\mathcal{J}(\theta) \approx \mathbb{E}_{\{o_i\} \sim \pi_{\theta_{\text{old}}}} \frac{1}{G} \sum_{i=1}^{G} \frac{1}{|o_i|} \sum_{t=1}^{|o_i|} \left[ \frac{\pi_\theta(o_{i,t} \mid q, o_{i,<t})}{\pi_{\theta_{\text{old}}}(o_{i,t} \mid q, o_{i,<t})} \left( w_1 \hat{A}_i + w_2 \Delta r_{i,t} \right) \right.$$
$$\left. - \beta D_{\text{KL}}\left[\pi_\theta\|\pi_{\text{ref}}\right] \right] - w_3 \, \text{PPL}(\pi_\theta, \{o_i\}), \tag{1}$$

where $\Delta r_{i,t} = D_{\text{KL},i,t}(Q\|P_{wm}) - D_{\text{KL},i,t}(Q\|P_h)$ is the token-wise KL reward, $\pi_{\text{ref}}$ denotes the reference model, and $w_1, w_2, w_3$ are the weights of the respective components. The remaining terms impose token-wise KL regularization and a perplexity penalty to preserve fluency.

**Rewards Design.** We employ two reward components to ensure semantic fidelity while evading watermark detection. First, the *semantic reward* ($A$) is computed from the P-SP score (Wieting et al., 2021) between the generated text and the original text, scaled by a sigmoid transformation to amplify gradients and preserve semantic similarity. We normalize $A$ across the batch to obtain the semantic advantage $\hat{A}_i = \frac{A_i - \text{mean}(A)}{\text{std}(A)}$, which stabilizes learning and preserves semantic consistency.

Second, the *token-wise KL reward* ($D_{\text{KL},i,t}$) steers $\pi_\theta$ toward human-like behavior while discouraging watermark-like patterns. Each token's distribution is encouraged to align with a human-like reference (approximating $P_h$) and diverge from a watermark-induced reference (approximating $P_{wm}$). In practice, $P_{wm}$ is instantiated by passing the watermarked instance $wr$ through a lightweight reference model $\pi_{\text{ref}}$ with a simple prompt, inspired by the weak rewriting behavior that typically preserves watermarks. $P_h$ is obtained by querying the same unwatermarked reference model $\pi_{\text{ref}}$ with the question $q$ that was used to generate the watermark, yielding a human-like distribution.

Finally, we estimate the token-wise KL divergence using the following estimator (Schulman, 2020):

$$D_{\text{KL},i,t}(Q\|P_*) \approx \frac{\pi_{\text{ref}}(o_{i,t} \mid *, o_{i,<t})}{\pi_\theta(o_{i,t} \mid wr, o_{i,<t})} - \log \frac{\pi_{\text{ref}}(o_{i,t} \mid *, o_{i,<t})}{\pi_\theta(o_{i,t} \mid wr, o_{i,<t})} - 1,$$

where $* \in \{h, wm\}$ indicates whether $P_*$ corresponds to the human-like distribution $P_h$ or the watermark-induced distribution $P_{wm}$. Both are approximated using the reference model $\pi_{\text{ref}}$ evaluated with different inputs: $P_h(o_{i,t}) \approx \pi_{\text{ref}}(o_{i,t} \mid q, o_{i,<t})$ and $P_{wm}(o_{i,t}) \approx \pi_{\text{ref}}(o_{i,t} \mid wr, o_{i,<t})$, where $q$ is the question and $wr$ is its corresponding watermarked response available to the attacker.

## 5 EXPERIMENTS

### 5.1 EXPERIMENTAL SETUP

We begin by introducing our experimental setups. Details can be found in Appendix D.

**Victim Models and Attackers.** To simulate diverse real-world scenarios, we consider three victim models of varying sizes from two well-known model families for watermark text generation: Qwen2.5-1.5B-Instruct, LLaMA3.1-8B-Instruct (Grattafiori et al., 2024), and Qwen2.5-32B-Instruct (Qwen et al., 2025). For attackers, we consider five models with different reasoning capabilities: Qwen3-0.6B, Qwen3-1.7B, Qwen3-4B, and Qwen3-8B (Yang et al., 2025) as reasoning-capable, and Qwen2.5-3B-Instruct as non-reasoning.

**Watermarking Schemes.** We evaluate ten distinct watermarking schemes from both logit-based and sampling-based families. For logit-based watermarking, we include EWD (Lu et al., 2024), KGW, KGW_selfhash (Kirchenbauer et al., 2023a), UPV (Liu et al., 2023a), SWEET (Lee et al., 2023), Unigram (Zhao et al., 2023), SIR (Liu et al., 2023b), and X-SIR (He et al., 2024). To reflect recent advances, we also incorporate the sampling-based SynthID-Text (Dathathri et al., 2024) and cryptographic PF-Watermark (Zhao et al., 2024). All schemes are implemented and detected using the widely-used MarkLLM toolkit (Pan et al., 2024) under their default settings.

**Datasets.** We consider four datasets: Reddit WritingPrompts (Verma et al., 2023), LFQA dataset (Krishna et al., 2023), MMW BookReport and FakeNews (Piet et al., 2025). For each watermarking scheme, we generate samples of 250, 500 and 1500 tokens. The training set comprises 100 prompt–watermarked response pairs generated from WritingPrompts. For evaluation, 250 and 500-token samples are created using 100 unique entries from each dataset. In the 1500-token setting, LFQA is excluded due to its short-text nature; instead, 400 samples are generated from BookReport, FakeNews, and WritingPrompts in a 1:1:2 ratio.

**Attack Methods.** Under the black-box threat model, in addition to RLCracker, we evaluate five attack methods to expose vulnerabilities in watermarking. **Base** directly paraphrases watermarked samples via simple prompting. **Think** leverages the model's reasoning ability to generate more diverse paraphrases. **SysP** expands the prompt context via system prompts to increase attack success. We also employ SIRA (Cheng et al., 2025), DIPPER (Krishna et al., 2023), and LLM Paraphraser (Liu et al., 2023b), where we adopt GPT-4o-2024-08-06.

**Implementation Details of RLCracker.** We apply RLCracker to strengthen watermark removal. The semantic reward employs a reparameterized sigmoid scaling, with 0.85 as the threshold separating positive from negative rewards, thereby promoting higher-quality rephrasings. Training on 100 samples of 500 tokens each, with a batch size of 48 and a group size $G = 12$, completes in approximately 1.5 hours for Qwen3-4B and 0.5 hours for Qwen3-0.6B using four NVIDIA A100 GPUs. In our experiments, we find that using dynamic $w_1$ with $w_2 = 0.9$, and $w_3 = 0.1$ yields the best performance across most watermarking schemes.

**Metric.** We use the Evasion Success Rate (ESR) to evaluate the robustness of watermarks. It is defined as the proportion of rephrased texts classified as unwatermarked while preserving high semantic similarity to the original (P-SP score $> 0.7$ following prior work (Jovanović et al., 2024)), relative to all rephrased texts. We also employ perplexity and ChatGPT as a Judge to evaluate the quality of rephrased texts (Cheng et al., 2025), with detailed numbers provided in Appendix E.1.

### 5.2 MAIN RESULTS

In this subsection, we evaluate the robustness of different watermarking schemes based on data generated by the Llama-3.1-8B-Instruct. Results are shown in Table 1 and 2, with additional details provided in Appendix E.1.

**System prompts are an overlooked adversarial tool: simple in design but powerful in effect.** While prior work has shown

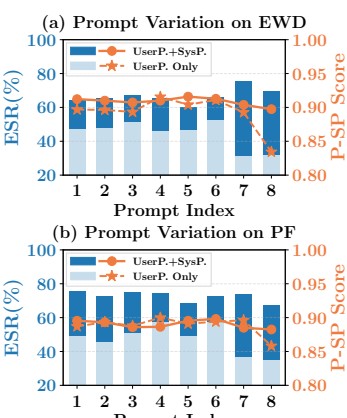

Figure 3: ESR and P-SP variation across user prompts (UserP.), with and without system prompt (SysP.).

Table 1: Evasion Success Rate (ESR, %) across models and watermarking schemes. **Bold** indicates the best ESR per model. RLCracker is tested using the same prompt as the *SysP.* method. Abbreviations: SWEET (SWE.), Unigram (Unig.), KGW_selfhash (KG_s.). More results for SIR, SynthID-Text, KGW and UPV watermarks are presented in Table 2, 20, 22, 26 and 28 in Appendix E.1.

| Models | Methods | 500 tokens | | | | | | 1500 tokens | | | | | |
|---|---|---|---|---|---|---|---|---|---|---|---|---|---|
| | | EWD | SWE. | XSIR | Unig. | KG_s. | PF | EWD | SWE. | XSIR | Unig. | KG_s. | PF |
| Qwen3-0.6B | Base | 13.5 | 21.3 | 29.0 | 35.8 | 35.5 | 14.5 | 3.50 | 4.25 | 26.0 | 5.00 | 17.5 | 11.8 |
| | SysP | 36.5 | 44.8 | 52.5 | 42.5 | 63.3 | 37.3 | 5.75 | 12.5 | 39.3 | 6.25 | 37.0 | 22.0 |
| | Think | 28.5 | 35.3 | 37.3 | 34.3 | 66.3 | 27.3 | 10.8 | 10.3 | 32.0 | 6.00 | 33.0 | 22.5 |
| | SIRA | 0.00 | 0.00 | 0.00 | 0.00 | 0.25 | 0.00 | 0.00 | 0.00 | 0.00 | 0.25 | 0.00 | 0.00 |
| | RLCracker | **91.5** | **92.5** | **86.5** | **67.0** | **85.3** | **76.8** | **63.3** | **61.5** | **60.0** | **87.5** | **66.8** | **62.0** |
| Qwen3-1.7B | Base | 34.2 | 42.8 | 43.0 | 34.0 | 56.8 | 36.5 | 4.3 | 6.8 | 30.3 | 5.0 | 32.5 | 19.3 |
| | SysP | 59.8 | 70.8 | 70.8 | 40.5 | 76.3 | 63.3 | 7.5 | 10.5 | 39.5 | 5.80 | 41.0 | 30.0 |
| | Think | 80.8 | 88.0 | 75.5 | 31.0 | 86.0 | 71.2 | 42.5 | 48.0 | 53.5 | 4.50 | 66.3 | 55.5 |
| | SIRA | 0.25 | 0.00 | 0.25 | 0.00 | 0.25 | 0.00 | 0.00 | 0.00 | 0.00 | 0.00 | 0.00 | 0.00 |
| | RLCracker | **91.8** | **91.0** | **86.0** | **73.0** | **87.5** | **79.5** | **62.7** | **61.3** | **62.3** | **77.8** | **68.0** | **61.0** |
| Qwen3-4B | Base | 39.3 | 53.3 | 50.3 | 39.3 | 70.0 | 47.8 | 10.0 | 12.3 | 38.0 | 4.50 | 36.0 | 25.8 |
| | SysP | 54.3 | 65.8 | 66.8 | 43.8 | 79.3 | 63.0 | 12.5 | 15.0 | 45.0 | 5.50 | 40.8 | 34.5 |
| | Think | 76.0 | 83.5 | 75.3 | 39.5 | 85.3 | 71.8 | 40.8 | 45.0 | 62.0 | 4.75 | 78.0 | 59.0 |
| | SIRA | 0.00 | 0.00 | 0.25 | 0.00 | 0.75 | 0.00 | 0.00 | 0.00 | 0.00 | 1.25 | 0.00 | 0.00 |
| | RLCracker | **93.3** | **95.8** | **85.5** | **73.5** | **88.8** | **82.3** | **64.0** | **65.3** | **66.3** | **63.8** | **82.3** | **64.3** |
| Qwen3-8B | Base | 72.0 | 70.8 | 77.0 | 40.5 | 81.3 | 68.3 | 33.5 | 30.0 | 48.8 | 5.25 | 58.8 | 44.3 |
| | SysP | 72.8 | 81.5 | 82.0 | 43.5 | 85.5 | 73.0 | 32.0 | 38.8 | 52.5 | 6.75 | 64.5 | 50.3 |
| | Think | 89.0 | 91.0 | 81.5 | 40.8 | 83.8 | 71.8 | 54.0 | 51.5 | 57.8 | 7.25 | 71.3 | 59.5 |
| | SIRA | 0.00 | 0.00 | 0.25 | 0.50 | 0.25 | 0.00 | 0.00 | 0.25 | 0.00 | 0.50 | 0.00 | 0.25 |
| | RLCracker | **94.8** | **96.3** | **90.2** | **80.5** | **90.8** | **84.3** | **70.0** | **71.3** | **69.3** | **81.8** | **84.3** | **73.3** |
| Qwen2.5-3B -Instruct | Base | 31.3 | 46.8 | 68.8 | 37.3 | 58.8 | 37.0 | 11.5 | 17.0 | 39.3 | 3.50 | 50.8 | 16.0 |
| | SysP | 75.5 | 83.0 | 78.5 | 43.5 | 84.5 | 74.0 | 27.3 | 35.5 | 45.3 | 4.50 | 56.8 | 42.8 |
| | SIRA | 0.25 | 0.25 | 0.25 | 0.75 | 0.75 | 0.00 | 0.00 | 0.00 | 0.00 | 0.00 | 0.50 | 0.00 |
| | RLCracker | **93.3** | **95.3** | **89.8** | **78.5** | **89.8** | **81.8** | **73.0** | **71.5** | **66.5** | **98.5** | **78.0** | **77.8** |
| GPT-4o | — | 71.0 | 81.5 | 77.3 | 49.8 | 86.8 | 78.0 | 10.3 | 20.5 | 50.3 | 6.75 | 61.8 | 49.3 |
| DIPPER | — | 30.3 | 52.8 | 0.00 | 33.3 | 72.0 | 43.5 | 1.15 | 5.75 | 0.00 | 4.50 | 41.5 | 21.0 |

that user prompts can impact watermark evasion (Kirchenbauer et al., 2023b;a), evaluations have largely focused on user input variation, overlooking the broader influence of system-level instructions. We confirm the known sensitivity to user prompts: Figure 3 shows ESR on EWD varies from 31% to 53% across prompt indices. However, our key insight goes even further: by introducing a role-assignment system prompt (framing the model as a skilled assistant), we observe a consistent boost in ESR and a reduction in variance. The dark blue bars (with SysP.) outperform the light blue (without SysP.) across the board. Crucially, this boost comes without sacrificing semantic fidelity, as shown by improved P-SP scores. Table 1 further confirms that SysP consistently outperforms Base across all watermark schemes, model sizes, and token lengths. For example, on Qwen3-4B (500 tokens), ESR rises from 39.3% to 54.3% on EWD, and from 70.0% to 79.3% on KGW_selfhash. This efficacy supports Theorem 1, showing that richer attack contexts improve success. Despite the effectiveness, system prompt design remains largely overlooked in watermark robustness evaluations. More details can be found in Appendix E.2.

**Paraphraser capability is critical yet not fully explored in evaluating watermark robustness.** While prior work has focused on text length as a vulnerability axis (Kirchenbauer et al., 2023b), it largely ignores the adversarial role of model strength. Tables 1 and 2 reveal a clear trend: as paraphraser capability improves, ESR increases significantly across all ten different watermarking schemes. These results highlight a fundamental limitation of current watermarking schemes: they do not scale with paraphraser ability and are easily broken by more powerful models. Note that Table 2 presents a separate ablation study with a different subset of watermarking schemes, so its results are not directly comparable to those in Table 1.

Table 2: ESRs on long-texts (1500 tokens) under Base attack.

| Model | KGW | SIR | UPV | Syn. |
|---|---|---|---|---|
| Qwen3-0.6B | 6.00 | 33.3 | 39.0 | 38.7 |
| Qwen3-1.7B | 9.25 | 35.3 | 39.0 | 49.0 |
| Qwen3-4B | 15.8 | 37.0 | 30.5 | 54.0 |
| Qwen3-8B | 46.8 | 62.5 | 70.0 | 78.3 |
| GPT-4o | 49.3 | 65.5 | 77.8 | 88.3 |

**The paraphraser's reasoning ability is an underexplored adversarial factor in watermark robustness.** Beyond model size, we find that exploiting the paraphraser's reasoning capability further degrades watermark resilience. When guided to perform intermediate reasoning before rewriting, the

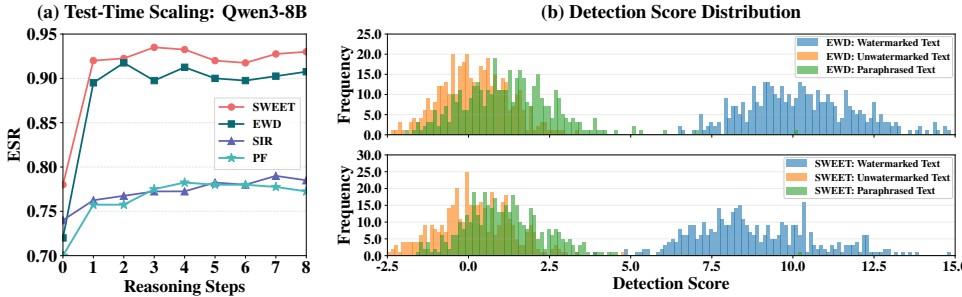

Figure 4: (a) shows the effectiveness of test-time scaling on watermark removal using Qwen3-8B; (b) illustrates the detection score distributions for EWD and SWEET watermarks across unwatermarked outputs, watermarked outputs, and paraphrased texts generated by Qwen3-4B trained with RLCracker.

model produces semantically faithful yet structurally transformed outputs that evade detection. As shown in Table 1 (*Think* rows), Qwen3-4B's ESR increases from 39.3% to 76.0% on the 500-token EWD watermark, and from 36.0% to 78.0% on the 1500-token KGW_selfhash watermark, gaining greater than 40 percentage points. Similar trends appear across models and watermark schemes, showing that watermark signals cannot survive structured paraphrasing enabled by reasoning.

**Reasoning depth is a compounding and underexamined threat to watermark robustness.** We extend our analysis by employing test-time scaling (Muennighoff et al., 2025), where the paraphraser engages in multiple iterations of reasoning before generating a paraphrase. Specifically, whenever the model autonomously determines that the current reasoning process has concluded, we enforce additional reasoning cycles based on the model's previous outputs, thereby increasing the number of reasoning steps. For the Qwen3-8B model, imposing eight reasoning steps on 500-token watermarked inputs significantly amplifies ESR. As shown in Figure 4(a), ESR rises consistently for SIR and PF, while EWD and SWEET remain stable near 90%. These results show that deeper reasoning enables more fluent and targeted rewrites, revealing that current watermarking schemes are not robust to increasingly capable inference-time attackers. Full results are in Appendix E.3.

**RLCracker outperforms baseline attacks by combining key adversarial factors.** Our earlier findings highlight two key drivers of watermark removal: system prompts and the reasoning capability of the paraphraser. RLCracker integrates both by using a fixed system prompt and reinforcement learning to enhance reasoning. As shown in Table 1, it consistently achieves higher ESR across models. On the 1500-token Unigram benchmark, RLCracker reaches 87.5% for Qwen3-0.6B and 98.5% for Qwen2.5-3B-Instruct, far surpassing GPT-4o's 6.75%. In

Table 3: Performance of Qwen2.5-3B-Ins. on 1500-tokens Unigram.

| Method | ESR | Rem.(%) | P-SP |
|---|---|---|---|
| Base | 3.50 | 20.5 | 0.85 |
| SysP. | 5.50 | 25.0 | 0.82 |
| SIRA | 0.50 | 88.0 | 0.47 |
| RLCracker | 98.5 | 100. | 0.92 |

contrast, while SIRA (Cheng et al., 2025) reports high removal rates, it fails under ESR evaluation, achieving near-zero success on long-form inputs due to poor semantic preservation. As shown in Table 3, SIRA removes 88% of watermarks (Rem.) but yields a low average P-SP score of just 0.47, indicating that its outputs are no longer semantically faithful to the original.

**RLCracker achieves distribution-level alignment with unwatermarked text.** Beyond high ESR scores, we further examine how RLCracker reshapes detection behavior. Specifically, we analyze detection score distributions for EWD and SWEET watermarks across unwatermarked, watermarked, and RLCracker-rephrased texts using 400 samples of 500-token inputs. As shown in Figure 4(b), RLCracker substantially lowers the detection scores of paraphrased outputs, shifting them away from watermarked distributions and toward unwatermarked ones. This illustrates that RLCracker's effectiveness stems not just from erasing watermark signals, but from aligning its output distribution with clean, unmarked text. Additional results are provided in Appendix E.4.

## 5.3 ABLATION STUDY

In this subsection, we examine RLCracker's generalization to OOD watermark texts and sensitivity to training data. ESR results are reported in Table 4, 5, and 6.

**RLCracker exhibits robust generalization to OOD watermarked texts.** To evaluate the performance of RLCracker on out-of-distribution (OOD) data, we test models trained on Llama-3.1-

Table 4: Cross-model watermark ESR (%) of RLCracker.

| Gen. Att. | Methods | Qwen2.5-1.5B-Ins. | | | Qwen2.5-32B-Ins. | | |
|---|---|---|---|---|---|---|---|
| | | EWD | SWE. | PF | EWD | SWE. | PF |
| Qwen3-0.6B | Base | 10.8 | 13.0 | 14.5 | 12.5 | 15.8 | 14.3 |
| | RLCracker | 87.3 | 84.0 | 69.3 | 96.0 | 95.8 | 81.5 |
| Qwen2.5-3B -Instruct | Base | 22.8 | 28.0 | 26.8 | 28.0 | 33.8 | 29.3 |
| | RLCracker | 86.0 | 84.8 | 73.5 | 97.3 | 98.8 | 84.5 |
| Qwen3-4B | Base | 21.0 | 22.8 | 36.0 | 42.3 | 48.3 | 43.8 |
| | RLCracker | 84.0 | 83.0 | 73.5 | 98.0 | 98.0 | 84.5 |

Table 5: EWD vs. Training set.

| Models | Tokens | Samples | | |
|---|---|---|---|---|
| | | 50 | 100 | 200 |
| Qwen3 (0.6B) | 250 | 82.5 | 87.8 | 89.3 |
| | 500 | 86.5 | 91.5 | 92.0 |
| | 1500 | 91.5 | 92.5 | 92.8 |
| Qwen3 (4B) | 250 | 87.8 | 92.0 | 92.8 |
| | 500 | 90.0 | 93.3 | 93.8 |
| | 1500 | 92.3 | 95.3 | 95.3 |

8B-Instruct outputs against 500-tokens watermarked texts generated by Qwen2.5-1.5B-Instruct and Qwen2.5-32B-Instruct. As shown in Table 4, RLCracker attains high removal success rates across domains, substantially outperforming the *Base* method. For instance, a 0.6B model achieves 96% ESR on 32B-generated EWD watermarks, compared to only 12.5% with *Base*. Moreover, ESR is generally higher when targeting outputs from larger models, which may be due to their higher-quality generations. These results demonstrate that RLCracker generalizes effectively across OOD data, achieving high ESR and robust performance in watermark removal. More details are in Appendix E.5.

**Small training sets suffice for RLCracker to achieve robust performance.** We evaluate RL-Cracker's performance with respect to two training data factors: (1) token length per sample and (2) number of training samples. Experiments are conducted using Qwen3-0.6B and Qwen3-4B on the EWD watermark. As shown in Table 5, we observe that even with only 50 samples of 250 tokens, Qwen3-0.6B achieves an ESR of 82.5%. ESR further improves to 91.5% as the token length increases under a fixed sample size. While enlarging the amount of training samples also yields gains, performance plateaus beyond 100 samples. These results suggest that relatively small training sets already suffice for RLCracker to achieve strong watermark removal. More details are in Appendix E.6.

**RLCracker shows minor degradation when trained on mixed-key data.** In real-world settings, collected watermarked data may be generated with different hash keys (Liu et al., 2024a), resulting in mixed watermark patterns. To evaluate RL-Cracker in this scenario, we construct three mixed-key training sets for watermark EWD, PF and SWEET. Each set contains 100 samples (500 tokens on average), with each sample generated using a distinct hash key (one sample per key). We train Qwen3-0.6B, 1.7B, and 4B on these datasets and evaluate on single-key test data generated using a key distinct from the training data. As shown in Table 6, mixed-key training causes only a slight performance decline compared to in-domain train-

Table 6: Performance of RLCracker trained on Single | Mixed keys data

| Model | Key | EWD | SWE. | PF |
|---|---|---|---|---|
| Qwen3 (0.6B) | Single | 91.5 | 92.5 | 76.8 |
| | Mixed | 88.8 | 90.5 | 75.5 |
| Qwen3 (1.7B) | Single | 91.8 | 91.0 | 79.5 |
| | Mixed | 90.8 | 90.0 | 78.3 |
| Qwen3 (4B) | Single | 93.3 | 95.8 | 82.3 |
| | Mixed | 92.3 | 94.3 | 80.5 |

ing that uses the same key as the test set. The maximum ESR reduction is around 3%, occurring for EWD with the 0.6B model. These results demonstrate that RLCracker maintains strong watermark removal performance even under mixed-key training. Details are in Appendix E.7.

## 6 CONCLUSION AND BROADER IMPACT

**Conclusion.** We introduce the adaptive robustness radius to characterize the worst-case resilience of LLM watermarking schemes and show that its KL approximation can be minimized by optimizing the attack context and model parameters. We further develop RLCracker, a detector-free RL attack that efficiently exploits distributional differences between watermarked and unwatermarked text with only 100 short training samples. RLCracker outperforms baselines and generalizes to long-form text with detection scores close to unwatermarked outputs.

**Broader Impact.** Our findings reveal a secondary use-case: RLCracker functions as a diagnostic stress-test for watermark designers. By implicitly identifying empirical distribution gaps, it provides a practical way to assess whether proposed low-distortion watermarks satisfy their indistinguishability objectives before deployment. We also highlight the underestimated role of prompts and model reasoning in watermark removal and hope our results motivate more rigorous evaluation protocols and more robust watermarking designs.

## ETHICS STATEMENT

This work adheres to the ICLR Code of Ethics. All datasets used are publicly available and widely adopted within the research community. None of the datasets contains personally identifiable information, and all usage complies with their respective licenses. The proposed methodology does not directly pose risks to security, privacy, or legal compliance. The contributions of this work are intended for advancing scientific research and should be deployed with caution in real-world applications. There are no conflicts of interest to declare, and the research has been conducted in accordance with the principles of integrity, transparency, and accountability.

## REPRODUCIBILITY STATEMENT

Comprehensive descriptions of our experimental setup, including implementation details, hyperparameters, and evaluations, are provided in Appendix D and Appendix E, along with additional empirical results and analyses. To ensure transparency and support the reproducibility of our work, we will publicly release the implementation code of our algorithm after the conclusion of the double-blind review period.

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

## A  USE OF LLMs

During the preparation of this manuscript, we employed GPT-4o[1] to assist with language refinement, including improving the clarity, coherence, and academic tone of the writing. However, we emphasize that LLMs were not involved in the development of key components of our work. Specifically, they were not used for designing research ideas, writing critical code, formulating the experimental methodology, analyzing results, or drafting the related works section. All core contributions and technical decisions were made independently by the authors.

## B  RELATED WORKS

### B.1  LLM WATERMARKING.

Watermarking schemes for LLMs, which embed imperceptible but algorithmically detectable patterns in generated text, have become essential for content attribution, copyright protection, and misuse mitigation (Wu et al., 2025; Liu et al., 2024b; Sander et al., 2024). Existing methods can be broadly classified into two dominant paradigms.

The first class of methods, referred to as *logit-based* or *KGW Family* approaches (Hu et al., 2023; Wu et al., 2023; Takezawa et al., 2023; Kirchenbauer et al., 2023a; Liu et al., 2023a; Zhao et al., 2023; Ren et al., 2024), involves partitioning the vocabulary into two distinct categories: "green" and "red" lists, with the partitioning determined by a secret hash key. During the text generation process, these methods modify the model's logits to bias the output towards tokens from the green list, thereby embedding a statistical signature into the generated text without significantly compromising its overall quality. The watermark is detectable by analyzing the frequency of greenlist tokens in the output, with a predefined threshold used to assess the likelihood that a given passage is watermarked (Kirchenbauer et al., 2023a). The primary distinction among methods within the KGW Family lies in the technique used for partitioning the greenlist and redlist tokens. For example, foundational methods (Kirchenbauer et al., 2023a) partition tokens based on prior context tokens, while other approaches in this class refine the watermarking technique by incorporating additional mechanisms, such as entropy-based adjustments (Lu et al., 2024). These refinements aim to enhance the robustness of the watermark while further reducing its detectability. The detection process remains focused on the statistical analysis of token distributions, utilizing the frequency of greenlist tokens as a key metric for watermark identification.

In contrast, the second paradigm, referred to as *sampling-based* or *Christ Family* methods (Christ et al., 2024; Kuditipudi et al., 2023), deviates from logit manipulation by focusing on guiding the sampling process to preferentially select certain tokens during generation. Rather than applying a fixed logit bias, these methods modify the token selection mechanism itself to embed the watermarks. By adapting strategies such as top-k sampling and temperature scaling, Christ-style methods increase the likelihood of selecting watermarked tokens. This approach can produce outputs that are statistically indistinguishable from unwatermarked text and are generally more robust to paraphrasing or other post-processing transformations (Zhao et al., 2024).

Recent advances in both paradigms have focused on minimizing detectable patterns, aiming to make the watermarked output indistinguishable from the natural text (Kuditipudi et al., 2023).

### B.2  LLM WATERMARK REMOVAL ATTACKS.

To evaluate the robustness of watermarking schemes, researchers have explored various watermark removal attacks.

A common approach involves text manipulation using traditional NLP techniques, such as word deletion, substitution, translation (He et al., 2024), or insertion (Kirchenbauer et al., 2023a). However, with advancements in watermarking, these techniques have become less effective (Kirchenbauer et al., 2023b).

Some researchers have extended these attacks by employing large language models (LLMs) in a strict black-box setting, where the attacker's knowledge is limited to the watermarked text. For example,

---

[1]https://chatgpt.com/

Krishna et al. (2023) proposed DIPPER, a paraphrase generation model fine-tuned on an aligned paragraph dataset using T5-XXL (Raffel et al., 2020). This model, along with other approaches like GPT Paraphraser (Liu et al., 2023b), removes watermarks through semantic rewriting.

Other approaches focus on bypassing watermark detectors by modifying specific tokens. Rastogi & Pruthi (2024) achieves watermark removal by predicting and replacing low-confidence tokens. Similarly, SIRA (Cheng et al., 2025) uses training-free paraphrasing templates guided by semantically constrained reference texts, while watermark smoothing attack (Chang et al., 2024) guides token selection during rewriting using a reference model to facilitate watermark removal.

Further methods train proxy models to approximate the watermarking process. WS (Jovanović et al., 2024) mimics the KGW detector to infer watermark patterns, while $B^4$ (Huang et al., 2024) learns the distribution of watermarked text to guide a paraphraser away from watermark-prone tokens.

Other research has framed watermark removal as a preference optimization problem. By constructing positive and negative sample pairs, DPO with reinforcement learning (RL) enhances watermark removal capabilities (Diaa et al., 2024). Additionally, the random walk attack (Zhang et al., 2023) perturbs the watermarked text iteratively using multiple models, relying on detector feedback as the termination condition.

While these methods effectively remove watermarks, they have notable limitations. For instance, GPT, DIPPER, and SIRA exhibit performance drops on longer texts ($\geq$500 tokens) and fail to reveal worst-case vulnerabilities of watermarking schemes. WS is tailored to the KGW watermark and requires large datasets (up to 30,000 samples) for score fitting, limiting generalizability. $B^4$ demands impractically large datasets (up to 100,000 samples), and the iterative nature of the random walk attack introduces significant computational and time overhead. These limitations underscore the urgent need for effective, generalizable, and practical attack strategies to rigorously evaluate the watermark security.

## C  PROOF OF THEOREM 1

For clarity, we restate the main theorem and its components. Let $P_{\mathbf{X},c,\theta}$ be the local watermark distribution on $\mathcal{V}^*$ (i.e., $\mathbf{X}' \sim P_{\mathbf{X},c,\theta}$). The mean detector score is $\mu(\mathbf{X}, s, c, \theta) := \mathbb{E}_{P_{\mathbf{X},c,\theta}}[f(\mathbf{X}', s)]$.

**Theorem 1** (KL lower bound for watermark robustness)**.** *Let the centered score $f(\mathbf{X}', s) - \mu(\mathbf{X}, s, c, \theta)$ be $\sigma^2(\mathbf{X}, s, c, \theta)$-sub-Gaussian under $P_{\mathbf{X},c,\theta}$. For any attack distribution $Q$ with $\mathrm{KL}(Q\|P_{\mathbf{X},c,\theta}) < \infty$, the expected score is bounded by:*

$$\mathbb{E}_{\mathbf{X}'' \sim Q}[f(\mathbf{X}'', s)] \geq \mu(\mathbf{X}, s, c, \theta) - \sqrt{2\,\sigma^2(\mathbf{X}, s, c, \theta)\,\mathrm{D}_{\mathrm{KL}}(Q\|P_{\mathbf{X},c,\theta})}. \tag{2}$$

*This implies that the KL adaptive radius is lower-bounded by a computable certificate $\rho^*$:*

$$r_{\mathrm{KL}}(\mathbf{X}, c, \theta) \geq \rho^*(\mathbf{X}, s, c, \theta) := \frac{[(\mu(\mathbf{X}, s, c, \theta) - \delta)_+]^2}{2\,\sigma^2(\mathbf{X}, s, c, \theta)},$$

*where $(x)_+ = \max\{x, 0\}$.*

The proof relies on the following change-of-measure inequality, which is a direct consequence of the Donsker-Varadhan variational representation of KL divergence.

**Lemma 1** (Donsker-Varadhan Inequality (Donsker & Varadhan, 1975))**.** *Let $P, Q$ be probability measures with $Q \ll P$. For any measurable function $g: \mathcal{V}^* \to \mathbb{R}$ and any $\lambda > 0$:*

$$\mathbb{E}_Q[g] \leq \frac{1}{\lambda}\Big(D_{\mathrm{KL}}(Q\|P) + \log \mathbb{E}_P[e^{\lambda g}]\Big).$$

*Proof.* Let $P = P_{\mathbf{X},c,\theta}$ and $\mu = \mu(\mathbf{X}, s, c, \theta)$. We first establish the bound in Equation 2. Applying Lemma 1 with $g = -(f - \mu)$ and replacing $\lambda$ with $-\lambda$ for $\lambda > 0$ yields:

$$\mathbb{E}_Q[f - \mu] \geq -\frac{1}{\lambda}\Big(\mathrm{KL}(Q\|P) + \log \mathbb{E}_P[e^{-\lambda(f - \mu)}]\Big).$$

By the sub-Gaussian assumption, $\log \mathbb{E}_P[e^{-\lambda(f - \mu)}] \leq \frac{1}{2}\sigma^2(\mathbf{X}, s, c, \theta)\lambda^2$. Substituting this into the inequality gives:

$$\mathbb{E}_Q[f - \mu] \geq -\frac{\mathrm{KL}(Q\|P)}{\lambda} - \frac{\sigma^2(\mathbf{X}, s, c, \theta)\lambda}{2}.$$

The right-hand side is maximized over $\lambda > 0$ at $\lambda^\star = \sqrt{2\,\mathrm{KL}(Q\|P)/\sigma^2(\mathbf{X}, s, c, \theta)}$, yielding the tightest lower bound:

$$\mathbb{E}_Q[f(\mathbf{X}, s)] - \mu \geq -\sqrt{2\,\sigma^2(\mathbf{X}, s, c, \theta)\,\mathrm{KL}(Q\|P)},$$

which proves Equation 2.

Next, to show $r_{\mathrm{KL}} \geq \rho^\star$, assume $\mu > \delta$ (otherwise $\rho^\star = 0$ and the bound is trivial). For any distribution $Q$ such that $\mathrm{KL}(Q\|P) < \rho^\star = \frac{(\mu-\delta)^2}{2\sigma^2(\mathbf{X}, s, c, \theta)}$, Equation 2 implies:

$$\mathbb{E}_Q[f(\mathbf{X}, s)] > \mu - \sqrt{2\sigma^2(\mathbf{X}, s, c, \theta) \cdot \frac{(\mu-\delta)^2}{2\sigma^2(\mathbf{X}, s, c, \theta)}} = \mu - (\mu - \delta) = \delta.$$

By the definition of $r_{\mathrm{KL}}$, this establishes that $r_{\mathrm{KL}}(\mathbf{X}, c, \theta) \geq \rho^\star(\mathbf{X}, s, c, \theta)$.

Let $R_{\mathrm{KL}}(\mathbf{X}, \mathcal{C}, \Theta) = \min_{c \in \mathcal{C},\, \theta \in \Theta} r_{\mathrm{KL}}(\mathbf{X}, c, \theta)$ denote the worst-case KL adaptive radius over context set $\mathcal{C}$ and parameter set $\Theta$. The monotonicity of $R_{\mathrm{KL}}$ follows directly from its definition as an infimum. Since $\mathcal{C} \times \Theta \subseteq \mathcal{C}' \times \Theta'$, the infimum over the larger set cannot be greater than the infimum over the smaller set.

Finally, we note two conditions for a strict decrease in the certified radius. (i) If an adversary finds a strategy $(c', \theta') \in \mathcal{C}' \times \Theta'$ where $\mu(\mathbf{X}, s, c', \theta') \leq \delta$, then $\rho^\star(\mathbf{X}, s, c', \theta') = 0$, causing $R_{\mathrm{KL}}(\mathbf{X}; \mathcal{C}', \Theta') = 0$. (ii) If the sub-Gaussian bound is tight (i.e., the score is truly Gaussian), one can construct an exponentially tilted distribution $Q$ that achieves the bound in Equation 2 with equality. This implies $r_{\mathrm{KL}}(\mathbf{X}, c', \theta') = \rho^\star(\mathbf{X}, s, c', \theta')$, and if this value is smaller than the previous worst-case radius, the radius strictly decreases.

$\square$

## C.1 Justification of Assumption 1

In this subsection, we justify Assumption 1 by demonstrating that the detected z-score is both empirically concentrated and theoretically sub-Gaussian. Figure 4 (b) shows that the detected z-scores of watermarked text (blue) exhibit tight concentration and light-tailed behavior. This is expected: the green-list detector sums many bounded token-level indicators, each contributing only a small centered increment, so the overall score behaves like a normalized sum of bounded differences, even when $\mathbf{X}'$ is produced by a complex sequential generator. Such sums are known to satisfy sub-Gaussian tail bounds under very mild conditions. The next proposition formalizes this intuition and establishes that the normalized green-list z-score is provably sub-Gaussian for *any* sequential text-generation process.

**Proposition 2** (Sub-Gaussianity of the normalized detector score). *Fix the key $s$ and sequence length $n$. Let $\mathbf{X}' = (X_1', \ldots, X_n')$ be any sequentially generated text under the local watermark distribution $P_{\mathbf{X}, c, \theta}$; that is, $X_i'$ is adapted to the filtration $\mathcal{F}_i := \sigma(X_1', \ldots, X_i')$ with no independence or Markov assumptions. For each position $i$, let $Y_i := \mathbf{1}\{X_i' \in \mathcal{G}_i(s)\} \in \{0, 1\}$, and define the variance term $V := \sum_{i=1}^n p_i(1 - p_i)$, $p_i := \mathbb{E}[Y_i]$. Define the normalized z-score as*

$$f(\mathbf{X}', s) := \frac{1}{\sqrt{V}} \sum_{i=1}^n (Y_i - p_i).$$

*Then $f(\mathbf{X}', s) - \mathbb{E}[f(\mathbf{X}', s)]$ is sub-Gaussian with variance proxy $n^2/(4V)$. In particular, for all $t \geq 0$,*

$$\mathbb{P}(|f(\mathbf{X}', s) - \mathbb{E}[f(\mathbf{X}', s)]| \geq t) \leq 2\exp\left(-\frac{2Vt^2}{n^2}\right).$$

**Remark 4** (Insight). *Proposition 2 follows from a simple and general fact: any bounded random variable is sub-Gaussian, and any linear combination of (possibly dependent) bounded variables remains sub-Gaussian with a variance proxy determined by the total range of the sum rather than by independence. Since each summand $(Y_i - p_i)$ lies in $[-1, 1]$, the aggregate fluctuation $\sum_{i=1}^n (Y_i - p_i)$ is itself a bounded random variable supported on an interval of length at most $n$, which directly yields the sub-Gaussian parameter $n^2/4$ used in Proposition 2. Normalizing by $\sqrt{V}$ produces the variance proxy $n^2/(4V)$ for the centered z-score.*

**Remark 5** (Illustration). *A simple example highlights the meaning of the variance proxy. Suppose a single Bernoulli variable $Z \sim \text{Bernoulli}(1/2)$ is repeated across all positions, so $Y_i := Z$ for all $i$. Then*

$$\sum_{i=1}^{n}(Y_i - p_i) = \pm\frac{n}{2}, \qquad V = \frac{n}{4}, \qquad f(\mathbf{X}', s) - \mathbb{E}[f(\mathbf{X}', s)] = \pm\sqrt{n}.$$

*Thus, for any fixed sequence length $n$, the centered z-score takes only two values in the interval $[-\sqrt{n}, \sqrt{n}]$ and is therefore sub-Gaussian with variance proxy $\Theta(n)$, exactly the form captured by $n^2/(4V) = n$ in Proposition 2. This example illustrates that the normalized detector score remains sub-Gaussian even under perfect dependence across positions, though its scale naturally grows with $n$.*

*Proof.* If $V = 0$, then $p_i \in \{0, 1\}$ for all $i$, so each $Y_i$ is almost surely constant and

$$\sum_{i=1}^{n}(Y_i - p_i) = 0 \quad \text{a.s.}$$

Hence $f(\mathbf{X}', s) \equiv 0$, and the stated tail bound holds trivially. Thus, in the remainder of the proof we assume $V > 0$.

Define

$$S := \sum_{i=1}^{n}(Y_i - p_i).$$

By linearity of expectation and the definition $p_i = \mathbb{E}[Y_i]$, we have

$$\mathbb{E}[S] = \sum_{i=1}^{n}\mathbb{E}[Y_i - p_i] = \sum_{i=1}^{n}\big(\mathbb{E}[Y_i] - p_i\big) = 0,$$

so $S$ is a centered random variable. Next, we bound the range of $S$. Since $Y_i \in \{0, 1\}$, we have

$$Y_i - p_i \in \{-p_i,\, 1 - p_i\} \subseteq [-1, 1] \quad \text{for each } i.$$

Therefore

$$S = \sum_{i=1}^{n}(Y_i - p_i) \in \Big[ -\sum_{i=1}^{n}p_i,\ \sum_{i=1}^{n}(1 - p_i)\Big] \quad \text{almost surely,}$$

and the length of this interval is

$$\Big(\sum_{i=1}^{n}(1 - p_i)\Big) - \Big(-\sum_{i=1}^{n}p_i\Big) = \sum_{i=1}^{n}\big[(1 - p_i) + p_i\big] = n.$$

In particular, $S$ is a centered random variable supported on an interval of length at most $n$.

By Hoeffding's lemma (Hoeffding, 1963), for a single bounded random variable, if a random variable $Z$ satisfies $\mathbb{E}[Z] = 0$ and $Z \in [a, b]$ almost surely, then for all $\lambda \in \mathbb{R}$,

$$\mathbb{E}\big[e^{\lambda Z}\big] \leq \exp\Big(\frac{\lambda^2(b - a)^2}{8}\Big).$$

Applying this to $Z := S$ with $b - a \leq n$ yields

$$\mathbb{E}\big[e^{\lambda S}\big] \leq \exp\Big(\frac{\lambda^2 n^2}{8}\Big) \quad \text{for all } \lambda \in \mathbb{R}.$$

Thus $S$ is sub-Gaussian with variance proxy $n^2/4$, in the sense that

$$\mathbb{P}(|S| \geq u) \leq 2\exp\Big(-\frac{2u^2}{n^2}\Big) \quad \text{for all } u \geq 0.$$

By definition, we have

$$f(\mathbf{X}', s) - \mathbb{E}[f(\mathbf{X}', s)] = \frac{1}{\sqrt{V}}\sum_{i=1}^{n}(Y_i - p_i) = \frac{S}{\sqrt{V}}.$$

Therefore, for any $t \geq 0$,

$$\mathbb{P}\big(|f(\mathbf{X}', s) - \mathbb{E}[f(\mathbf{X}', s)]| \geq t\big) = \mathbb{P}\big(|S| \geq t\sqrt{V}\big) \leq 2\exp\Big(-\frac{2(t^2 V)}{n^2}\Big) = 2\exp\Big(-\frac{2V t^2}{n^2}\Big).$$

This shows that $f(\mathbf{X}', s) - \mathbb{E}[f(\mathbf{X}', s)]$ is sub-Gaussian with variance proxy $n^2/(4V)$ and establishes the claimed tail bound. $\qquad\square$

# D  EXPERIMENTAL SETUP AND CONFIGURATION

## D.1  WATERMARK ALGORITHM SETTING

In this subsection, we detail the hyperparameter settings used for the watermarking algorithm evaluated in Section 5. To ensure consistency and reproducibility, we adopt the default hyperparameter configuration provided by the publicly available and widely used MarkLLM toolkit (Pan et al., 2024)[2]. This toolkit is recognized in the community for its robustness and ease of integration, and has been frequently employed in prior watermarking studies.

---

**Hyperparameters for the KGW watermark**

"algorithm_name": "KGW",
"gamma": 0.5,
"delta": 2.0,
"hash_key": 15485863,
"prefix_length": 4,
"z_threshold": 4.0,
"f_scheme": "time",
"window_scheme": "left"

---

**Hyperparameters for the KGW_selfhash watermark**

"algorithm_name": "KGW",
"gamma": 0.25,
"delta": 2.0,
"hash_key": 15485863,
"prefix_length": 4,
"z_threshold": 4.0,
"f_scheme": "min",
"window_scheme": "self"

---

**Hyperparameters for the EWD watermark**

"algorithm_name": "EWD",
"gamma": 0.5,
"delta": 2.0,
"hash_key": 15485863,
"prefix_length": 1,
"z_threshold": 4.0

---

**Hyperparameters for the SIR watermark**

"algorithm_name": "SIR",
"delta": 1.0,
"chunk_length": 10,
"scale_dimension": 300,
"z_threshold": 0.2,
"transform_model_input_dim": 1024,
"transform_model_name": "watermark/sir/model/transform_model_cbert.pth",
"embedding_model_path": "watermark/sir/model/compositional-bert-large-uncased/",
"mapping_name": "watermark/sir/mapping/300_mapping_128256.json"

---

[2]https://github.com/THU-BPM/MarkLLM

**Hyperparameters for the Unigram watermark**

```
"algorithm_name": "Unigram",
"gamma": 0.5,
"delta": 2.0,
"hash_key": 15485863,
"z_threshold": 4.0
```

**Hyperparameters for the PF watermark**

```
"algorithm_name": "PF",
"ngram": 8,
"seed": 0,
"seeding": "hash",
"salt_key": 35317,
"payload": 0,
"max_seq_len": 2048
```

**Hyperparameters for the SWEET watermark**

```
"algorithm_name": "SWEET",
"gamma": 0.5,
"delta": 2.0,
"hash_key": 15485863,
"z_threshold": 4.0,
"prefix_length": 1,
"entropy_threshold": 0.9
```

**Hyperparameters for the SynthID-Text watermark**

```
"algorithm_name": "SynthID",
"ngram_len": 5,
"keys": [ 654, 400, 836, 123, 340, 443, 597, 160, 57, 29, 590, 639, 13, 715, 468, 990, 966,
226, 324, 585, 118, 504, 421, 521, 129, 669, 732, 225, 90, 960 ],
"sampling_table_size": 65536,
"sampling_table_seed": 0,
"watermark_mode": "non-distortionary",
"num_leaves": 2,
"context_history_size": 1024,
"detector_type": "mean",
"threshold": 0.52
```

**Hyperparameters for the UPV watermark**

```
"algorithm_name": "UPV",
"gamma": 0.5,
"delta": 2.0,
"z_threshold": 4.0,
"prefix_length": 1,
"bit_number": 16,
"sigma": 0.01,
"default_top_k": 20,
"generator_model_name": "watermark/upv/model/generator_model_b16_p1.pt",
"detector_model_name": "watermark/upv/model/detector_model_b16_p1_z4.pt",
"detect_mode": "network"
```

> **Hyperparameters for the XSIR watermark**
>
> "algorithm_name": "XSIR",
> "delta": 1.0,
> "chunk_length": 10,
> "scale_dimension": 300,
> "z_threshold": 0.2,
> "transform_model_input_dim": 768,
> "dictionary": "watermark/xsir/dictionary/dictionary.txt",
> "transform_model_name": "watermark/xsir/model/transform_model_x-sbert_10K.pth",
> "embedding_model_path": "watermark/xsir/model/paraphrase-multilingual-mpnet-base-v2",
> "mapping_name": "watermark/xsir/mapping/300_mapping_llama_Ins.json"

## D.2 BASELINES.

We consider six distinct watermark removal (attack) methods. Their implementation details are described below:

**Base.** The Base method performs direct paraphrasing of watermarked samples using simple prompting. Specifically, we wrap the watermarked text using the following template:

> **Prompt template used in the Base method**
>
> ######Target Text: [watermarked text]
> ######Instruction: Rewrite the target text above using different words but keeping the same meaning and similar length.
> ######Your Response:

**Think.** The Think method is built on the Base method by enabling the model's reasoning capability to produce more diverse and semantically rich paraphrases. We use the same wrapping prompt as in the Base method.

**SysP.** The SysP method further enriches the prompting context by applying a system prompt in addition to the base prompt. Specifically, we use the standard chat template and prepend the following system message:

> *You are an AI assistant skilled in rewriting prompts in diverse and effective ways.*
> *You can provide well-structured and detailed rewordings that maintain the original*
> *meaning while improving clarity and variety.*

**SIRA.** In our implementation, we strictly adopt the hyperparameter settings from the original SIRA paper (Cheng et al., 2025). In particular, the self-information masking threshold is set to $\varepsilon = 0.30$, meaning that tokens whose self-information exceeds the 30th percentile are masked during the attack. All algorithm implementations and watermarking applications are based on the official SIRA code[3]. For each watermarking scheme, we utilize the **same model** both to compute self-information and to conduct the attack. This alignment ensures that our robustness evaluation is conducted under a threat model where the adversary uses the same model architecture for attack and estimation, but is still constrained in practical terms (e.g. limited resources).

**DIPPER.** We adopt the DIPPER (Krishna et al., 2023) paraphrase model as one of our baseline methods. We adopt the same settings following (Cheng et al., 2025), setting lexical diversity = 60 and order diversity = 40. These parameters respectively control the extent of vocabulary variation and the degree of reordering of sentences or content segments. Such settings allow us to test watermarking schemes under more challenging rewriting attacks while still preserving text coherence and meaning.

**LLM Paraphraser.** In this setting, we utilize GPT-4o-2024-08-06 as the paraphrasing model, with its performance assessed under the prompt configuration of our Base method.

---

[3]https://github.com/Allencheng97/Self-information-Rewrite-Attack

**Implementation Details.** All methods—Base, Think, SysP, and SIRA—are executed using the vLLM inference engine to accelerate generation. We set the sampling parameters as follows: temperature = 0.7, top_p = 0.95, and top_k = 20.

**Think + SysP.** Additionally, we evaluate a combined method that integrates both the system prompt and reasoning capability (i.e., Think + SysP). Results, presented in Tables 8, 20, 22, 24, 26, 28, show that combining these two techniques further improves attack performance. This aligns with our theoretical finding that enriching the attack context reduces the adaptive robustness radius of watermarking schemes.

### D.3 IMPLEMENTATION DETAILS OF RLCRACKER.

Recall that our RLCracker method adopts a token-wise optimization framework inspired by GRPO (Shao et al., 2024), requiring only question–watermarked response pairs $(q, wr)$ to effectively learn watermark evasion policies. The RLCracker is implemented based on the GRPO component from the Hugging Face TRL library (von Werra et al., 2020), where we directly adopt the implementation of the KL divergence term $D_{\mathrm{KL}}[\pi_\theta|\pi_{ref}]$. Following the KL divergence setting in OpenR1 project (Hugging Face, 2025), we use $\beta = 0.04$ to control the distributional shift between the policy model and the reference model.

The attack policy $\pi_\theta$ is updated iteratively by sampling outputs $\{o_1, \ldots, o_G\}$ from the previous policy $\pi_{\theta_{\mathrm{old}}}$ and maximizing the following training objective:

$$
\mathcal{J}(\theta) \approx \mathbb{E}_{\{o_i\} \sim \pi_{\theta_{\mathrm{old}}}} \frac{1}{G} \sum_{i=1}^{G} \frac{1}{|o_i|} \sum_{t=1}^{|o_i|} \Bigg[ \frac{\pi_\theta(o_{i,t} \mid q, o_{i,<t})}{\pi_{\theta_{\mathrm{old}}}(o_{i,t} \mid q, o_{i,<t})} \big(w_1 \hat{A}_i + w_2 \Delta r_{i,t}\big)
$$
$$
- \beta D_{\mathrm{KL}}\big[\pi_\theta \| \pi_{\mathrm{ref}}\big] \Bigg] - w_3 \mathrm{PPL}(\pi_\theta, \{o_i\}),
\tag{3}
$$

where $\Delta r_{i,t} = D_{\mathrm{KL},i,t}(Q\|P_{wm}) - D_{\mathrm{KL},i,t}(Q\|P_h)$, $\pi_{\mathrm{ref}}$ denotes the reference policy approximating human-like distributions, and $w_1, w_2, w_3$ are the weights of the respective components. The remaining terms impose token-wise KL regularization and a perplexity penalty to preserve fluency. Here, we use the original state of the policy model as our reference model $\pi_{\mathrm{ref}}$. Notably, to better fit $P_{wm}$, we use the Qwen3-0.6B model as the reference for the KL rewards during the training of Qwen3-8B.

**Reward Components.** We incorporate two reward signals to encourage semantic preservation and effective watermark evasion:

- **Semantic Reward.** The semantic reward $A_i \in [0, 1]$ is computed based on the P-SP score (Wieting et al., 2021) between the generated output and the original watermarked response. To enhance gradient flow and emphasize semantic fidelity, we apply a sigmoid-based scaling with a threshold of 0.85. This maps P-SP scores in the range [0.7, 1.0] to reward values between -1 and 1. Specifically,

$$
A_i = \frac{1}{1 + e^{-x}}, \quad \text{where } x = \log\left(\frac{0.975}{0.025}\right) \cdot \text{semantic\_score}.
$$

  The advantage term is normalized as $\hat{A}_i = \frac{A_i - \mathrm{mean}(A)}{\mathrm{std}(A)}$.

- **Token-wise KL Reward.** This reward encourages each token's distribution under $\pi_\theta$ to align with the human-like reference while diverging from watermark-induced patterns. The token-wise KL term is estimated using the standard PPO-style estimator (Schulman, 2020):

$$
D_{\mathrm{KL},i,t}(Q\|P_*) \approx \frac{\pi_{\mathrm{ref}}(o_{i,t} \mid *, o_{i,<t})}{\pi_\theta(o_{i,t} \mid wr, o_{i,<t})} - \log \frac{\pi_{\mathrm{ref}}(o_{i,t} \mid *, o_{i,<t})}{\pi_\theta(o_{i,t} \mid wr, o_{i,<t})} - 1,
$$

  where $* \in \{h, wm\}$ specifies whether the reference distribution corresponds to the human-like distribution $P_h$ or the watermark-induced distribution $P_{wm}$. In practice, we approximate $P_{wm}$ by passing the watermarked instance $wr$ through the reference model $\pi_{\mathrm{ref}}$ with a simple paraphrasing prompt, and we approximate $P_h$ by querying the same reference model with the original question $q$. During training, we use the *same* instruction prompt as the *Base method* to ensure consistency with prior watermarking setups.

**Reward Balancing and Dynamic Scaling.** We introduce three coefficients $w_1$, $w_2$, and $w_3$ to weight the contributions of the semantic advantage, KL reward, and perplexity penalty, respectively. To better balance semantic fidelity and watermark evasion effectiveness, we adopt a dynamic scaling strategy for $w_1$: $w_1 = \max\left(w_1' \cdot (1 - \text{mean}(A_i)), 1\right)$, which increases the emphasis on semantic alignment when P-SP scores are low, and encourages distributional exploration when semantic fidelity is already high.

**Hyperparameters.** For each watermarking scheme, we train a dedicated RLCracker model. The training data consists of 100 watermarked samples, each containing 500 tokens. We train for 10 epochs using a batch size of 48 and a group size of $G = 12$, with a cosine learning rate scheduler. For the Qwen3-4B model, we set the learning rate to $2 \times 10^{-7}$, while for the remaining three models, we use a learning rate of $1 \times 10^{-6}$. Training takes approximately 1.5 hours for Qwen3-4B and 0.5 hours for Qwen3-0.6B when using four NVIDIA A100 GPUs.

We find that RLCracker is quite sensitive to the values of the weights $w_1'$, $w_2$, and $w_3$: changes in these weights lead to significant variation in performance. As shown in Figure 5, we trained Qwen-3-4B to remove the EWD watermark, and recorded how both the removal rate and the PSP score vary when each of the three weights is changed. Here, the removal rate is defined as the proportion of watermark that is removed, regardless of semantic fidelity. When testing $w_1'$, we fix $w_2 = 0.9$ and $w_3 = 0.1$; when testing $w_2$, we fix $w_1' = 12$ and $w_3 = 0.1$; when testing $w_3$, we fix $w_1' = 12$ and $w_2 = 0.9$. From these experiments, we observe that setting $(w_1' = 12, \ w_2 = 0.9, \ w_3 = 0.1)$ yields strong performance: the model achieves a high semantic similarity (P-SP score of 0.87) while also maintaining a removal rate of 95%, resulting in an ESR of 93.5%. To facilitate reproducibility, we list the specific weight configurations used for each watermarking scheme and attack model in Table 7.

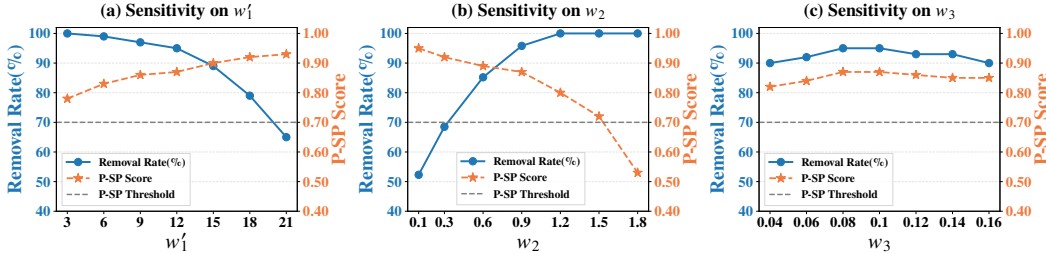

Figure 5: Sensitivity of RLCracker to Weight Settings under Qwen-3-4B for EWD Watermark

Table 7: Hyperparameters used for RLCracker across models and watermarking schemes.

| Model | Weight | EWD | PF | SWEET | Unigram | KGW | KGW_self | SIR | XSIR | UPV | SynthID |
|---|---|---|---|---|---|---|---|---|---|---|---|
| Qwen3-0.6B | $w_1'$ | 12 | 12 | 12 | 5 | 6 | 8 | 12 | 12 | 12 | 12 |
| | $w_2$ | 0.9 | 0.9 | 0.9 | 0.08 | 0.1 | 0.5 | 0.9 | 0.9 | 0.9 | 0.9 |
| | $w_3$ | 0.1 | 0.1 | 0.1 | 0.1 | 0.1 | 0.1 | 0.1 | 0.1 | 0.1 | 0.1 |
| Qwen3-1.7B | $w_1'$ | 12 | 12 | 12 | 5 | 6 | 8 | 12 | 12 | 12 | 12 |
| | $w_2$ | 0.9 | 0.9 | 0.9 | 0.08 | 0.1 | 0.5 | 0.9 | 0.9 | 0.9 | 0.9 |
| | $w_3$ | 0.1 | 0.1 | 0.1 | 0.1 | 0.1 | 0.1 | 0.1 | 0.1 | 0.1 | 0.1 |
| Qwen3-4B | $w_1'$ | 12 | 12 | 12 | 5 | 6 | 8 | 12 | 12 | 12 | 12 |
| | $w_2$ | 0.9 | 0.9 | 0.9 | 0.08 | 0.1 | 0.5 | 0.9 | 0.9 | 0.9 | 0.9 |
| | $w_3$ | 0.1 | 0.1 | 0.1 | 0.1 | 0.1 | 0.1 | 0.1 | 0.1 | 0.1 | 0.1 |
| Qwen3-8B | $w_1'$ | 12 | 12 | 12 | 5 | 6 | 8 | 12 | 12 | 12 | 12 |
| | $w_2$ | 0.9 | 0.9 | 0.9 | 0.08 | 0.1 | 0.5 | 0.9 | 0.9 | 0.9 | 0.9 |
| | $w_3$ | 0.1 | 0.1 | 0.1 | 0.1 | 0.1 | 0.1 | 0.1 | 0.1 | 0.1 | 0.1 |
| Qwen2.5-3B -Instruct | $w_1'$ | 0 | 0 | 0 | 0 | 0 | 0 | 0 | 0 | 0 | 0 |
| | $w_2$ | 0.9 | 0.6 | 0.8 | 0.03 | 0.1 | 0.4 | 0.7 | 0.7 | 0.8 | 0.5 |
| | $w_3$ | 0.1 | 0.1 | 0.1 | 0.1 | 0.1 | 0.1 | 0.1 | 0.1 | 0.1 | 0.1 |

## E  EXTENDED EXPERIMENTAL RESULTS AND ANALYSIS

### E.1  ROBUSTNESS OF WATERMARKING SCHEMES ACROSS MODELS AND METHODS.

We present detailed watermark-evasion results for various methods across different models and watermarking schemes, evaluated on two lengths of watermarked texts (500 and 1500 tokens). We compare the following methods: Base, SysP, Think, Think+SysP, SIRA, and our proposed RLCracker. Each table reports, for each method and watermark scheme, the evasion success rate (ESR), the Removal Rate (Rem.), the proportion of watermark removed regardless of semantic similarity, and the average P-SP score after rephrasing. We also use perplexity (PPL.) and ChatGPT as a judge (GPTScore) to evaluate the quality of rephrased texts. Specifically, we use Llama3.1-8B-Instruct as the base model for measuring perplexity. For ChatGPT as a judge, we use GPT-4o-2024-08-06. The prompt template is as follows:

> **Prompt template for ChatGPT as a Judge**
>
> You are an impartial evaluator. You will be provided with an original text and a paraphrased version of that text. "par Your task is to assess the paraphrased text based on the following criteria: "par 1. The degree to which the paraphrased text maintains the meaning of the original text. "par 2. The fluency, coherence, and clarity of the paraphrased text. "par 3. The extent to which the style and tone of the paraphrased text match those of the original text. "par Please provide an objective evaluation and rate the paraphrased text on a scale of 1 to 10: "par - A rating of 1 indicates that the paraphrased text completely alters the meaning of the original text. "par - A rating of 10 indicates that the paraphrased text preserves the exact meaning of the original text, maintaining fluency, clarity, and style. "par Your response should follow this strict format and return the score only:   [watermarked text].
> Here's the original text:  [watermarked text]
> Here's the paraphrased text:  [rephrased text]

**Notably, the performance of RLCracker is evaluated using the same prompt as the SysP. method with the system prompt.**

As shown in Tables 8, 20, 22, 24, 26, 28, under the Base method, smaller models ($\leq$ 4B parameters) consistently exhibit low ESR. For example, Qwen3-0.6B attains approximately 13.5% ESR on 500-token EWD texts, but only about 3.5% for 1500-token EWD, with a removal rate of 6.2%. These observations align with prior work (Kirchenbauer et al., 2023b). At the same time, enriching the attack context via SysP, Think, or their combination (Think+SysP) yields substantial improvements in ESR, in agreement with Theorem 1. Moreover, as shown in Tables 19, 21, 23, 25, 27, 29, all four methods maintain high semantic similarity, as evidenced by PPL remaining close to that of the original watermarked text, and the GPT score consistently exceeding 8. However, the extent of improvement varies across watermark schemes, and no single method proves to be uniformly optimal.

We also observe that SIRA (Cheng et al., 2025) maintains low ESR across different schemes. Although its removal rate is relatively high, its average P-SP scores remain low, which limits its overall effectiveness. Furthermore, its PPL, which exceeds that of the original text, and its GPT score below 5 indicate that the generated text quality is suboptimal. In contrast, RLCracker consistently achieves strong performance across all models, delivering high removal rates while preserving high average P-SP. When using the same prompt, RLCracker outperforms all baselines, and in some settings even surpasses GPT-4o in ESR, demonstrating its superior effectiveness.

### E.2  IMPACT OF PROMPTING STRATEGIES

We evaluate the impact of different prompting strategies using eight distinct prompts. Each prompt is assigned an index, as shown in the corresponding table. As shown in Section 5, the choice of prompt leads to substantial variation in model ESR, underscoring the importance of considering prompt diversity when assessing watermark robustness. Only by validating vulnerability under various prompts can we obtain a complete picture of a watermarking scheme's robustness.

The eight user prompts are as follows (index and content):

Table 8: Evasion Success Rate (ESR, %) across models and watermarking schemes (500 tokens).

| Model | Method | EWD | | | SWEET | | | XSIR | | | Unigram | | |
|---|---|---|---|---|---|---|---|---|---|---|---|---|---|
| | | ESR | Rem. | P-SP | ESR | Rem. | P-SP | ESR | Rem. | P-SP | ESR | Rem. | P-SP |
| Qwen3-0.6B | Base | 13.5 | 14.8 | 0.94 | 21.2 | 22.2 | 0.94 | 29.0 | 31.2 | 0.92 | 35.8 | 64.8 | 0.78 |
| | SysP. | 36.5 | 39.0 | 0.91 | 44.8 | 46.8 | 0.93 | 52.5 | 55.8 | 0.90 | 42.5 | 82.2 | 0.73 |
| | Think | 28.5 | 33.0 | 0.91 | 35.9 | 40.6 | 0.92 | 37.2 | 40.8 | 0.91 | 34.2 | 76.8 | 0.71 |
| | Think+SysP. | 48.2 | 53.2 | 0.89 | 58.8 | 62.5 | 0.91 | 50.7 | 55.0 | 0.89 | 37.5 | 88.8 | 0.67 |
| | SIRA | 0.0 | 40.2 | 0.25 | 0.0 | 47.8 | 0.16 | 0.0 | 0.0 | 0.31 | 0.0 | 60.2 | 0.46 |
| | RLCracker | 91.5 | 98.5 | 0.83 | 92.5 | 99.0 | 0.82 | 86.5 | 94.5 | 0.83 | 67.0 | 71.5 | 0.86 |
| Qwen3-1.7B | Base | 34.2 | 35.0 | 0.94 | 42.8 | 44.0 | 0.94 | 43.0 | 45.0 | 0.93 | 34.0 | 69.0 | 0.76 |
| | SysP. | 59.8 | 61.0 | 0.92 | 70.8 | 72.2 | 0.92 | 70.8 | 73.0 | 0.89 | 40.5 | 81.5 | 0.73 |
| | Think | 80.8 | 86.2 | 0.85 | 88.0 | 93.8 | 0.85 | 75.5 | 83.2 | 0.83 | 31.0 | 96.8 | 0.64 |
| | Think+SysP. | 84.0 | 89.0 | 0.85 | 86.5 | 92.0 | 0.85 | 79.8 | 87.2 | 0.83 | 34.8 | 97.8 | 0.65 |
| | SIRA | 0.2 | 86.0 | 0.28 | 0.0 | 91.2 | 0.19 | 0.2 | 85.5 | 0.33 | 0.0 | 93.5 | 0.39 |
| | RLCracker | 91.8 | 97.8 | 0.83 | 91.0 | 97.2 | 0.86 | 86.0 | 96.5 | 0.81 | 73.0 | 76.5 | 0.81 |
| Qwen3-4B | Base | 39.2 | 41.2 | 0.94 | 53.2 | 55.0 | 0.94 | 50.2 | 52.2 | 0.94 | 39.2 | 54.5 | 0.84 |
| | SysP. | 54.2 | 54.8 | 0.94 | 65.8 | 67.5 | 0.94 | 66.8 | 69.8 | 0.92 | 43.8 | 72.2 | 0.79 |
| | Think | 76.0 | 80.0 | 0.87 | 83.5 | 88.5 | 0.88 | 75.2 | 80.5 | 0.86 | 38.2 | 82.0 | 0.72 |
| | Think+SysP. | 78.8 | 83.0 | 0.87 | 88.0 | 91.0 | 0.88 | 79.2 | 85.0 | 0.85 | 38.8 | 85.0 | 0.72 |
| | SIRA | 0.0 | 78.5 | 0.28 | 0.0 | 83.0 | 0.18 | 0.2 | 79.8 | 0.33 | 0.8 | 89.2 | 0.41 |
| | RLCracker | 93.3 | 95.8 | 0.87 | 95.8 | 98.8 | 0.85 | 85.5 | 93.5 | 0.85 | 73.5 | 75.2 | 0.87 |
| Qwen3-8B | Base | 72.0 | 73.5 | 0.91 | 78.4 | 81.8 | 0.92 | 77.0 | 79.8 | 0.90 | 40.5 | 63.7 | 0.82 |
| | SysP. | 72.8 | 73.8 | 0.92 | 81.5 | 83.0 | 0.92 | 82.0 | 83.8 | 0.89 | 43.5 | 71.5 | 0.80 |
| | Think | 89.5 | 94.0 | 0.85 | 92.0 | 95.8 | 0.86 | 81.5 | 87.8 | 0.83 | 40.8 | 82.2 | 0.73 |
| | Think+SysP. | 90.2 | 95.2 | 0.85 | 92.0 | 96.5 | 0.85 | 86.8 | 93.8 | 0.82 | 41.0 | 87.8 | 0.72 |
| | SIRA | 0.0 | 80.2 | 0.29 | 0.0 | 84.8 | 0.19 | 0.2 | 81.0 | 0.34 | 0.5 | 86.0 | 0.45 |
| | RLCracker | 94.8 | 95.5 | 0.88 | 96.3 | 98.0 | 0.88 | 90.2 | 92.8 | 0.88 | 80.5 | 88.3 | 0.80 |
| Qwen2.5-3B -Instruct | Base | 31.2 | 42.0 | 0.89 | 46.8 | 53.5 | 0.90 | 68.8 | 71.3 | 0.90 | 37.2 | 58.5 | 0.81 |
| | SysP. | 75.5 | 78.5 | 0.90 | 83.0 | 88.8 | 0.90 | 78.5 | 81.0 | 0.89 | 43.5 | 72.0 | 0.80 |
| | SIRA | 0.2 | 76.8 | 0.28 | 0.2 | 90.8 | 0.19 | 0.2 | 81.5 | 0.33 | 0.8 | 96.5 | 0.40 |
| | RLCracker | 93.3 | 98.0 | 0.85 | 95.3 | 99.0 | 0.85 | 89.8 | 92.2 | 0.84 | 78.5 | 90.2 | 0.81 |
| GPT-4o | — | 71.0 | 72.2 | 0.93 | 81.5 | 83.2 | 0.93 | 77.2 | 79.2 | 0.90 | 49.8 | 89.8 | 0.73 |
| DIPPER | — | 30.3 | 31.5 | 0.82 | 52.8 | 56.3 | 0.83 | 0.0 | 0.0 | 0.83 | 33.3 | 78.8 | 0.65 |

1. ######Instruction: Paraphrase the above textbfs.######Your Response:

2. ######Instruction: You are an expert copyeditor. Please rewrite the target text above in your own voice and paraphrase all sentences. Ensure that the final output contains the same information as the original text and has roughly the same length. Do not leave out any important details when rewriting in your own voice. ######Your Response:

3. ######Instruction: As an expert copyeditor, please rewrite the target text above in your own voice while ensuring that the final output contains the same information as the original text and has roughly the same length. Please paraphrase all sentences and do not omit any crucial details. Additionally, please take care to provide any relevant information about public figures, organizations, or other entities mentioned in the text to avoid any potential misunderstandings or biases. ######Your Response:

4. ######Instruction: Rewrite the target text above using different words but keeping the same meaning and similar length. ######Your Response:

5. ######Instruction: Paraphrase the target text above without changing its meaning or length. ######Your Response:

6. ######Instruction: Restate the target text above using different wording. Keep the meaning and text length nearly the same. ######Your Response:

7. ######Instruction: Generate a version of the target text above that means the same and is about the same length. ######Your Response:

8. ######Instruction: Write a similar target text to the one above, keeping both the meaning and the number of words roughly the same. ######Your Response:

To further enrich the attack context, we incorporate a system prompt alongside the user prompt to guide the model's performance from a system perspective. Specifically, we employ the same system prompt used in the SysP method, which is as follows:

> *You are an AI assistant skilled in rewriting prompts in diverse and effective ways. You can provide well-structured and detailed rewordings that maintain the original meaning while improving clarity and variety.*

### E.3 REASONING AND TEST-TIME SCALING

In Section 5.2, we empirically validate the effectiveness of test-time scaling, previously shown to be theoretically beneficial in mathematical reasoning tasks (Muennighoff et al., 2025), for watermark removal in natural language text. Specifically, we evaluate on test sets constructed from watermarked data (500-token sequences) generated using SWEET, EWD, SIR, and PF watermarking schemes.

In each trial, we truncate the reasoning text generated by the Qwen3-8B model just before the final output and append the following prompt: *"Wait, I should rethink and reformulate it using different terms and structure, same meaning and same length."* This prompt encourages the model to continue reasoning in a deeper and more varied manner, effectively simulating the test-time scaling mechanism. We conduct all generations using the vLLM framework with the following decoding settings: temperature = 0.7, top-$p$ = 0.95, and top-$k$ = 20. The results are presented in Figure 6.

We observe that as the number of rethinking steps increases, both the ESR and the removal rate improve across all four watermarking schemes. However, the average P-SP score shows a sharp drop once rethinking is initiated, and only fluctuates mildly in subsequent steps. This drop is expected: the P-SP score positively correlates with watermark preservation. A higher P-SP score indicates higher semantic similarity and potentially lexical overlap with the original text, which in turn implies that the watermark is more likely to be preserved. As the model begins to rethink, the watermarked tokens are often rephrased or replaced, leading to reduced semantic similarity and hence a decrease in the P-SP score.

The consistent rise in ESR during this process demonstrates the efficacy of test-time scaling in weakening watermark robustness. This finding aligns well with our Theorem 1, which suggests that richer generation contexts can degrade watermark stability.

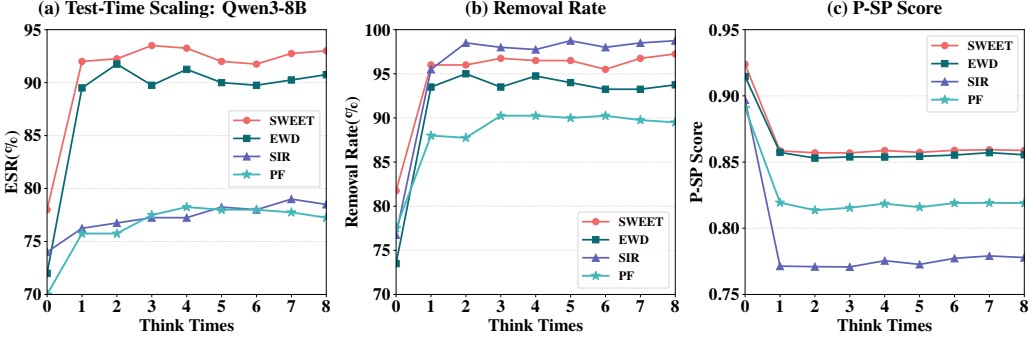

Figure 6: (a) shows test-time scaling on Qwen3-8B, (b) shows remove rate on Qwen3-8B, (c) shows p-sp score on Qwen3-8B.

### E.4 DETECTION SCORE DISTRIBUTION

The detection scores shown in Figure 4(b) indicate that RLCracker effectively pushes the distribution of rephrased texts toward that of unwatermarked outputs, and away from the watermarked distribution. To obtain the unwatermarked distribution, we use LLaMA3.1-8B-Instruct to generate outputs on the same test set without watermarking, and compute their detection scores. For the rephrased text distribution, we use detection scores of paraphrased outputs generated by Qwen3-4B trained with RLCracker. The clear separation between the rephrased and watermarked distributions further demonstrates the effectiveness of RLCracker in removing watermarks, highlighting a fundamental vulnerability in current watermarking schemes.

## E.5 RLCRACKER GENERALIZATION TO OUT-OF-DISTRIBUTION SETTINGS

In Section 5.3, we empirically evaluate whether models trained on data generated by LLaMA3.1-8B can effectively remove watermark patterns embedded in texts produced by other models under the same key $s$. Specifically, we consider three watermarking schemes (EWD, SWEET, and PF) and generate 400 test samples of 500 tokens each using Qwen2.5-1.5B-Instruct and Qwen2.5-32B-Instruct, following the watermarking settings described in Appendix C. We then assess model performance under both the base method and RLCracker, using the following paraphrasing prompt: ######Instruction: Generate a version of the target text above that means the same and is about the same length. ######Your Response: The detailed results are provided in Table 9 and Table 10.

Our findings show that RLCracker generalizes effectively to out-of-distribution (OOD) data, consistently achieving high Effective Semantic Rewriting (ESR) scores and strong watermark removal performance. The ESR achieved by RLCracker is generally higher when targeting outputs from larger models, which may be attributed to their higher quality and more coherent generations. These results further confirm both the effectiveness and the strong generalization ability of RLCracker, while also highlighting a fundamental vulnerability of current watermarking schemes.

Table 9: Performance of RLCracker on Qwen2.5-1.5B-Instruct generated data (500 tokens).

| Model | Method | EWD | | | SWEET | | | PF | | |
|---|---|---|---|---|---|---|---|---|---|---|
| | | ESR | Rem. | P-SP | ESR | Rem. | P-SP | ESR | Rem. | P-SP |
| Qwen3-0.6B | Base | 10.8 | 16.0 | 0.89 | 13.0 | 20.0 | 0.89 | 14.5 | 21.3 | 0.84 |
| | RLCracker | 87.3 | 98.5 | 0.81 | 84.0 | 99.0 | 0.79 | 69.3 | 96.0 | 0.75 |
| Qwen3-1.7B | Base | 22.3 | 28.0 | 0.89 | 25.3 | 31.5 | 0.90 | 29.8 | 37.5 | 0.86 |
| | RLCracker | 82.3 | 99.3 | 0.78 | 74.3 | 99.5 | 0.76 | 65.3 | 92.0 | 0.74 |
| Qwen2.5-3B -Instruct | Base | 22.8 | 34.0 | 0.86 | 28.0 | 39.8 | 0.86 | 26.8 | 40.0 | 0.92 |
| | RLCracker | 86.0 | 99.5 | 0.80 | 84.8 | 98.8 | 0.79 | 73.5 | 94.5 | 0.76 |
| Qwen3-4B | Base | 21.0 | 26.5 | 0.90 | 22.8 | 36.8 | 0.88 | 36.0 | 45.3 | 0.86 |
| | RLCracker | 84.0 | 96.5 | 0.81 | 83.0 | 96.8 | 0.81 | 73.5 | 93.8 | 0.77 |

Table 10: Performance of RLCracker on Qwen2.5-32B-Instruct generated data (500 tokens).

| Model | Method | EWD | | | SWEET | | | PF | | |
|---|---|---|---|---|---|---|---|---|---|---|
| | | ESR | Rem. | P-SP | ESR | Rem. | P-SP | ESR | Rem. | P-SP |
| Qwen3-0.6B | Base | 12.5 | 13.3 | 0.94 | 15.8 | 16.5 | 0.95 | 14.3 | 20.8 | 0.88 |
| | RLCracker | 96.0 | 99.8 | 0.84 | 95.8 | 98.8 | 0.85 | 81.5 | 94.5 | 0.79 |
| Qwen3-1.7B | Base | 36.0 | 36.8 | 0.94 | 41.5 | 42.3 | 0.94 | 34.0 | 40.0 | 0.88 |
| | RLCracker | 95.8 | 99.8 | 0.85 | 94.8 | 98.8 | 0.84 | 80.3 | 97.3 | 0.77 |
| Qwen2.5-3B -Instruct | Base | 28.0 | 28.8 | 0.93 | 33.8 | 35.3 | 0.93 | 29.3 | 36.0 | 0.88 |
| | RLCracker | 97.3 | 100. | 0.86 | 98.8 | 99.5 | 0.87 | 84.5 | 96.3 | 0.81 |
| Qwen3-4B | Base | 42.3 | 43.3 | 0.94 | 48.3 | 52.8 | 0.94 | 43.8 | 53.3 | 0.88 |
| | RLCracker | 98.0 | 98.8 | 0.88 | 98 | 99.8 | 0.87 | 84.5 | 96.0 | 0.83 |

## E.6 IMPACT OF TRAINING SET LENGTH

In Section 5.3, we conduct an empirical study to examine how RLCracker's effectiveness depends on two key training factors: the token length per sample and the total number of training samples. We train the model using data generated by Llama3.1-8B-Instruct with EWD watermark. We train the model for 10 epochs, using the learning rate of $1 \times 10^{-6}$ for Qwen3-0.6B and $2 \times 10^{-7}$ for Qwen3-4B. We test the model performance on 500-token EWD test dataset. Our results, detailed in Table 11, indicate that Qwen3-0.6B attains an ESR of 82.5% even when trained on as few as 50 samples of 250 tokens each. Increasing the token length under a fixed sample size further boosts ESR to 91.5%. Expanding the number of training samples also improves performance, although the gains diminish once the sample size exceeds 100. These findings suggest that RLCracker can achieve strong watermark removal even with relatively modest training resources.

Table 11: Performance of RLCracker on EWD watermark under different scales of training sets.

| Model | Tokens | 50 Samples | | | 100 Samples | | | 200 Samples | | |
|---|---|---|---|---|---|---|---|---|---|---|
| | | ESR | Rem. | P-SP | ESR | Rem. | P-SP | ESR | Rem. | P-SP |
| Qwen3-0.6B | 250 | 82.5 | 90.2 | 0.87 | 87.8 | 92.8 | 0.86 | 89.3 | 93.5 | 0.86 |
| | 500 | 86.5 | 93.3 | 0.85 | 91.5 | 98.5 | 0.83 | 92.0 | 98.3 | 0.83 |
| | 1500 | 91.5 | 98.3 | 0.83 | 92.5 | 98.5 | 0.83 | 92.8 | 99.3 | 0.82 |
| Qwen3-4B | 250 | 87.8 | 94.5 | 0.86 | 92.0 | 95.3 | 0.87 | 92.8 | 98.3 | 0.87 |
| | 500 | 90.0 | 96.3 | 0.87 | 93.3 | 95.8 | 0.87 | 93.8 | 99.8 | 0.87 |
| | 1500 | 92.3 | 97.0 | 0.87 | 95.3 | 98.0 | 0.86 | 95.3 | 100. | 0.86 |

## E.7 RLCRACKER EFFECTIVENESS ON MIXED-KEY TRAINING DATA

**Training data.** To mimic realistic collection pipelines where watermarked text may be produced under different hash keys (Liu et al., 2024a), we construct mixed-key training sets for watermark EWD, PF, and SWEET. We construct the training datasets from the Reddit WritingPrompts (Verma et al., 2023). During the dataset construction process, we randomly generate 100 distinct 8-digit hash keys. Each hash key is used to generate a unique question-watermarked response pair, which forms the basis of our training data. Subsequently, we use this reconstructed mixed-key training set to train the model, with all other watermarking hyperparameters and text-generation settings following the configuration used in the single-key experiments.

Table 12: ESR, Removal Rate and P-SP of RLCracker trained on Single | Mixed keys data

| Model | Method | EWD | | | SWEET | | | PF | | |
|---|---|---|---|---|---|---|---|---|---|---|
| | | ESR | Rem. | P-SP | ESR | Rem. | P-SP | ESR | Rem. | P-SP |
| Qwen3-0.6B | Single | 91.5 | 98.5 | 0.83 | 92.5 | 99.0 | 0.82 | 76.8 | 94.8 | 0.78 |
| | Mixed | 88.8 | 91.3 | 0.87 | 90.5 | 96.5 | 0.88 | 75.5 | 88.3 | 0.81 |
| Qwen3-1.7B | Single | 91.8 | 97.8 | 0.83 | 91.0 | 97.2 | 0.86 | 79.5 | 95.8 | 0.80 |
| | Mixed | 90.8 | 98.5 | 0.83 | 90.0 | 96.5 | 0.86 | 78.3 | 94.3 | 0.81 |
| Qwen3-4B | Single | 93.3 | 95.8 | 0.87 | 95.8 | 98.8 | 0.85 | 82.3 | 95.0 | 0.82 |
| | Mixed | 92.3 | 95.5 | 0.86 | 94.3 | 97.5 | 0.87 | 80.5 | 89.5 | 0.85 |

**Evaluation.** We evaluate models trained on the mixed-key datasets against single-key test sets, where the test key is the default MarkLLM key shown in Section D.1. For each model (Qwen3-0.6B/1.7B/4B) and watermark (EWD, SWEET, PF), we report ESR, Rem. (removal rate), and P-SP score. As summarized in Table 12, mixed-key training yields only a slight degradation relative to in-domain single-key training, with the largest ESR drop being ∼2% for PF on the 4B model, indicating that RLCracker remains robust under realistic mixed-key conditions.

## E.8 IMPACT OF P-SP SCORE THRESHOLD ON ESR

We analyzed the changes in ESR across four methods (Base, SysP., Think, and RLCracker) using the Qwen3-4B model as the P-SP score threshold varies. As depicted in Figure 7, ESR increases for all methods as the P-SP score threshold decreases. The Base method effectively preserves the semantics of the text but struggles to remove the watermark adequately. In contrast, RLCracker demonstrates the highest ESR, successfully eliminating the watermark while maintaining semantic integrity, with ESR levels stabilizing above 90%.

## E.9 POTENTIAL APPLICATION OF ADAPTIVE ROBUSTNESS RADIUS.

For designers of green-list based watermarks, who have full knowledge of the details of their algorithms and detection hyperparameter settings, our adaptive robustness radius $r_{\text{KL}(s,c,\boldsymbol{\theta})}$ offers an estimate of their watermark's resistance to removal attacks. We empirically explore this potential application by validating the relationship between the radius and removal rate (Rem). Specifically, we calculate the radius for four watermarking schemes, EWD, SWEET, KGW, and KGW_self, on

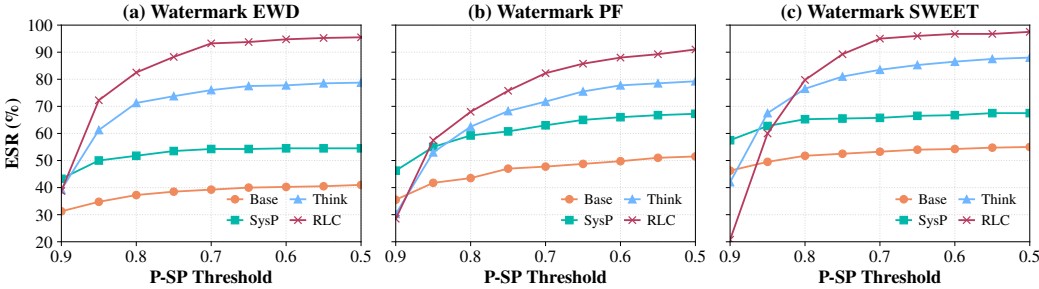

Figure 7: Variation of ESR with P-SP score threshold across three watermarks on 500-token texts

1500-token texts across different models and attack contexts. As shown in Table 13, we observe a strong negative correlation between the radius and removal rate, with Pearson and Spearman coefficients of -0.77 and below -0.9, respectively. These results indicate that our robustness radius provides a useful metric for assessing watermark robustness against removal attacks.

Table 13: Correlation between removal rate (Rem.) and adaptive robustness radius.

| Model | Method | EWD | | SWEET | | KGW | | KGW_self | |
|---|---|---|---|---|---|---|---|---|---|
| | | Rem. | Radius | Rem. | Radius | Rem. | Radius | Rem. | Radius |
| Qwen3-0.6B | Base | 6.20 | 1.35 | 10.2 | 0.97 | 33.8 | 0.21 | 23.5 | 0.32 |
| | SysP. | 13.0 | 0.58 | 25.2 | 0.33 | 55.8 | 0.01 | 47.2 | 0.04 |
| Qwen3-1.7B | Base | 8.50 | 0.95 | 14.5 | 0.64 | 41.8 | 0.11 | 37.5 | 0.14 |
| | SysP. | 15.5 | 0.58 | 23.0 | 0.41 | 55.2 | 0.02 | 49.0 | 0.05 |
| Qwen3-4B | Base | 19.8 | 0.57 | 27.8 | 0.36 | 42.5 | 0.10 | 41.8 | 0.09 |
| | SysP. | 24.8 | 0.41 | 32.5 | 0.25 | 57.5 | 0.00 | 47 | 0.02 |
| Qwen3-8B | Base | 51.5 | 0.04 | 59.2 | 0.00 | 85.0 | 0.00 | 72.0 | 0.00 |
| | SysP. | 48.8 | 0.02 | 64.5 | 0.00 | 90.0 | 0.00 | 75.2 | 0.00 |
| Pearson Correlation | | -0.90 | | -0.90 | | -0.77 | | -0.83 | |
| Spearman Correlation | | -0.95 | | -0.98 | | -0.99 | | -0.90 | |

### E.10 Impact of Distribution Mismatches on Watermark Removal Attack Success.

To investigate the impact of distribution mismatches between the watermark generator and the attacker model on the success rate of watermark removal attacks, we conducted cross-model experiments using the Qwen3 family models: 0.6B, 1.7B, and 4B. For each model, we generated 500-token EWD watermarked texts, which were then utilized to perform watermark removal attacks and to train the RLCracker model.

As shown in the Table 14, when the distributions of the watermark generator and attacker model are similar, RLCracker achieves a high ESR, significantly outperforming the other baseline methods. Notably, even when the watermark generator and attacker model are the same, RLCracker can effectively direct the model to generate paraphrased text that diverges from the watermarked distribution. This highlights that RLCracker's effectiveness does not rely on internal output distribution discrepancies between the watermark generator and attacker model.

### E.11 Effectiveness Comparison across Adaptive Watermark Removal Attack.

To further assess the effectiveness of RLCracker, we conducted additional comparisons with two adaptive attacks featuring stronger threat model settings: the DPO attack (Diaa et al., 2024) and the WS attack (Jovanović et al., 2024). These attacks assume complete knowledge of the watermark detector's details or the full algorithmic details. Specifically, we implemented the DPO attack following (Diaa et al., 2024), training the Qwen2.5-3B-Instruct model with both 100 preference pairs (DPO(100), matching the data budget used by RLCracker) and 7,000 preference pairs (DPO(7000), as outlined in (Diaa et al., 2024)). We then compared the performance of the DPO attack and

Table 14: Limited impact of distribution mismatches on watermark removal attack success

| Gen. Att. | Method | Qwen3-0.6B | | | Qwen3-1.7B | | | Qwen3-4B | | |
|---|---|---|---|---|---|---|---|---|---|---|
| | | ESR | Rem. | P-SP | ESR | Rem. | P-SP | ESR | Rem. | P-SP |
| Qwen3-0.6B | Base | 12.3 | 13.0 | 0.94 | 21.0 | 21.8 | 0.94 | 28.8 | 30.0 | 0.93 |
| | Sysp. | 17.3 | 18.3 | 0.94 | 48 | 49.5 | 0.92 | 36.5 | 38.5 | 0.92 |
| | Think | 17.3 | 23.5 | 0.91 | 65.0 | 71.5 | 0.87 | 63.8 | 69.5 | 0.87 |
| | RLCracker | 87.0 | 95.8 | 0.81 | 87.3 | 96.0 | 0.82 | 88.3 | 98.0 | 0.81 |
| Qwen3-1.7B | Base | 24.3 | 24.8 | 0.94 | 26.0 | 26.8 | 0.96 | 59.3 | 59.3 | 0.95 |
| | Sysp. | 41.0 | 41.8 | 0.94 | 56.5 | 57.5 | 0.94 | 70.5 | 70.8 | 0.94 |
| | Think | 30.5 | 33.0 | 0.93 | 77.3 | 79.0 | 0.90 | 79.8 | 82.3 | 0.90 |
| | RLCracker | 90.8 | 96.0 | 0.86 | 91.3 | 97.5 | 0.86 | 92.0 | 98.3 | 0.88 |
| Qwen3-4B | Base | 30.8 | 31.3 | 0.96 | 39.0 | 39.5 | 0.97 | 49.8 | 50.0 | 0.96 |
| | Sysp. | 46.5 | 46.8 | 0.95 | 70.5 | 71.5 | 0.95 | 69.8 | 70.5 | 0.95 |
| | Think | 37.5 | 40.0 | 0.93 | 88.5 | 86.8 | 0.91 | 79.3 | 81.8 | 0.92 |
| | RLCracker | 91.5 | 98.3 | 0.87 | 92.3 | 97.8 | 0.87 | 93.3 | 97.5 | 0.87 |

RLCracker on both 500-token and 1500-token test sets. For the WS attack, we followed the approach in (Jovanović et al., 2024) and tested its performance using the Qwen2.5-3B-Instruct model on the 500-token KGW and KGW_self test sets.

As shown in Table 15, RLCracker significantly outperforms both DPO variants across all watermarking schemes, with the performance gap being particularly pronounced on longer (1500-token) sequences. Similarly, as presented in Table 16, RLCracker also outperforms the WS attack, further demonstrating that our improvements are due to the distributional framework rather than adjustments to the model tuning or scale.

Table 15: Performance of RLCracker and DPO attacker on Qwen2.5-3B-Instruct.

| Length | Method | Unigram | | | KGW_self | | | PF | | |
|---|---|---|---|---|---|---|---|---|---|---|
| | | ESR | Rem. | P-SP | ESR | Rem. | P-SP | ESR | Rem. | P-SP |
| 500 tokens | DPO(100) | 45.2 | 73.0 | 0.81 | 78.8 | 88.8 | 0.89 | 69.3 | 80.3 | 0.85 |
| | DPO(7000) | 69.3 | 84.5 | 0.71 | 81.0 | 92.5 | 0.82 | 71.8 | 89.5 | 0.79 |
| | RLCracker | 78.5 | 90.2 | 0.81 | 89.8 | 93.5 | 0.86 | 81.8 | 95.2 | 0.84 |
| 1500 tokens | DPO(100) | 7.25 | 27.3 | 0.69 | 65.8 | 70.5 | 0.87 | 41.3 | 60.8 | 0.83 |
| | DPO(7000) | 20.8 | 45.3 | 0.67 | 74.0 | 85.3 | 0.84 | 47.0 | 71.5 | 0.75 |
| | RLCracker | 98.5 | 100. | 0.92 | 78.0 | 89.0 | 0.83 | 77.8 | 90.3 | 0.81 |

Table 16: Performance of WS, DPO attacker, and RLCracker on 500 token test set.

| Method | KGW | | | KGW_self | | |
|---|---|---|---|---|---|---|
| | ESR | Rem. | P-SP | ESR | Rem. | P-SP |
| WS | 91.0 | 98.5 | 0.87 | 89.0 | 92.5 | 0.86 |
| DPO(7000) | 67.3 | 89.0 | 0.73 | 81.0 | 92.5 | 0.82 |
| RLCracker | 97.5 | 99.3 | 0.86 | 89.8 | 93.5 | 0.86 |

We further conducted a direct comparison of RLCracker with the LoRA checkpoint provided by Diaa et al. (2024), which was trained using 7,000 preference pairs. As shown in the Table 17 and 18, despite being trained on only 7,000 samples and employing smaller base models (e.g., Qwen3-0.6B and Qwen2.5-3B), RLCracker consistently matches or outperforms the DPO model across both 500-token and 1500-token settings for the KGW and KGW_self watermarks. In the case of the Unigram watermark, where the DPO paraphrasers exhibit considerable underperformance, RLCracker achieves significantly higher ESR and stronger P-SP fidelity, underscoring its robust generalization capability. These results clearly demonstrate that RLCracker not only adapts effectively across various watermark types but also provides comparable or superior performance to the DPO paraphraser, all while operating under significantly lighter training requirements.

Table 17: Performance of RLCracker and DPO attacker across models.

| Method | KGW (500) | | | KGW (1500) | | | KGW_self (500) | | | KGW_self (1500) | | |
|---|---|---|---|---|---|---|---|---|---|---|---|---|
| | ESR | Rem. | P-SP | ESR | Rem. | P-SP | ESR | Rem. | P-SP | ESR | Rem. | P-SP |
| DPO(Llama3.1-8B) | 67.5 | 95.3 | 0.71 | 47.5 | 92.5 | 0.76 | 84.5 | 91.3 | 0.86 | 74.0 | 88.3 | 0.84 |
| RLC.(Qwen3-0.6B) | 80.8 | 84.0 | 0.89 | 48.5 | 51.5 | 0.88 | 85.2 | 90.5 | 0.87 | 66.8 | 80.0 | 0.84 |
| RLC.(Qwen2.5-3B) | 97.5 | 99.3 | 0.86 | 58.0 | 67.3 | 0.90 | 89.8 | 93.5 | 0.86 | 78.0 | 89.0 | 0.83 |
| RLC.(Qwen3-8B) | 91.5 | 94.5 | 0.90 | 64.5 | 87.5 | 0.78 | 90.8 | 94.8 | 0.90 | 84.3 | 97.3 | 0.83 |

Table 18: Performance of RLCracker and DPO attacker across models.

| Method | Unigram (500) | | | Unigram (1500) | | |
|---|---|---|---|---|---|---|
| | ESR | Rem. | P-SP | ESR | Rem. | P-SP |
| DPO(Llama3.2-3B) | 58.5 | 100 | 0.70 | 15.5 | 100 | 0.66 |
| DPO(Qwen2.5-3B) | 61.3 | 85.0 | 0.71 | 16.8 | 96.0 | 0.66 |
| RLCracker(Qwen3-0.6B) | 67.0 | 71.5 | 0.86 | 90.2 | 90.5 | 0.93 |
| RLCracker(Qwen2.5-3B) | 78.5 | 90.3 | 0.81 | 98.5 | 100. | 0.92 |
| RLCracker(Qwen3-8B) | 80.5 | 88.3 | 0.80 | 81.8 | 88.5 | 0.84 |

## E.12 DIFFERENCE BETWEEN APPROXIMATED $P_h$ AND THE TRUE HUMAN-WRITTEN DISTRIBUTION.

In RLCracker, we use a lightweight reference model (e.g., Qwen3-0.6B) to approximate the distribution of human-written text. Since it's not possible to directly obtain the true human-written distribution, we evaluate the effectiveness of this approximation by comparing the detection score distributions of model-generated text and natural text under various watermark detection algorithms. This comparison serves as a proxy to measure how well models of different sizes approximate the human-written distribution.

Specifically, we randomly selected 1,000 samples from the C4 dataset, used the model to continue writing, and then calculated the detection scores for both model-generated and natural text using the SWEET, KGW, and EWD watermark detectors. As shown in Figures 8, 9, and 10, we observe that as the model size increases, the detection score distribution of model-generated text shifts closer to that of natural text. This suggests that the model's ability to approximate the human-written distribution ($P_h$) improves as its size grows. We also notice that these shifts are generally subtle, suggesting that even smaller models can effectively approximate the human-written distribution $P_h$.

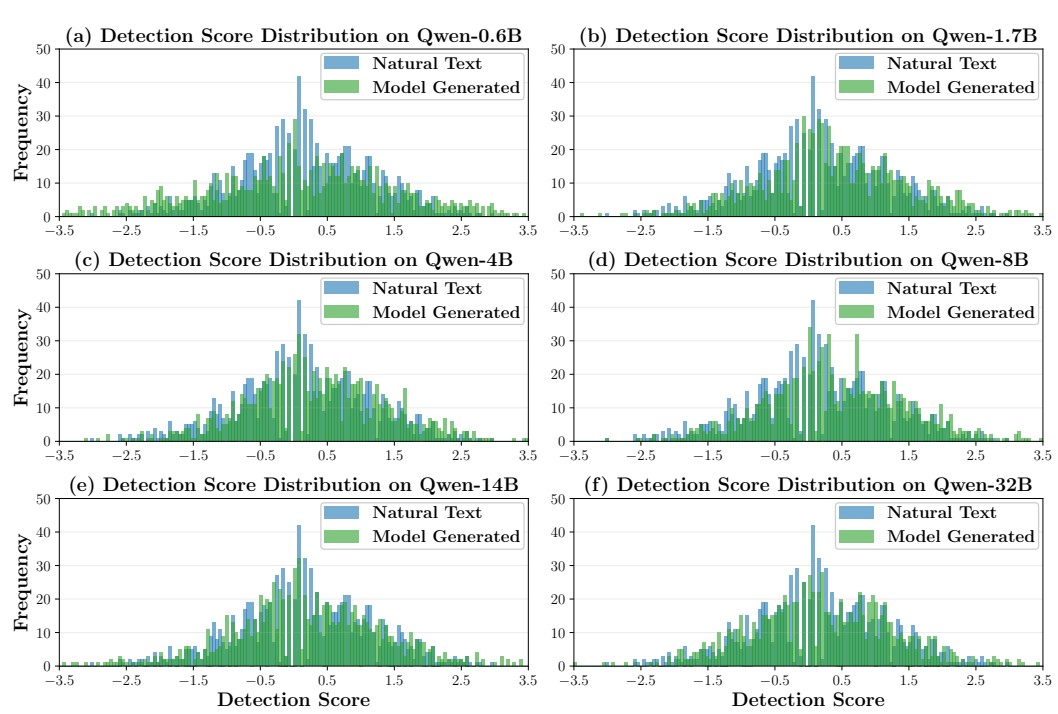

Figure 8: SWEET Detection Score Distribution across Different Model Sizes

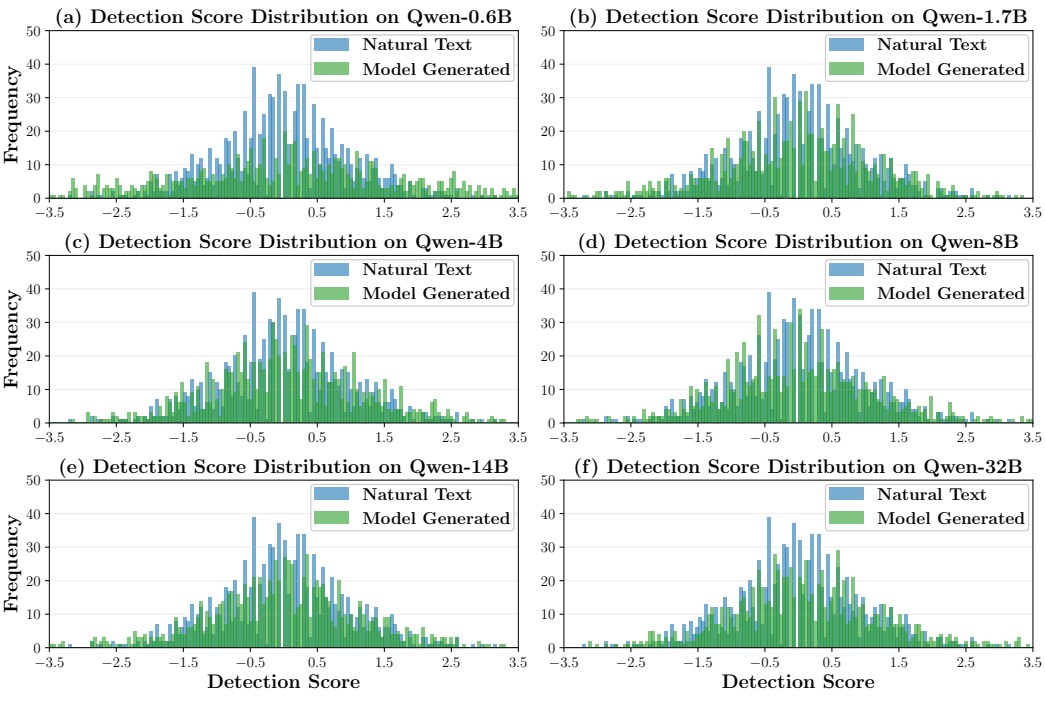

Figure 9: KGW Detection Score Distribution across Different Model Sizes

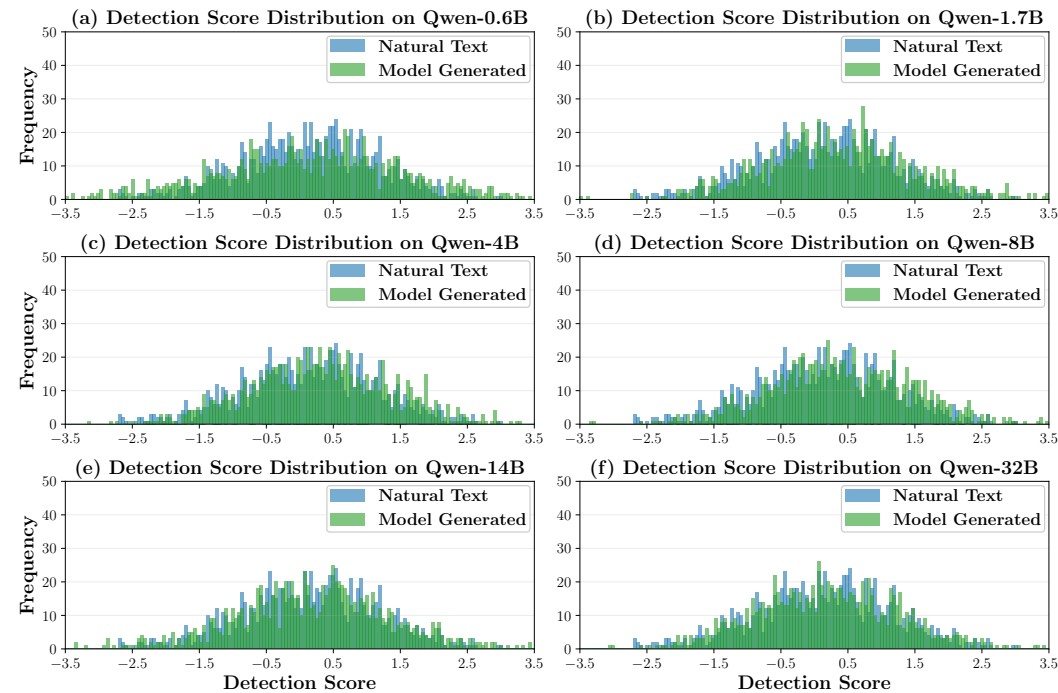

Figure 10: EWD Detection Score Distribution across Different Model Sizes

Table 19: Rephrased text quality across models and watermarking schemes (500 tokens).

| Model | Method | EWD | | | SWEET | | | XSIR | | | Unigram | | |
|---|---|---|---|---|---|---|---|---|---|---|---|---|---|
| | | ESR | PPL. | GPTS. | ESR | PPL. | GPTS. | ESR | PPL. | GPTS. | ESR | PPL. | GPTS. |
| Target Watermarked Text | | — | 3.58 | — | — | 3.53 | — | — | 3.11 | — | — | 16.5 | — |
| Qwen3-0.6B | Base | 13.5 | 3.93 | 8.85 | 21.2 | 3.78 | 8.87 | 29.0 | 3.13 | 8.72 | 35.8 | 16.56 | 8.63 |
| | SysP. | 36.5 | 3.94 | 8.84 | 44.8 | 3.78 | 8.91 | 52.5 | 3.14 | 8.69 | 42.5 | 16.63 | 8.66 |
| | Think | 28.5 | 3.9 | 8.71 | 35.9 | 3.76 | 8.81 | 37.2 | 3.02 | 8.74 | 34.2 | 16.46 | 8.57 |
| | Think+SysP. | 48.2 | 3.91 | 8.82 | 58.8 | 3.77 | 8.84 | 50.7 | 3.06 | 8.73 | 37.5 | 16.49 | 8.59 |
| | SIRA | 0.00 | 5.87 | 4.40 | 0.00 | 5.71 | 4.51 | 0.0 | 4.62 | 8.76 | 0.0 | 4.39 | 4.26 |
| | RLCracker | 91.5 | 3.78 | 8.72 | 92.5 | 3.64 | 8.89 | 86.5 | 2.83 | 4.46 | 67.0 | 16.17 | 8.48 |
| Qwen3-1.7B | Base | 34.2 | 3.97 | 8.87 | 42.8 | 3.84 | 8.89 | 43.0 | 3.29 | 8.78 | 34.0 | 16.81 | 8.69 |
| | SysP. | 59.8 | 3.99 | 8.90 | 70.8 | 3.92 | 8.91 | 70.8 | 3.3 | 8.77 | 40.5 | 16.91 | 8.69 |
| | Think | 80.8 | 3.96 | 8.83 | 88.0 | 3.8 | 8.84 | 75.5 | 3.23 | 8.81 | 31.0 | 16.75 | 8.68 |
| | Think+SysP. | 84.0 | 3.96 | 8.85 | 86.5 | 3.82 | 8.85 | 79.8 | 3.26 | 8.78 | 34.8 | 16.8 | 8.69 |
| | SIRA | 0.25 | 5.95 | 4.53 | 0.00 | 5.79 | 4.54 | 0.2 | 5.17 | 4.51 | 0.0 | 4.41 | 4.37 |
| | RLCracker | 91.8 | 3.83 | 8.79 | 91.0 | 3.67 | 8.71 | 86.0 | 3.06 | 8.81 | 73.0 | 16.56 | 8.6 |
| Qwen3-4B | Base | 39.2 | 4.03 | 8.94 | 53.2 | 4.03 | 8.96 | 50.2 | 3.38 | 8.83 | 39.2 | 17.3 | 8.73 |
| | SysP. | 54.2 | 4.04 | 8.88 | 65.8 | 4.06 | 8.95 | 66.8 | 3.39 | 8.82 | 43.8 | 17.38 | 8.73 |
| | Think | 76.0 | 4.0 | 8.84 | 83.5 | 3.99 | 8.91 | 75.2 | 3.33 | 8.84 | 38.2 | 17.0 | 8.72 |
| | Think+SysP. | 78.8 | 4.03 | 8.93 | 88.0 | 4.02 | 8.92 | 79.2 | 3.34 | 8.84 | 38.8 | 17.29 | 8.72 |
| | SIRA | 0.00 | 5.99 | 4.51 | 0.0 | 5.93 | 4.55 | 0.2 | 5.3 | 8.84 | 0.8 | 4.44 | 4.4 |
| | RLCracker | 93.3 | 4.47 | 8.81 | 95.8 | 4.35 | 8.73 | 85.5 | 3.2 | 4.54 | 73.5 | 16.86 | 8.62 |
| Qwen3-8B | Base | 72.0 | 4.12 | 8.97 | 78.4 | 4.26 | 9.12 | 77.0 | 3.59 | 8.85 | 40.5 | 18.05 | 8.75 |
| | SysP. | 72.8 | 4.26 | 9.01 | 81.5 | 4.27 | 9.13 | 82.0 | 3.63 | 8.84 | 43.5 | 18.36 | 8.75 |
| | Think | 89.5 | 4.1 | 8.87 | 92.0 | 4.14 | 9.03 | 81.5 | 3.57 | 8.86 | 40.8 | 17.53 | 8.74 |
| | Think+SysP. | 90.2 | 4.12 | 8.91 | 92.0 | 4.22 | 9.06 | 86.8 | 3.58 | 8.86 | 41.0 | 17.74 | 8.74 |
| | SIRA | 0.00 | 6.05 | 4.46 | 0.0 | 6.07 | 4.51 | 0.2 | 5.41 | 8.87 | 0.5 | 4.49 | 4.44 |
| | RLCracker | 94.8 | 3.97 | 8.88 | 96.3 | 3.96 | 8.91 | 90.2 | 3.36 | 4.57 | 80.5 | 17.37 | 8.66 |
| Qwen2.5-3B -Instruct | Base | 31.2 | 4.05 | 8.79 | 46.8 | 3.91 | 8.83 | 68.8 | 3.33 | 8.79 | 37.2 | 16.97 | 8.69 |
| | SysP. | 75.5 | 4.07 | 8.82 | 83.0 | 3.98 | 8.86 | 78.5 | 3.38 | 8.81 | 43.5 | 17.12 | 8.75 |
| | SIRA | 0.25 | 6.01 | 4.35 | 0.2 | 5.86 | 4.33 | 0.2 | 5.24 | 4.34 | 0.8 | 4.33 | 4.34 |
| | RLCracker | 93.3 | 3.98 | 8.73 | 95.3 | 3.75 | 8.69 | 89.8 | 3.16 | 8.69 | 78.5 | 16.81 | 8.6 |
| GPT-4o | — | 71.0 | 4.01 | 9.13 | 81.5 | 3.81 | 9.21 | 77.2 | 3.41 | 9.15 | 49.8 | 18.32 | 9.12 |
| DIPPER | — | 30.3 | 5.93 | 8.76 | 52.8 | 4.06 | 8.77 | 0.0 | 5.21 | 8.79 | 33.3 | 18.92 | 8.79 |

Table 20: Evasion Success Rate (ESR, %) across models and watermarking schemes (500 tokens).

| Model | Method | KGW_self | | | KGW | | | SIR | | | PF | | |
|---|---|---|---|---|---|---|---|---|---|---|---|---|---|
| | | ESR | Rem. | P-SP | ESR | Rem. | P-SP | ESR | Rem. | P-SP | ESR | Rem. | P-SP |
| Qwen3-0.6B | Base | 35.5 | 39.0 | 0.92 | 60.0 | 90.0 | 0.79 | 92.2 | 98.8 | 0.90 | 14.5 | 18.8 | 0.89 |
| | SysP. | 63.2 | 66.8 | 0.90 | 52.2 | 93.5 | 0.74 | 92.5 | 99.2 | 0.89 | 37.2 | 45.0 | 0.87 |
| | Think | 66.3 | 69.1 | 0.89 | 45.2 | 93.0 | 0.69 | 89.0 | 98.2 | 0.87 | 27.3 | 36.8 | 0.85 |
| | Think+SysP. | 69.2 | 74.8 | 0.87 | 34.5 | 96.5 | 0.64 | 86.0 | 97.2 | 0.85 | 53.8 | 62.0 | 0.85 |
| | SIRA | 0.2 | 57.0 | 0.29 | 24.8 | 86.8 | 0.50 | 0.2 | 89.2 | 0.21 | 0.0 | 39.8 | 0.20 |
| | RLCracker | 85.2 | 90.5 | 0.87 | 80.8 | 84.0 | 0.89 | 90.5 | 94.2 | 0.91 | 76.8 | 94.8 | 0.78 |
| Qwen3-1.7B | Base | 56.8 | 59.2 | 0.92 | 62.3 | 90.0 | 0.81 | 94.8 | 99.8 | 0.92 | 36.5 | 41.5 | 0.90 |
| | SysP. | 76.2 | 80.5 | 0.90 | 51.2 | 94.5 | 0.73 | 93.8 | 100.0 | 0.89 | 63.2 | 70.0 | 0.88 |
| | Think | 86.0 | 96.5 | 0.83 | 28.2 | 98.8 | 0.63 | 77.8 | 98.2 | 0.78 | 71.2 | 86.2 | 0.80 |
| | Think+SysP. | 86.0 | 96.8 | 0.83 | 28.2 | 99.5 | 0.63 | 77.0 | 97.8 | 0.77 | 77.5 | 90.5 | 0.80 |
| | SIRA | 0.2 | 96.2 | 0.32 | 5.8 | 97.5 | 0.39 | 0.2 | 99.0 | 0.21 | 0.0 | 86.8 | 0.25 |
| | RLCracker | 87.5 | 89.2 | 0.88 | 76.8 | 81.0 | 0.91 | 94.5 | 100.0 | 0.91 | 79.5 | 95.8 | 0.80 |
| Qwen3-4B | Base | 70.0 | 73.0 | 0.92 | 72.0 | 89.8 | 0.87 | 52.2 | 53.5 | 0.93 | 47.8 | 52.8 | 0.90 |
| | SysP. | 79.2 | 82.8 | 0.91 | 68.8 | 94.8 | 0.82 | 67.5 | 70.2 | 0.91 | 63.0 | 68.5 | 0.90 |
| | Think | 85.2 | 92.8 | 0.86 | 51.7 | 97.8 | 0.73 | 79.0 | 90.5 | 0.82 | 71.8 | 83.5 | 0.83 |
| | Think+SysP. | 88.2 | 96.5 | 0.85 | 47.5 | 98.2 | 0.72 | 79.5 | 93.2 | 0.80 | 81.2 | 89.2 | 0.84 |
| | SIRA | 0.2 | 92.8 | 0.32 | 12.2 | 98.8 | 0.44 | 0.2 | 84.0 | 0.21 | 0.0 | 88.8 | 0.24 |
| | RLCracker | 88.8 | 98.5 | 0.84 | 89.0 | 92.2 | 0.90 | 96.8 | 100.0 | 0.92 | 82.2 | 95.0 | 0.82 |
| Qwen3-8B | Base | 81.3 | 89.8 | 0.89 | 74.5 | 91.5 | 0.88 | 74.0 | 76.8 | 0.90 | 70.0 | 75.5 | 0.91 |
| | SysP. | 85.5 | 91.5 | 0.90 | 72.0 | 92.8 | 0.85 | 79.8 | 83.2 | 0.88 | 73.0 | 79.5 | 0.90 |
| | Think | 83.8 | 94.0 | 0.83 | 53.8 | 95.8 | 0.76 | 76.2 | 92.2 | 0.78 | 75.7 | 82.5 | 0.84 |
| | Think+SysP. | 86.2 | 94.5 | 0.82 | 57.5 | 98.0 | 0.75 | 71.0 | 95.2 | 0.76 | 74.0 | 87.8 | 0.81 |
| | SIRA | 0.2 | 93.0 | 0.32 | 22.2 | 97.0 | 0.52 | 0.2 | 85.2 | 0.22 | 0.0 | 82.8 | 0.26 |
| | RLCracker | 90.8 | 94.8 | 0.90 | 91.5 | 94.5 | 0.90 | 96.3 | 99.8 | 0.88 | 84.3 | 91.3 | 0.87 |
| Qwen2.5-3B -Instruct | Base | 58.8 | 61.3 | 0.92 | 46.0 | 69.5 | 0.86 | 64.0 | 72.8 | 0.88 | 37.0 | 45.0 | 0.90 |
| | SysP. | 84.5 | 88.0 | 0.90 | 31.8 | 65.8 | 0.79 | 75.5 | 87.8 | 0.86 | 74.0 | 80.3 | 0.89 |
| | SIRA | 0.2 | 92.2 | 0.32 | 2.2 | 99.0 | 0.42 | 0.2 | 90.5 | 0.23 | 0.0 | 78.2 | 0.26 |
| | RLCracker | 89.8 | 93.5 | 0.86 | 97.5 | 99.3 | 0.86 | 74.0 | 90.8 | 0.82 | 81.8 | 95.2 | 0.84 |
| GPT-4o | — | 86.8 | 91.2 | 0.90 | 53.8 | 96.0 | 0.73 | 94.8 | 98.8 | 0.88 | 79.0 | 87.8 | 0.88 |
| DIPPER | — | 72.0 | 78.3 | 0.80 | 29.0 | 93.8 | 0.61 | 61.8 | 76.5 | 0.78 | 43.5 | 49.8 | 0.81 |

Table 21: Rephrased text quality across models and watermarking schemes (500 tokens).

| Model | Method | KGW_self | | | KGW | | | SIR | | | PF | | |
|---|---|---|---|---|---|---|---|---|---|---|---|---|---|
| | | ESR | PPL. | GPTS. | ESR | PPL. | GPTS. | ESR | PPL. | GPTS. | ESR | PPL. | GPTS. |
| Target Watermarked Text | | — | 4.36 | — | — | 17.5 | — | — | 3.24 | — | — | 10.6 | — |
| Qwen3-0.6B | Base | 35.5 | 4.43 | 8.71 | 60.0 | 17.25 | 8.68 | 92.2 | 3.3 | 8.79 | 14.5 | 9.96 | 8.69 |
| | SysP. | 63.2 | 4.43 | 8.73 | 52.2 | 17.32 | 8.7 | 92.5 | 3.35 | 8.79 | 37.2 | 9.97 | 8.7 |
| | Think | 66.3 | 4.39 | 8.7 | 45.2 | 17.15 | 8.66 | 89.0 | 3.26 | 8.78 | 27.3 | 9.67 | 8.68 |
| | Think+SysP. | 69.2 | 4.42 | 8.71 | 34.5 | 17.22 | 8.66 | 86.0 | 3.3 | 8.78 | 53.8 | 9.85 | 8.69 |
| | SIRA | 0.2 | 6.05 | 4.39 | 24.8 | 19.03 | 4.35 | 0.2 | 4.87 | 4.44 | 0.0 | 12.3 | 4.36 |
| | RLCracker | 85.2 | 4.17 | 8.61 | 80.8 | 17.01 | 8.58 | 90.5 | 3.11 | 8.68 | 76.8 | 9.41 | 8.59 |
| Qwen3-1.7B | Base | 56.8 | 4.55 | 8.75 | 62.3 | 17.47 | 8.72 | 94.8 | 3.43 | 8.81 | 36.5 | 10.22 | 8.73 |
| | SysP. | 76.2 | 4.57 | 8.76 | 51.2 | 17.54 | 8.73 | 93.8 | 3.52 | 8.82 | 63.2 | 10.23 | 8.73 |
| | Think | 86.0 | 4.49 | 8.75 | 28.2 | 17.41 | 8.71 | 77.8 | 3.38 | 8.81 | 71.2 | 10.11 | 8.72 |
| | Think+SysP. | 86.0 | 4.53 | 8.75 | 28.2 | 17.45 | 8.71 | 77.0 | 3.41 | 8.81 | 77.5 | 10.11 | 8.72 |
| | SIRA | 0.2 | 6.44 | 4.44 | 5.8 | 19.4 | 4.4 | 0.2 | 5.07 | 4.49 | 0.0 | 12.84 | 4.4 |
| | RLCracker | 87.5 | 4.36 | 8.66 | 76.8 | 17.28 | 8.63 | 94.5 | 3.25 | 8.72 | 79.5 | 9.94 | 8.63 |
| Qwen3-4B | Base | 70.0 | 4.73 | 8.79 | 72.0 | 17.73 | 8.74 | 52.2 | 3.68 | 8.85 | 47.8 | 10.66 | 8.73 |
| | SysP. | 79.2 | 4.8 | 8.8 | 68.8 | 17.74 | 8.76 | 67.5 | 3.69 | 8.85 | 63.0 | 10.67 | 8.79 |
| | Think | 85.2 | 4.7 | 8.78 | 51.7 | 17.66 | 8.73 | 79.0 | 3.62 | 8.84 | 71.8 | 10.46 | 8.63 |
| | Think+SysP. | 88.2 | 4.7 | 8.78 | 47.5 | 17.7 | 8.74 | 79.5 | 3.65 | 8.84 | 81.2 | 10.65 | 8.70 |
| | SIRA | 0.2 | 6.58 | 4.47 | 12.2 | 19.55 | 4.43 | 0.2 | 5.29 | 4.53 | 0.0 | 13.05 | 4.43 |
| | RLCracker | 88.8 | 4.56 | 8.69 | 89.0 | 17.45 | 8.65 | 96.8 | 3.48 | 8.75 | 82.2 | 5.03 | 8.61 |
| Qwen3-8B | Base | 81.3 | 4.91 | 8.83 | 74.5 | 17.85 | 8.78 | 74.0 | 3.85 | 8.9 | 70.0 | 10.98 | 8.81 |
| | SysP. | 85.5 | 5.06 | 8.86 | 72.0 | 17.91 | 8.83 | 79.8 | 3.98 | 8.9 | 73.0 | 11.16 | 8.82 |
| | Think | 83.8 | 4.87 | 8.81 | 53.8 | 17.81 | 8.78 | 76.2 | 3.76 | 8.87 | 75.7 | 10.81 | 8.78 |
| | Think+SysP. | 86.2 | 4.9 | 8.82 | 57.5 | 17.82 | 8.78 | 71.0 | 3.77 | 8.89 | 74.0 | 10.83 | 8.81 |
| | SIRA | 0.2 | 6.81 | 4.51 | 22.2 | 19.75 | 4.47 | 0.2 | 5.41 | 4.56 | 0.0 | 13.48 | 4.45 |
| | RLCracker | 90.8 | 4.72 | 8.73 | 91.5 | 17.69 | 8.69 | 96.3 | 3.62 | 8.78 | 84.3 | 10.67 | 8.68 |
| Qwen2.5-3B -Instruct | Base | 58.8 | 4.53 | 8.79 | 46.0 | 17.51 | 8.77 | 64.0 | 3.52 | 8.87 | 37.0 | 10.52 | 8.76 |
| | SysP. | 84.5 | 4.55 | 8.82 | 31.8 | 17.52 | 8.82 | 75.5 | 3.55 | 8.91 | 74.0 | 10.54 | 8.83 |
| | SIRA | 0.2 | 6.52 | 4.46 | 2.2 | 20.71 | 4.38 | 0.2 | 5.21 | 4.54 | 0.0 | 13.18 | 4.44 |
| | RLCracker | 89.8 | 4.41 | 8.71 | 97.5 | 17.36 | 8.64 | 74.0 | 3.37 | 8.78 | 81.8 | 10.34 | 8.67 |
| GPT-4o | — | 86.8 | 4.36 | 9.15 | 53.8 | 18.23 | 9.14 | 94.8 | 3.28 | 9.16 | 79.0 | 11.23 | 9.06 |
| DIPPER | — | 72.0 | 5.09 | 8.69 | 29.0 | 19.93 | 8.69 | 61.8 | 5.07 | 8.78 | 43.5 | 13.85 | 8.79 |

Table 22: Evasion Success Rate (ESR, %) across models and watermarking schemes (500 tokens).

| Model | Method | SynthID-Text | | | UPV | | |
|---|---|---|---|---|---|---|---|
| | | ESR | Rem. | P-SP | ESR | Rem. | P-SP |
| Qwen3-0.6B | Base | 44.5 | 47.0 | 0.93 | 53.5 | 56.8 | 0.91 |
| | SysP. | 73.5 | 76.0 | 0.92 | 76.0 | 79.5 | 0.90 |
| | Think | 60.8 | 64.5 | 0.91 | 60.2 | 67.5 | 0.88 |
| | Think+SysP. | 81.8 | 85.2 | 0.89 | 79.5 | 84.8 | 0.87 |
| | SIRA | 0.0 | 61.0 | 0.19 | 0.2 | 62.0 | 0.30 |
| | RLCracker | 96.5 | 99.5 | 0.86 | 91.2 | 97.5 | 0.84 |
| Qwen3-1.7B | Base | 66.8 | 68.5 | 0.94 | 63.0 | 65.8 | 0.91 |
| | SysP. | 90.5 | 92.8 | 0.91 | 87.2 | 90.8 | 0.89 |
| | Think | 91.5 | 97.0 | 0.85 | 87.5 | 98.5 | 0.81 |
| | Think+SysP. | 92.8 | 99.0 | 0.84 | 90.0 | 99.0 | 0.81 |
| | SIRA | 0.2 | 99.8 | 0.23 | 0.0 | 98.5 | 0.32 |
| | RLCracker | 95.2 | 98.8 | 0.86 | 90.5 | 97.0 | 0.85 |
| Qwen3-4B | Base | 82.2 | 84.5 | 0.93 | 66.2 | 68.8 | 0.92 |
| | SysP. | 93.0 | 94.5 | 0.93 | 83.5 | 85.2 | 0.91 |
| | Think | 92.8 | 97.8 | 0.87 | 90.2 | 97.8 | 0.84 |
| | Think+SysP. | 95.2 | 99.8 | 0.86 | 91.2 | 97.8 | 0.85 |
| | SIRA | 0.2 | 98.8 | 0.21 | 0.2 | 95.8 | 0.32 |
| | RLCracker | 96.5 | 99.0 | 0.89 | 94.0 | 99.0 | 0.85 |
| Qwen3-8B | Base | 93.0 | 95.0 | 0.91 | 84.5 | 87.8 | 0.91 |
| | SysP. | 97.0 | 99.0 | 0.91 | 90.0 | 93.0 | 0.90 |
| | Think | 90.8 | 98.2 | 0.84 | 88.2 | 98.5 | 0.82 |
| | Think+SysP. | 93.5 | 99.8 | 0.84 | 91.5 | 99.0 | 0.82 |
| | SIRA | 0.2 | 97.2 | 0.22 | 0.0 | 96.2 | 0.34 |
| | RLCracker | 98.0 | 99.5 | 0.91 | 95.5 | 98.3 | 0.89 |
| Qwen2.5-3B -Instruct | Base | 78.0 | 82.0 | 0.91 | 81.5 | 85.5 | 0.90 |
| | SysP. | 94.8 | 99.3 | 0.89 | 92.0 | 97.0 | 0.87 |
| | SIRA | 0.2 | 99.0 | 0.23 | 0.2 | 97.5 | 0.33 |
| | RLCracker | 93.0 | 98.5 | 0.88 | 90.5 | 98.2 | 0.86 |
| GPT-4o | — | 96.8 | 99.0 | 0.91 | 92.0 | 96.8 | 0.89 |
| DIPPER | — | 90.5 | 96.3 | 0.82 | 81.8 | 89.8 | 0.81 |

Table 23: Rephrased text quality across models and watermarking schemes (500 tokens).

| Model | Method | SynthID-Text | | | UPV | | |
|---|---|---|---|---|---|---|---|
| | | ESR | PPL. | GPTS. | ESR | PPL. | GPTS. |
| Target Watermarked Text | | — | 2.75 | — | — | 16.1 | — |
| Qwen3-0.6B | Base | 44.5 | 2.93 | 8.73 | 53.5 | 16.09 | 8.66 |
| | SysP. | 73.5 | 2.96 | 8.73 | 76 | 16.2 | 8.66 |
| | Think | 60.8 | 2.76 | 8.72 | 60.2 | 16.06 | 8.63 |
| | Think+SysP. | 81.8 | 2.88 | 8.73 | 79.5 | 16.06 | 8.64 |
| | SIRA | 0.0 | 4.01 | 4.37 | 0.2 | 18.33 | 4.32 |
| | RLCracker | 96.5 | 2.6 | 8.62 | 91.2 | 15.78 | 8.54 |
| Qwen3-1.7B | Base | 66.8 | 3.05 | 8.76 | 63 | 16.38 | 8.7 |
| | SysP. | 90.5 | 3.06 | 8.76 | 87.2 | 16.44 | 8.7 |
| | Think | 91.5 | 3.0 | 8.75 | 87.5 | 16.34 | 8.69 |
| | Think+SysP. | 92.8 | 3.03 | 8.75 | 90.0 | 16.36 | 8.7 |
| | SIRA | 0.2 | 4.38 | 4.44 | 0.0 | 19.08 | 4.37 |
| | RLCracker | 95.2 | 2.86 | 8.66 | 90.5 | 16.21 | 8.61 |
| Qwen3-4B | Base | 82.2 | 3.15 | 8.81 | 66.2 | 16.82 | 8.74 |
| | SysP. | 93.0 | 3.21 | 8.82 | 83.5 | 16.89 | 8.74 |
| | Think | 92.8 | 3.1 | 8.77 | 90.2 | 16.64 | 8.72 |
| | Think+SysP. | 95.2 | 3.11 | 8.8 | 91.2 | 16.66 | 8.73 |
| | SIRA | 0.2 | 4.46 | 4.46 | 0.2 | 19.27 | 4.42 |
| | RLCracker | 96.5 | 2.95 | 8.69 | 94.0 | 16.44 | 8.64 |
| Qwen3-8B | Base | 93.0 | 3.4 | 8.85 | 84.5 | 17.23 | 8.82 |
| | SysP. | 97.0 | 3.83 | 8.9 | 90.0 | 17.25 | 8.84 |
| | Think | 90.8 | 3.35 | 8.84 | 88.2 | 17.03 | 8.78 |
| | Think+SysP. | 93.5 | 3.35 | 8.85 | 91.5 | 17.17 | 8.8 |
| | SIRA | 0.2 | 4.7 | 4.53 | 0.0 | 19.69 | 4.45 |
| | RLCracker | 98.0 | 3.21 | 8.75 | 95.5 | 16.79 | 8.68 |
| Qwen2.5-3B -Instruct | Base | 78.0 | 3.08 | 8.75 | 81.5 | 16.55 | 8.74 |
| | SysP. | 94.8 | 3.08 | 8.81 | 92.0 | 16.56 | 8.76 |
| | SIRA | 0.2 | 4.34 | 4.43 | 0.2 | 19.35 | 4.32 |
| | RLCracker | 93.0 | 2.92 | 8.66 | 90.5 | 16.39 | 8.62 |
| GPT-4o | — | 96.8 | 2.78 | 9.15 | 92.0 | 16.12 | 9.11 |
| DIPPER | — | 90.5 | 2.89 | 8.78 | 81.8 | 16.93 | 8.77 |

Table 24: Evasion Success Rate (ESR, %) across models and watermarking schemes (1500 tokens).

| Model | Method | EWD | | | SWEET | | | XSIR | | | Unigram | | |
|---|---|---|---|---|---|---|---|---|---|---|---|---|---|
| | | ESR | Rem. | P-SP | ESR | Rem. | P-SP | ESR | Rem. | P-SP | ESR | Rem. | P-SP |
| Qwen3-0.6B | Base | 3.5 | 6.2 | 0.81 | 4.2 | 10.2 | 0.79 | 26.0 | 45.5 | 0.71 | 5.0 | 53.0 | 0.71 |
| | SysP. | 5.8 | 13.0 | 0.80 | 12.5 | 25.2 | 0.78 | 39.2 | 61.0 | 0.69 | 6.2 | 76.5 | 0.61 |
| | Think | 10.8 | 16.5 | 0.81 | 10.2 | 20.0 | 0.80 | 32.0 | 53.8 | 0.73 | 6.0 | 75.5 | 0.59 |
| | Think+SysP. | 14.8 | 24.2 | 0.80 | 16.2 | 31.0 | 0.78 | 38.2 | 65.2 | 0.72 | 5.2 | 89.2 | 0.52 |
| | SIRA | 0.0 | 38.8 | 0.27 | 0.0 | 48.5 | 0.29 | 0.0 | 0.0 | 0.25 | 0.2 | 62.0 | 0.52 |
| | RLCracker | 63.2 | 93.8 | 0.72 | 63.2 | 96.8 | 0.71 | 60.0 | 97.0 | 0.66 | 90.2 | 90.5 | 0.93 |
| Qwen3-1.7B | Base | 4.2 | 8.5 | 0.83 | 6.8 | 14.5 | 0.82 | 30.2 | 49.5 | 0.75 | 5.0 | 60.8 | 0.70 |
| | SysP. | 7.5 | 15.5 | 0.82 | 10.5 | 23.0 | 0.82 | 39.5 | 64.8 | 0.74 | 5.8 | 72.5 | 0.65 |
| | Think | 42.5 | 71.8 | 0.72 | 48.0 | 80.8 | 0.72 | 53.5 | 91.5 | 0.65 | 4.5 | 96.8 | 0.50 |
| | Think+SysP. | 41.5 | 64.5 | 0.74 | 45.5 | 75.8 | 0.73 | 59.0 | 92.5 | 0.68 | 6.2 | 96.2 | 0.53 |
| | SIRA | 0.0 | 49.5 | 0.30 | 0.0 | 51.2 | 0.33 | 0.0 | 76.2 | 0.27 | 0.0 | 82.0 | 0.46 |
| | RLCracker | 62.7 | 93.5 | 0.72 | 61.3 | 96.8 | 0.71 | 62.2 | 95.2 | 0.67 | 77.8 | 82.5 | 0.86 |
| Qwen3-4B | Base | 10.0 | 19.8 | 0.83 | 12.2 | 27.8 | 0.82 | 38.0 | 83.0 | 0.61 | 4.5 | 43.5 | 0.76 |
| | SysP. | 12.5 | 24.8 | 0.83 | 15.0 | 32.5 | 0.82 | 45.0 | 94.0 | 0.61 | 5.5 | 65.0 | 0.69 |
| | Think | 40.8 | 61.5 | 0.76 | 45.0 | 71.0 | 0.75 | 62.0 | 90.8 | 0.71 | 4.8 | 79.5 | 0.62 |
| | Think+SysP. | 38.2 | 60.5 | 0.76 | 46.2 | 73.0 | 0.75 | 63.2 | 93.2 | 0.70 | 6.5 | 86.8 | 0.60 |
| | SIRA | 0.0 | 47.8 | 0.30 | 0.2 | 50.2 | 0.34 | 0.0 | 70.5 | 0.28 | 1.2 | 89.0 | 0.46 |
| | RLCracker | 64.8 | 97.2 | 0.72 | 65.2 | 98.2 | 0.72 | 66.3 | 97.2 | 0.72 | 63.7 | 67.2 | 0.89 |
| Qwen3-8B | Base | 33.5 | 51.5 | 0.80 | 30.0 | 59.2 | 0.80 | 48.8 | 74.2 | 0.74 | 5.2 | 61.0 | 0.73 |
| | SysP. | 32.0 | 48.8 | 0.80 | 38.8 | 64.5 | 0.83 | 52.5 | 81.0 | 0.72 | 6.8 | 71.2 | 0.69 |
| | Think | 54.0 | 79.8 | 0.73 | 51.5 | 73.0 | 0.78 | 57.8 | 86.2 | 0.70 | 7.2 | 76.2 | 0.65 |
| | Think+SysP. | 56.2 | 81.0 | 0.74 | 59.8 | 80.5 | 0.74 | 64.5 | 96.8 | 0.68 | 7.2 | 79.0 | 0.66 |
| | SIRA | 0.0 | 53.0 | 0.31 | 0.2 | 55.8 | 0.35 | 0.0 | 79.5 | 0.29 | 0.5 | 80.0 | 0.52 |
| | RLCracker | 70.0 | 90.3 | 0.78 | 71.3 | 95.5 | 0.76 | 69.3 | 96.5 | 0.71 | 81.8 | 88.5 | 0.84 |
| Qwen2.5-3B -Instruct | Base | 11.5 | 18.8 | 0.83 | 17.0 | 35.5 | 0.83 | 39.2 | 49.8 | 0.85 | 3.5 | 20.5 | 0.85 |
| | SysP. | 27.3 | 36.3 | 0.82 | 35.5 | 50.0 | 0.84 | 43.2 | 55.3 | 0.83 | 5.5 | 25.0 | 0.82 |
| | SIRA | 0.0 | 33.0 | 0.29 | 0.0 | 50.5 | 0.33 | 0.0 | 84.2 | 0.26 | 0.5 | 88.0 | 0.47 |
| | RLCracker | 73.0 | 82.0 | 0.81 | 71.5 | 83.3 | 0.80 | 65.5 | 82.5 | 0.78 | 98.5 | 100. | 0.92 |
| GPT | — | 10.2 | 17.2 | 0.86 | 20.5 | 28.5 | 0.88 | 50.2 | 70.5 | 0.78 | 6.8 | 92.8 | 0.58 |
| DIPPER | — | 1.15 | 10.0 | 0.62 | 5.75 | 22.3 | 0.66 | 0.0 | 0.0 | 0.60 | 4.50 | 76.5 | 0.50 |

Table 25: Rephrased text quality across models and watermarking schemes (1500 tokens).

| Model | Method | EWD | | | SWEET | | | XSIR | | | Unigram | | |
|---|---|---|---|---|---|---|---|---|---|---|---|---|---|
| | | ESR | PPL. | GPTS. | ESR | PPL. | GPTS. | ESR | PPL. | GPTS. | ESR | PPL. | GPTS. |
| Target Watermarked Text | | — | 3.76 | — | — | 3.61 | — | — | 3.81 | — | — | 18.1 | — |
| Qwen3-0.6B | Base | 3.5 | 3.57 | 8.53 | 4.2 | 3.41 | 8.63 | 26.0 | 3.59 | 8.58 | 5.0 | 18.41 | 8.46 |
| | SysP. | 5.8 | 3.58 | 8.54 | 12.5 | 3.41 | 8.63 | 39.2 | 3.6 | 8.6 | 6.2 | 18.44 | 8.47 |
| | Think | 10.8 | 3.55 | 8.51 | 10.2 | 3.36 | 8.61 | 32.0 | 3.54 | 8.57 | 6.0 | 18.34 | 8.45 |
| | Think+SysP. | 14.8 | 3.56 | 8.53 | 16.2 | 3.38 | 8.62 | 38.2 | 3.58 | 8.57 | 5.2 | 18.38 | 8.46 |
| | SIRA | 0.0 | 4.78 | 4.17 | 0.0 | 4.74 | 4.26 | 0.0 | 4.77 | 4.26 | 0.2 | 20.6 | 4.04 |
| | RLCracker | 63.2 | 3.41 | 8.4 | 63.2 | 3.21 | 8.52 | 60.0 | 3.33 | 8.49 | 90.2 | 18.16 | 8.35 |
| Qwen3-1.7B | Base | 4.2 | 3.65 | 8.58 | 6.8 | 3.56 | 8.67 | 30.2 | 3.83 | 8.62 | 5.0 | 18.63 | 8.49 |
| | SysP. | 7.5 | 3.68 | 8.58 | 10.5 | 3.57 | 8.67 | 39.5 | 3.87 | 8.63 | 5.8 | 18.64 | 8.49 |
| | Think | 42.5 | 3.62 | 8.55 | 48.0 | 3.5 | 8.65 | 53.5 | 3.76 | 8.62 | 4.5 | 18.61 | 8.49 |
| | Think+SysP. | 41.5 | 3.62 | 8.56 | 45.5 | 3.54 | 8.66 | 59.0 | 3.78 | 8.62 | 6.2 | 18.62 | 8.49 |
| | SIRA | 0.0 | 4.89 | 4.25 | 0.0 | 4.87 | 4.34 | 0.0 | 5.03 | 4.31 | 0.0 | 20.88 | 4.18 |
| | RLCracker | 62.7 | 3.47 | 8.47 | 61.3 | 3.37 | 8.57 | 62.2 | 3.58 | 8.53 | 77.8 | 18.45 | 8.4 |
| Qwen3-4B | Base | 10.0 | 3.81 | 8.62 | 12.2 | 3.71 | 8.68 | 38 | 3.92 | 8.64 | 4.5 | 18.79 | 8.51 |
| | SysP. | 12.5 | 3.81 | 8.63 | 15.0 | 3.71 | 8.68 | 45.0 | 3.97 | 8.66 | 5.5 | 18.8 | 8.53 |
| | Think | 40.8 | 3.76 | 8.59 | 45.0 | 3.69 | 8.67 | 62.0 | 3.91 | 8.64 | 4.8 | 18.75 | 8.5 |
| | Think+SysP. | 38.2 | 3.8 | 8.62 | 46.2 | 3.7 | 8.67 | 63.2 | 3.91 | 8.64 | 6.5 | 18.78 | 8.51 |
| | SIRA | 0.0 | 5.0 | 4.28 | 0.2 | 5.04 | 4.37 | 0.0 | 5.28 | 4.33 | 1.2 | 21.07 | 4.2 |
| | RLCracker | 64.8 | 3.58 | 8.46 | 65.2 | 3.52 | 8.61 | 66.3 | 3.75 | 8.55 | 63.7 | 18.55 | 8.42 |
| Qwen3-8B | Base | 33.5 | 3.93 | 8.66 | 30.0 | 3.96 | 8.71 | 48.8 | 4.13 | 8.71 | 5.2 | 19.16 | 8.63 |
| | SysP. | 32.0 | 4.15 | 8.67 | 38.8 | 4.0 | 8.75 | 52.5 | 4.23 | 8.72 | 6.8 | 19.66 | 8.64 |
| | Think | 54.0 | 3.92 | 8.66 | 51.5 | 3.79 | 8.7 | 57.8 | 4.08 | 8.68 | 7.2 | 19.08 | 8.55 |
| | Think+SysP. | 56.2 | 3.92 | 8.66 | 59.8 | 3.91 | 8.71 | 64.5 | 4.1 | 8.69 | 7.2 | 19.1 | 8.57 |
| | SIRA | 0.0 | 5.12 | 4.34 | 0.2 | 5.12 | 4.38 | 0.0 | 5.42 | 4.36 | 0.5 | 21.23 | 4.24 |
| | RLCracker | 70.0 | 3.77 | 8.57 | 71.3 | 3.62 | 8.62 | 69.3 | 3.9 | 8.59 | 81.8 | 18.91 | 8.46 |
| Qwen2.5-3B -Instruct | Base | 11.5 | 3.81 | 8.61 | 17.0 | 3.59 | 8.65 | 39.2 | 3.81 | 8.63 | 3.5 | 18.79 | 8.5 |
| | SysP. | 27.3 | 3.83 | 8.63 | 35.5 | 3.64 | 8.78 | 43.2 | 3.85 | 8.68 | 5.5 | 18.8 | 8.54 |
| | SIRA | 0.0 | 4.63 | 4.28 | 0.0 | 4.73 | 4.31 | 0.0 | 4.98 | 4.29 | 0.5 | 20.91 | 4.19 |
| | RLCracker | 73.0 | 3.6 | 8.52 | 71.5 | 3.46 | 8.55 | 65.5 | 3.67 | 8.54 | 98.5 | 18.65 | 8.42 |
| GPT-4o | — | 10.2 | 3.62 | 8.83 | 20.5 | 3.67 | 8.85 | 50.2 | 3.79 | 8.79 | 6.8 | 18.44 | 8.75 |
| DIPPER | — | 1.15 | 3.89 | 8.54 | 5.75 | 3.98 | 8.53 | 0.0 | 5.32 | 8.51 | 4.5 | 22.32 | 8.49 |

Table 26: Evasion Success Rate (ESR, %) across models and watermarking schemes (1500 tokens).

| Model | Method | KGW_self | | | KGW | | | SIR | | | PF | | |
|---|---|---|---|---|---|---|---|---|---|---|---|---|---|
| | | ESR | Rem. | P-SP | ESR | Rem. | P-SP | ESR | Rem. | P-SP | ESR | Rem. | P-SP |
| Qwen3-0.6B | Base | 17.5 | 23.5 | 0.88 | 6.0 | 33.8 | 0.81 | 73.2 | 98.8 | 0.76 | 11.8 | 17.0 | 0.83 |
| | SysP. | 37.0 | 47.2 | 0.87 | 15.0 | 55.8 | 0.77 | 72.1 | 97.5 | 0.76 | 22.0 | 35.0 | 0.81 |
| | Think | 33.0 | 42.0 | 0.88 | 15.8 | 50.2 | 0.77 | 73.8 | 98.2 | 0.78 | 22.5 | 37.2 | 0.80 |
| | Think+SysP. | 47.8 | 58.5 | 0.86 | 19.8 | 62.7 | 0.76 | 72.2 | 99.8 | 0.76 | 30.0 | 49.8 | 0.79 |
| | SIRA | 0.0 | 51.0 | 0.29 | 0.0 | 45.2 | 0.28 | 0.2 | 94.5 | 0.27 | 0.0 | 33.0 | 0.38 |
| | RLCracker | 66.8 | 80.0 | 0.84 | 48.5 | 51.5 | 0.88 | 75.8 | 100.0 | 0.75 | 62.0 | 96.5 | 0.72 |
| Qwen3-1.7B | Base | 32.5 | 37.5 | 0.90 | 9.2 | 41.8 | 0.79 | 78.8 | 100.0 | 0.81 | 19.2 | 26.2 | 0.84 |
| | SysP. | 41.0 | 49.0 | 0.90 | 16.0 | 55.2 | 0.78 | 78.5 | 100.0 | 0.81 | 30.0 | 41.8 | 0.83 |
| | Think | 66.3 | 83.5 | 0.80 | 47.0 | 95.5 | 0.70 | 61.3 | 99.0 | 0.68 | 55.5 | 88.0 | 0.74 |
| | Think+SysP. | 68.5 | 73.8 | 0.81 | 52.5 | 97.0 | 0.72 | 64.0 | 98.2 | 0.69 | 60.2 | 92.0 | 0.74 |
| | SIRA | 0.0 | 69.5 | 0.31 | 0.0 | 70.0 | 0.30 | 0.2 | 99.8 | 0.29 | 0.0 | 63.2 | 0.43 |
| | RLCracker | 68.0 | 78.8 | 0.88 | 60.5 | 74.3 | 0.75 | 80.0 | 99.8 | 0.82 | 61.0 | 92.8 | 0.73 |
| Qwen3-4B | Base | 36.0 | 41.8 | 0.91 | 15.8 | 42.5 | 0.82 | 37.0 | 54.2 | 0.83 | 25.8 | 33.5 | 0.83 |
| | SysP. | 40.8 | 47.0 | 0.90 | 19.2 | 57.5 | 0.80 | 45.0 | 63.2 | 0.82 | 34.5 | 45.2 | 0.83 |
| | Think | 78.0 | 90.0 | 0.84 | 52.2 | 93.5 | 0.74 | 70.5 | 96.2 | 0.73 | 59.0 | 83.8 | 0.76 |
| | Think+SysP. | 79.5 | 93.0 | 0.83 | 54.2 | 96.0 | 0.74 | 64.2 | 90.2 | 0.72 | 65.5 | 91.0 | 0.76 |
| | SIRA | 0.0 | 51.0 | 0.29 | 0.0 | 45.2 | 0.28 | 0.2 | 94.5 | 0.27 | 0.0 | 33.0 | 0.38 |
| | RLCracker | 82.2 | 97.3 | 0.81 | 57.3 | 70.5 | 0.83 | 65.0 | 96.5 | 0.71 | 64.8 | 96.5 | 0.74 |
| Qwen3-8B | Base | 58.8 | 72.0 | 0.83 | 46.8 | 85.0 | 0.78 | 62.5 | 82.2 | 0.79 | 44.3 | 59.5 | 0.84 |
| | SysP. | 64.5 | 75.2 | 0.88 | 51.0 | 90.0 | 0.78 | 64.0 | 84.0 | 0.79 | 50.2 | 68.2 | 0.82 |
| | Think | 71.3 | 94.8 | 0.81 | 55.8 | 96.2 | 0.73 | 64.5 | 96.8 | 0.69 | 59.5 | 87.2 | 0.75 |
| | Think+SysP. | 78.2 | 97.2 | 0.81 | 58.0 | 98.5 | 0.74 | 62.7 | 96.2 | 0.69 | 63.2 | 93.8 | 0.76 |
| | SIRA | 0.0 | 67.2 | 0.32 | 0.0 | 70.2 | 0.31 | 0.0 | 84.0 | 0.30 | 0.2 | 61.5 | 0.44 |
| | RLCracker | 84.3 | 97.3 | 0.83 | 64.5 | 87.5 | 0.78 | 82.5 | 96.5 | 0.78 | 73.3 | 91.5 | 0.79 |
| Qwen2.5-3B -Instruct | Base | 38.0 | 47.0 | 0.89 | 20.5 | 40.3 | 0.87 | 56.2 | 66.5 | 0.88 | 15.2 | 28.5 | 0.84 |
| | SysP. | 63.7 | 70.0 | 0.87 | 44.8 | 59.8 | 0.84 | 54.2 | 70.0 | 0.84 | 39.5 | 54.8 | 0.84 |
| | SIRA | 0.2 | 71.5 | 0.31 | 0.5 | 66.5 | 0.31 | 0.2 | 88.0 | 0.31 | 0.0 | 58.0 | 0.43 |
| | RLCracker | 78.0 | 89.0 | 0.83 | 58.0 | 67.3 | 0.90 | 62.0 | 80.3 | 0.81 | 77.8 | 90.3 | 0.81 |
| GPT-4o | — | 61.8 | 68.2 | 0.90 | 49.2 | 88.5 | 0.79 | 81.8 | 98.5 | 0.82 | 49.3 | 67.0 | 0.80 |
| DIPPER | — | 41.5 | 64.0 | 0.72 | 11.5 | 52.8 | 0.65 | 14.8 | 88.9 | 0.49 | 21.0 | 39.0 | 0.69 |

Table 27: Rephrased text quality across models and watermarking schemes (1500 tokens).

| Model | Method | KGW_self | | | KGW | | | SIR | | | PF | | |
|---|---|---|---|---|---|---|---|---|---|---|---|---|---|
| | | ESR | PPL. | GPTS. | ESR | PPL. | GPTS. | ESR | PPL. | GPTS. | ESR | PPL. | GPTS. |
| Target Watermarked Text | | — | 3.57 | — | — | 17.9 | — | — | 3.8 | — | — | 12.8 | — |
| Qwen3-0.6B | Base | 17.5 | 3.3 | 8.54 | 6.0 | 17.7 | 8.44 | 73.2 | 3.73 | 8.48 | 11.8 | 11.66 | 8.4 |
| | SysP. | 37.0 | 3.34 | 8.55 | 15.0 | 17.73 | 8.45 | 72.1 | 3.73 | 8.49 | 22.0 | 11.68 | 8.4 |
| | Think | 33.0 | 3.29 | 8.53 | 15.8 | 17.38 | 8.42 | 73.8 | 3.67 | 8.45 | 22.5 | 11.38 | 8.36 |
| | Think+SysP. | 47.8 | 3.3 | 8.54 | 19.8 | 17.66 | 8.43 | 72.2 | 3.69 | 8.48 | 30.0 | 11.64 | 8.38 |
| | SIRA | 0.0 | 4.59 | 4.22 | 0.0 | 19.56 | 4.07 | 0.2 | 4.84 | 4.1 | 0.0 | 13.32 | 4.03 |
| | RLCracker | 66.8 | 3.15 | 8.45 | 48.5 | 17.24 | 8.33 | 75.8 | 3.49 | 8.36 | 62.0 | 11.15 | 8.27 |
| Qwen3-1.7B | Base | 32.5 | 3.47 | 8.57 | 9.2 | 17.84 | 8.49 | 78.8 | 3.8 | 8.52 | 19.2 | 11.77 | 8.42 |
| | SysP. | 41.0 | 3.48 | 8.57 | 16.0 | 17.85 | 8.5 | 78.5 | 3.8 | 8.52 | 30.0 | 11.8 | 8.44 |
| | Think | 66.3 | 3.44 | 8.57 | 47.0 | 17.8 | 8.49 | 61.3 | 3.77 | 8.51 | 55.5 | 11.75 | 8.41 |
| | Think+SysP. | 68.5 | 3.45 | 8.57 | 52.5 | 17.84 | 8.49 | 64.0 | 3.77 | 8.51 | 60.2 | 11.76 | 8.41 |
| | SIRA | 0.0 | 4.75 | 4.25 | 0.0 | 20.17 | 4.17 | 0.2 | 5.05 | 4.19 | 0.0 | 14.01 | 4.1 |
| | RLCracker | 68.0 | 3.33 | 8.48 | 60.5 | 17.67 | 8.4 | 80.0 | 3.64 | 8.41 | 61 | 11.62 | 8.32 |
| Qwen3-4B | Base | 36.0 | 3.65 | 8.6 | 15.8 | 18.08 | 8.52 | 37 | 3.83 | 8.55 | 25.8 | 11.99 | 8.46 |
| | SysP. | 40.8 | 3.68 | 8.6 | 19.2 | 18.17 | 8.52 | 45 | 3.85 | 8.57 | 34.5 | 12.02 | 8.46 |
| | Think | 78.0 | 3.62 | 8.58 | 52.2 | 17.98 | 8.52 | 70.5 | 3.82 | 8.55 | 59.0 | 11.91 | 8.45 |
| | Think+SysP. | 79.5 | 3.62 | 8.59 | 54.2 | 18.03 | 8.52 | 64.2 | 3.82 | 8.55 | 65.5 | 11.96 | 8.45 |
| | SIRA | 0.0 | 4.9 | 4.28 | 0.0 | 20.31 | 4.2 | 0.2 | 5.11 | 4.24 | 0.0 | 14.11 | 4.14 |
| | RLCracker | 82.2 | 3.51 | 8.5 | 57.3 | 17.86 | 8.43 | 65.0 | 3.69 | 8.47 | 64.8 | 11.75 | 8.34 |
| Qwen3-8B | Base | 58.8 | 3.76 | 8.65 | 46.8 | 18.57 | 8.56 | 62.5 | 3.94 | 8.61 | 44.3 | 12.48 | 8.51 |
| | SysP. | 64.5 | 3.77 | 8.66 | 51.0 | 18.64 | 8.57 | 64.0 | 4.06 | 8.61 | 50.2 | 12.49 | 8.56 |
| | Think | 71.3 | 3.74 | 8.63 | 55.8 | 18.48 | 8.55 | 64.5 | 3.91 | 8.59 | 59.5 | 12.2 | 8.49 |
| | Think+SysP. | 78.2 | 3.74 | 8.64 | 58.0 | 18.55 | 8.55 | 62.7 | 3.93 | 8.61 | 63.2 | 12.38 | 8.5 |
| | SIRA | 0.0 | 5.13 | 4.32 | 0.0 | 20.59 | 4.23 | 0.0 | 5.17 | 4.28 | 0.2 | 14.44 | 4.17 |
| | RLCracker | 84.3 | 3.63 | 8.55 | 64.5 | 18.18 | 8.45 | 82.5 | 3.78 | 8.51 | 73.3 | 12.07 | 8.4 |
| Qwen2.5-3B -Instruct | Base | 38.0 | 3.56 | 8.56 | 20.5 | 17.9 | 8.48 | 56.2 | 3.83 | 8.57 | 15.2 | 11.83 | 8.47 |
| | SysP. | 63.7 | 3.63 | 8.61 | 44.8 | 17.94 | 8.51 | 54.2 | 3.9 | 8.66 | 39.5 | 11.85 | 8.48 |
| | SIRA | 0.2 | 4.7 | 4.22 | 0.5 | 20.04 | 4.14 | 0.2 | 4.95 | 4.16 | 0.0 | 13.95 | 4.13 |
| | RLCracker | 78.0 | 3.4 | 8.45 | 58 | 17.76 | 8.39 | 62 | 3.67 | 8.44 | 77.8 | 11.7 | 8.37 |
| GPT-4o | — | 61.8 | 3.51 | 8.83 | 49.2 | 17.54 | 8.82 | 81.8 | 3.81 | 8.82 | 49.3 | 11.87 | 8.79 |
| DIPPER | — | 41.5 | 4.13 | 8.54 | 11.5 | 18.83 | 8.51 | 14.8 | 4.83 | 8.54 | 21 | 12.34 | 8.47 |

Table 28: Evasion Success Rate (ESR, %) across models and watermarking schemes (1500 tokens).

| Model | Method | SynthID-Text | | | UPV | | |
|---|---|---|---|---|---|---|---|
| | | ESR | Rem. | P-SP | ESR | Rem. | P-SP |
| Qwen3-0.6B | Base | 38.8 | 48.5 | 0.85 | 39.0 | 47.2 | 0.86 |
| | SysP. | 56.5 | 71.5 | 0.83 | 60.0 | 69.5 | 0.84 |
| | Think | 51.2 | 65.5 | 0.84 | 47.2 | 61.8 | 0.83 |
| | Think+SysP. | 64.8 | 80.5 | 0.83 | 60.5 | 74.8 | 0.81 |
| | SIRA | 0.2 | 64.2 | 0.33 | 1.0 | 58.2 | 0.42 |
| | RLCracker | 79.8 | 98.8 | 0.80 | 86.5 | 99.2 | 0.79 |
| Qwen3-1.7B | Base | 49.0 | 57.5 | 0.87 | 39.0 | 45.2 | 0.89 |
| | SysP. | 60.8 | 74.2 | 0.86 | 48.8 | 56.8 | 0.87 |
| | Think | 77.8 | 99.8 | 0.76 | 69.5 | 97.5 | 0.74 |
| | Think+SysP. | 79.0 | 99.8 | 0.78 | 80.2 | 98.0 | 0.77 |
| | SIRA | 0.2 | 94.5 | 0.36 | 2.0 | 83.5 | 0.45 |
| | RLCracker | 78.0 | 96.5 | 0.81 | 79.2 | 93.5 | 0.79 |
| Qwen3-4B | Base | 54.0 | 68.2 | 0.87 | 30.5 | 38.2 | 0.90 |
| | SysP. | 70.8 | 83.5 | 0.87 | 41.0 | 49.8 | 0.89 |
| | Think | 82.2 | 98.8 | 0.80 | 83.8 | 95.8 | 0.80 |
| | Think+SysP. | 83.2 | 99.8 | 0.80 | 84.5 | 96.8 | 0.79 |
| | SIRA | 0.2 | 95.5 | 0.36 | 2.0 | 70.2 | 0.47 |
| | RLCracker | 80.0 | 98.2 | 0.82 | 83.0 | 96.8 | 0.80 |
| Qwen3-8B | Base | 78.3 | 93.8 | 0.84 | 70.0 | 81.0 | 0.85 |
| | SysP. | 82.8 | 97.0 | 0.85 | 78.0 | 87.0 | 0.86 |
| | Think | 81.0 | 99.8 | 0.78 | 85.2 | 98.5 | 0.79 |
| | Think+SysP. | 81.2 | 100.0 | 0.77 | 89.2 | 99.2 | 0.80 |
| | SIRA | 0.2 | 97.5 | 0.37 | 1.2 | 76.2 | 0.48 |
| | RLCracker | 85.8 | 99.3 | 0.84 | 87.5 | 97.5 | 0.81 |
| Qwen2.5-3B -Instruct | Base | 81.0 | 89.5 | 0.89 | 80.0 | 89.5 | 0.87 |
| | SysP. | 81.2 | 88.8 | 0.86 | 74.8 | 85.8 | 0.83 |
| | SIRA | 0.2 | 96.0 | 0.36 | 0.5 | 88.8 | 0.45 |
| | RLCracker | 83.3 | 91.0 | 0.82 | 81.5 | 91.5 | 0.85 |
| GPT-4o | — | 88.2 | 99.5 | 0.89 | 77.8 | 84.5 | 0.88 |
| DIPPER | — | 50.2 | 99.5 | 0.66 | 58.0 | 89.8 | 0.72 |

Table 29: Rephrased text quality across models and watermarking schemes (1500 tokens).

| Model | Method | SynthID-Text | | | UPV | | |
|---|---|---|---|---|---|---|---|
| | | ESR | PPL. | GPTS. | ESR | PPL. | GPTS. |
| Target Watermarked Text | | — | 2.85 | — | — | 21.7 | — |
| Qwen3-0.6B | Base | 38.8 | 2.86 | 8.49 | 39.0 | 21.5 | 8.43 |
| | SysP. | 56.5 | 2.87 | 8.49 | 60.0 | 21.57 | 8.44 |
| | Think | 51.2 | 2.81 | 8.47 | 47.2 | 21.42 | 8.41 |
| | Think+SysP. | 64.8 | 2.82 | 8.48 | 60.5 | 21.46 | 8.41 |
| | SIRA | 0.2 | 3.43 | 4.09 | 1.0 | 23.78 | 4.06 |
| | RLCracker | 79.8 | 2.7 | 8.37 | 86.5 | 21.13 | 8.31 |
| Qwen3-1.7B | Base | 49.0 | 2.93 | 8.54 | 39.0 | 21.93 | 8.46 |
| | SysP. | 60.8 | 2.94 | 8.55 | 48.8 | 21.95 | 8.47 |
| | Think | 77.8 | 2.91 | 8.53 | 69.5 | 21.79 | 8.45 |
| | Think+SysP. | 79.0 | 2.92 | 8.53 | 80.2 | 21.89 | 8.46 |
| | SIRA | 0.2 | 3.59 | 4.21 | 2.0 | 24.39 | 4.14 |
| | RLCracker | 78.0 | 2.78 | 8.44 | 79.2 | 21.58 | 8.37 |
| Qwen3-4B | Base | 54.0 | 2.98 | 8.57 | 30.5 | 22.16 | 8.48 |
| | SysP. | 70.8 | 3.01 | 8.58 | 41.0 | 22.2 | 8.49 |
| | Think | 82.2 | 2.98 | 8.56 | 83.8 | 21.99 | 8.48 |
| | Think+SysP. | 83.2 | 2.98 | 8.57 | 84.5 | 22.02 | 8.48 |
| | SIRA | 0.2 | 3.67 | 4.26 | 2.0 | 24.67 | 4.18 |
| | RLCracker | 80.0 | 2.86 | 8.48 | 83 | 21.84 | 8.4 |
| Qwen3-8B | Base | 78.3 | 3.12 | 8.63 | 70.0 | 23.03 | 8.5 |
| | SysP. | 82.8 | 3.12 | 8.63 | 78.0 | 23.11 | 8.55 |
| | Think | 81.0 | 3.06 | 8.59 | 85.2 | 22.38 | 8.5 |
| | Think+SysP. | 81.2 | 3.08 | 8.6 | 89.2 | 22.67 | 8.5 |
| | SIRA | 0.2 | 3.72 | 4.28 | 1.2 | 24.94 | 4.19 |
| | RLCracker | 85.8 | 2.91 | 8.51 | 87.5 | 22.17 | 8.41 |
| Qwen2.5-3B -Instruct | Base | 81.0 | 2.96 | 8.57 | 80.0 | 21.71 | 8.49 |
| | SysP. | 81.2 | 2.97 | 8.61 | 74.8 | 21.72 | 8.49 |
| | SIRA | 0.2 | 3.72 | 4.15 | 0.5 | 24.4 | 4.09 |
| | RLCracker | 83.3 | 2.83 | 8.49 | 81.5 | 21.57 | 8.4 |
| GPT-4o | — | 88.2 | 2.63 | 8.81 | 77.8 | 21.14 | 8.78 |
| DIPPER | — | 50.2 | 2.93 | 8.51 | 58.0 | 22.52 | 8.46 |

