# OpenReview forum: "RLCracker: Exposing the Vulnerability of LLM Watermarks with Adaptive RL Attacks"
_ICLR.cc/2026/Conference — ICLR 2026 Conference Desk Rejected Submission_

### Official Review · Reviewer_nyNq · 2025-10-16

**Soundness:** 2
**Presentation:** 3
**Contribution:** 1
**Rating:** 2
**Confidence:** 4

**Summary:**

The authors investigate the robustness of text watermarking schemes for LLMs under adaptive adversarial conditions. They introduce the notion of an adaptive robustness radius as a formal metric for quantifying the semantic margin within which a watermark remains detectable. They also derive a KL-based relaxation that yields a tractable, distribution-level certificate of robustness. Building on this framework, they propose RLCracker, a reinforcement-learning-based watermark removal method that jointly optimizes a semantic similarity reward and a token-wise KL-divergence reward to steer generated text away from the watermark distribution while preserving meaning.

**Strengths:**

- The authors present a clear and well-motivated study exposing critical vulnerabilities in current LLM watermarking schemes under adaptive attacks.
- They introduce the adaptive robustness radius framework and its KL relaxation that provides a formal basis for evaluating worst-case watermark robustness.
- Their method demonstrates strong empirical performance, with RLCracker achieving high watermark removal success while preserving semantic fidelity across diverse models and watermarking methods.
- The paper is well-written and clearly presented.

**Weaknesses:**

- The central idea, training adaptive paraphrasers via reinforcement learning to remove LLM watermarks has already been presented in [1], which the paper does not cite or discuss. That prior work introduced the same adaptive attack setup, nearly identical threat model (no-box, offline, keyless attacker), and a conceptually similar optimization objective. The adaptive robustness radius and KL relaxation are reformulations of equivalent robustness objectives used in that work, not genuinely new theory. The omission of this citation is a significant scholarly lapse and undermines the paper’s originality claim.

- While the adaptive robustness radius is mathematically formalized, it remains mostly a conceptual rephrasing without rigorous empirical validation or ablation tying the theory to the observed performance gains. There is no quantitative analysis showing how the proposed theoretical certificate correlates with actual attack success rates or robustness metrics.

- The proposed RL-based optimization (GRPO) differs from existing DPO-based adaptive attacks only in implementation details, yet the paper presents it as a conceptual breakthrough. The authors should explicitly benchmark RLCracker against preference-optimization baselines such as DPO or PPO-based adaptive paraphrasers to substantiate the claim that their reinforcement-learning formulation yields superior performance.

- The empirical comparison is narrow and misses two key papers [1, 2] on adaptive watermark removal. Without explicit head-to-head comparison or discussion, it is unclear whether RLCracker’s improvements stem from genuine methodological advances or simply better tuning and model scaling. The main surveyed methods aren't adaptive in nature.

**References**

[1] Abdulrahman Diaa, Toluwani Aremu, and Nils Lukas. Optimizing Adaptive Attacks against Watermarks for Language Models. arXiv preprint arXiv:2410.02440, 2025.

[2] Nikola Jovanović, Robin Staab, and Martin Vechev. Watermark Stealing in Large Language Models. arXiv preprint arXiv:2402.19361, 2024.

**Questions:**

- How does *RLCracker* differ fundamentally from Diaa et al. (2025)? Both share nearly identical goals, threat models, and optimization formulations. Please explicitly explain what is new beyond replacing preference optimization (DPO) with GRPO-based RL. If the contribution lies in theoretical formalization (the adaptive robustness radius), please clarify how this framework provides insights or guarantees not already implied by prior robustness definitions.

- Can the authors include experiments comparing RLCracker against DPO-optimized or preference-tuned paraphrasers from prior work?

- The authors define an adaptive robustness radius and derives a KL-based certificate. How does this theoretical quantity correlate with actual watermark evasion rates in experiments?

- Given the dual-use risk of releasing effective watermark-removal tools, how do the authors plan to mitigate potential misuse? Are there safeguards or access restrictions envisioned for releasing code or trained models?

- Could the authors show examples where RLCracker fails or partially removes a watermark, along with explanations of why?

**Details Of Ethics Concerns:**

The paper presents a method explicitly designed to remove or defeat watermarking mechanisms in LLM outputs. These watermarking techniques are intended for content attribution, copyright protection, and misuse prevention. However, the paper should be a cause for further advancement in robust watermark research. While the benefits of publication outweigh the risks (in the living world), I suggest that an ethics reviewer with expertise in AI safety, content attribution, and responsible disclosure of security vulnerabilities is better suited to assess if open-sourcing the implementation and trained models is okay.

---

> ### Author Response · Authors · 2025-11-18
> **Official Response by Authors to Reviewer nyNq (1/3)**
>
> We sincerely thank the reviewer for the thoughtful and constructive feedback. Below, we provide a detailed point-by-point response and hope that our clarifications and revisions effectively address your concerns.
>
> > **Q1: How does RLCracker differ fundamentally from Diaa et al. (2025)? What is the main contribution of RLCracker?**
>
> **A:** We thank the reviewer for the comment. RL is not our novelty, our key contribution is a *distributional robustness lens* that eliminates the need for watermark-generation or verification internals.
>
> 1. **Prior work *requires* full knowledge of *KeyGen*/*Embed*/*Verify*; our method requires none.** Diaa et al.[1] explicitly restrict their adaptive attackers to those ``who know the provider’s watermark algorithms (*KeyGen*, *Embed*, *Verify*)'' (Sec. 3), and their optimization objective (Eq. (2)) depends directly on simulating this full pipeline. In contrast, our distributional view models a watermark as a ***local distribution*** $P_{X,c,\theta}$, letting us approximate watermark generation with a weak paraphraser and approximate verification via KL divergence, provably justified by our lower bound. This perspective fundamentally removes the need for any watermarking internals (no *KeyGen*, no *Embed*, no *Verify*), enabling a true no-box, offline attack.
>
> 2. **Our theory provides an approximate detectability test without the verifier.** While computing the exact adaptive radius requires the detector, Theorem 1 provides a ***necessary condition*** for evasion: the adversarial paraphrasing distribution must move sufficiently far from the local watermark distribution in KL space. This yields a smooth, verifier-free proxy for detectability and provides a principled reward signal for training. Diaa et al.[1] cannot express such a test, as their instance-level formulation relies entirely on a surrogate verifier.
> 3. **Our training objective targets distributional shift, not preference alignment.** Diaa et al.[1] optimize preference pairs using a surrogate verifier, treating watermark removal as alignment. RLCracker instead drives a KL-guided shift away from the watermark’s local distribution and toward human-like text, directly matching the distributional robustness framework above.
>
> In short, the novelty is not RL but the ***distributional modeling + KL robustness framework***, which bypasses watermark internals and yields guarantees unavailable in prior work. We have revised Sec. 4.1 in the updated manuscript to highlight this contribution more clearly, and we also added explicit notation-level differences to avoid confusion with the formulation used in Diaa et al.[1].

---

> > ### Author Response · Authors · 2025-11-18
> > **Official Response by Authors to Reviewer nyNq (2/3)**
> >
> > > **Q2: What is genuinely new about using GRPO instead of DPO? The authors should compare RLCracker against DPO-based or preference-optimized paraphrasers to demonstrate any advantage.**
> >
> > **A:** We thank the reviewer for the question. We want to point out that RL itself is not a breakthrough, our novelty lies in the ***distributional robustness lens*** introduced in Sec. 4.1, not in the choice of optimizer.
> >
> > DPO-based adaptive attacks require ***positive and negative*** preference pairs, which in turn require a surrogate verifier and detailed knowledge of the watermark-generation algorithms. This makes DPO fundamentally incompatible with our no-box, offline setting, where the attacker cannot access *KeyGen*, *Embed*, or *Verify*. Our new distributional lens resolves this challenge: the KL lower bound provides a ***verifier-free approximation*** of detectability, so the attack only needs to enlarge the distributional shift away from the local watermark distribution. GRPO fits this setting naturally, since it optimizes smooth rewards without relying on a verifier, whereas DPO cannot operate without one.
> >
> > Nevertheless, to directly address the reviewer’s concern, we implemented the DPO attack following Diaa et al.[1], training Qwen2.5-3B-Instruct with both 100 preference pairs (matching the data budget used by RLCracker) and 7000 preference pairs (the configuration used in [1]. As shown below, ***RLCracker decisively outperforms both DPO variants across all watermarking schemes***, with the gap becoming especially large on long (1500-token) sequences.
> >
> > These results confirm that, even when supplied with thousands of preference pairs and surrogate verifier access, the DPO attack remains substantially weaker than RLCracker trained under the stricter no-box assumptions.
> > | Length       | Method       | Unigram ESR | Unigram P-SP | KGW_self ESR | KGW_self P-SP | PF ESR  | PF P-SP |
> > |--------------|--------------|-------------|--------------|--------------|---------------|---------|---------|
> > | 500 tokens   | DPO(100)     | 45.2        | 0.81         | 78.8         | 0.89          | 69.3    | 0.85    |
> > |              | DPO(7000)    | 69.3        | 0.71         | 81           | 0.82          | 71.8    | 0.79    |
> > |              | RLCracker    | 78.5        | 0.81         | 89.8         | 0.86          | 81.8    | 0.84    |
> > | 1500 tokens  | DPO(100)     | 7.25        | 0.69         | 65.8         | 0.87          | 41.3    | 0.83    |
> > |              | DPO(7000)    | 20.8        | 0.67         | 74           | 0.84          | 47      | 0.75    |
> > |              | RLCracker    | 98.5        | 0.92         | 78           | 0.83          | 77.8    | 0.81    |
> >
> > > **Q3: The empirical comparison is narrow and misses two key papers. The main surveyed methods aren't adaptive in nature.**
> >
> > **A:** We thank the reviewer for the helpful comments. The two referenced papers assume a ***much stronger*** threat model than ours. Prior work defines an “adaptive attacker’’ as one who knows ***all*** watermarking mechanisms (*KeyGen*, *Embed*, *Verify*) and can compute or query the exact scoring rule. In contrast, RLCracker uses a more general form of adaptiveness: the attacker has ***no*** access to watermark internals and only needs to ***sample*** watermarked data of the same type. This distributional adaptiveness, shifting the paraphrasing distribution away from the watermark’s local distribution, does not require any detector or watermark-specific logic.
> >
> > Under this no-box assumption, prior baselines become incompatible:
> >
> > -- Diaa et al.[1] require explicit access to detection rules to construct positive/negative pairs.
> > -- Jovanović et al.[2] rely on KGW scoring, 30,000 adaptive queries, and a surrogate verifier for reward shaping.
> >
> > These assumptions violate our setting. Our improvement is that we can still create ***large and informative watermarked corpora*** using only a weak paraphraser and our distributional robustness lens that no verifier and no watermark internals required.
> >
> > Nevertheless, to address the reviewer’s concern, we implemented the WS attack using Qwen2.5-3B-Instruct. As shown below, **RLCracker outperforms WS and also surpasses DPO-based attacks**, demonstrating that our gains stem from the distributional framework rather than tuning or scale.
> >
> > We have also updated Appendix B to discuss these adaptive attacks in detail and clarify the distinction between traditional algorithm-aware adaptiveness and our distributional, no-box adaptiveness.
> >
> > | Method      | KGW ESR | KGW P-SP | KGW_self ESR | KGW_self P-SP |
> > |-------------|---------|----------|--------------|---------------|
> > | WS          | 91      | 0.87     | 89           | 0.86          |
> > | DPO(7000)   | 67.3    | 0.73     | 81           | 0.82          |
> > | RLCracker   | 97.5    | 0.86     | 89.8         | 0.86          |

---

> > > ### Author Response · Authors · 2025-11-18
> > > **Official Response by Authors to Reviewer nyNq (3/3)**
> > >
> > > > **Q4: How does this theoretical quantity correlate with actual watermark evasion rates in experiments?**
> > >
> > > **A:** We thank the reviewer for the thoughtful question. The adaptive robustness radius $r_{\mathrm{KL}}$ itself is not computable, as it requires optimization over all adversarial distributions. Our empirical analysis therefore uses the ***computable lower bound*** $\rho^*$ derived in Theorem 1, which provides a certified approximation of $r_{\mathrm{KL}}$ under mild assumptions.
> > >
> > > As reported in Appendix E.9 (Table 13), this computable certificate correlates strongly with actual watermark evasion performance across four watermarking schemes (EWD, SWEET, KGW, KGW\_self) and the entire Qwen3 family: Pearson correlations reach $-0.77$ and Spearman correlations exceed $-0.9$. Smaller $\rho^*$ consistently predicts higher removal rates.
> > >
> > > We further validated this relationship through controlled ablations. For Qwen3-0.6B, modifying only the attack context via the SysP setup reduces the certificate from $1.35$ to $0.58$, while simultaneously increasing the removal rate, exactly matching the theoretical prediction that shrinking $\rho^*$ expands the feasible adversarial region.
> > >
> > > Together, these cross-model correlations and context ablations demonstrate that the computable lower bound $\rho^*$ is not merely conceptual but a ***practically predictive*** indicator of watermark vulnerability, making it a useful self-evaluation tool for watermark designers, who already have access to the detector.
> > >
> > >
> > >
> > > > **Q5: How do the authors plan to mitigate potential misuse? Are there safeguards or access restrictions envisioned for releasing code or trained models?**
> > >
> > > **A:** We thank the reviewer for raising this important point. Our work aims to expose structural weaknesses in current watermarking schemes so that designers can ***systematically self-evaluate the robustness of their watermarks before deployment***. The goal is to improve safety, not to facilitate misuse, and we have considered the dual-use implications carefully.
> > >
> > > To mitigate risks, we will ***not*** release trained models or the full attack pipeline. Only non-sensitive components will be made public for basic reproducibility. Access to the core code will follow a controlled-release process (similar to Meta’s): researchers must request access individually, agree to responsible-use terms, and provide an academic or institutional affiliation. This ensures that legitimate research is supported while minimizing the risk of broad misuse.
> > >
> > > We believe this approach strikes an appropriate balance between scientific transparency and responsible stewardship. We are also open to working with the program chairs on any further safeguards they recommend.
> > >
> > >
> > >
> > > > **Q6: Could the authors show examples where RLCracker fails or partially removes a watermark, along with explanations of why?**
> > >
> > > **A:** We thank the reviewer for the insightful question. To evaluate RLCracker’s effectiveness, we focus on both watermark removal and semantic preservation. As shown in Tables 20-25 in Appendix D, RLCracker effectively removes watermarks but faces challenges with longer texts (1500 tokens), where the trade-off between watermark removal and semantic fidelity becomes more pronounced. While RLCracker reduces watermark signals, complete removal often introduces slight semantic shifts, lowering the P-SP score, leading to partial removal. This aligns with prior work [3], which found that watermark in longer texts (i.e., longer than 800 tokens) is harder to remove.
> > >
> > > [1] Abdulrahman Diaa, Toluwani Aremu, and Nils Lukas. Optimizing Adaptive Attacks against Watermarks for Language Models. arXiv preprint arXiv:2410.02440, 2025.
> > >
> > > [2] Nikola Jovanović, Robin Staab, and Martin Vechev. Watermark Stealing in Large Language Models. arXiv preprint arXiv:2402.19361, 2024.
> > >
> > > [3] Kirchenbauer, John, et al. "On the reliability of watermarks for large language models." arXiv preprint arXiv:2306.04634 (2023).

---

> > > > ### Comment · Reviewer_aaTd · 2025-11-25
> > > > **Public release of code**
> > > >
> > > > Personally, I think it would be fine to publicly release the code, as originally stated in the paper's Reproducibility Statement. I think that the potential for real-world harm is not very significant, and the core method and many implementation details are already described in the paper anyways. But I will defer to the ethics review chairs who have more expertise on this matter.

---

> > > > > ### Comment · Reviewer_nyNq · 2025-11-25
> > > > >
> > > > > I agree!

---

> ### Author Response · Authors · 2025-11-22
> **Looking Forward to Constructive Feedback and Further Discussion**
>
> Dear Reviewer nyNq,
>
> Thank you very much for your thoughtful review. We greatly appreciate the time and effort you've invested in assessing our manuscript. We have carefully revised our response and would be grateful if you could take a moment to review our updated general response, as well as our specific response to your comments. We hope that these revisions address your concerns, but if there are still aspects that may warrant further discussion, we would greatly appreciate your guidance.
>
> Once again, we deeply appreciate your time and valuable feedback.
>
> Best regards,
>  The Authors

---

> ### Comment · Reviewer_nyNq · 2025-11-22
>
> Thank you for providing a detailed rebuttal that successfully clarifies some of my main concerns, most particularly regarding the distinction between RLCracker and prior adaptive attacks (Diaa et al. 2025).
>
> That said, while I recognize this distinction, the end goal remains largely the same to demonstrate that adaptive paraphrasers can effectively erase watermarks while maintaining semantic fidelity. In fact, Diaa and co already showed that a paraphraser trained on one watermark can generalize to others without explicit key/watermark knowledge. I think their model weights are publicly available, so a direct experimental comparison between RLCracker and their paraphraser on a standard watermark such as KGW (with a sufficiently high δ) would make the empirical contribution far more convincing.
>
> However, the paper's goal is to not compete but to show that watermarks are still not robust, which brings me to my next point, that evasion attacks (adaptive or non-adaptive) on watermarking are becoming increasingly incremental, largely because of the fundamental limitations in watermark robustness highlighted in “Watermarks in the Sand” (Zhang et al., 2024). Regardless, it remains valuable to publish new attack methodologies to serve as benchmarks for future robust watermark designs to ensure that new watermarking techniques are stress-tested under the most adaptive threat models. So, kudos!
>
> That said, the paper still lacks a formal theoretical grounding for why the proposed KL-based radius provides a measurable or predictive notion of robustness (or maybe I am not seeing it), and the connection between this quantity and empirical detection outcomes remains heuristic. While the authors added additional baselines and clarified ethical aspects, their evaluation scope is still limited, particularly in comparisons to other adaptive paraphrasers and across longer text domains. It might not matter, current experiments show that it works.
>
> Despite these remaining issues, I acknowledge that the rebuttal sufficiently differentiates the paper’s conceptual framing and threat model from Diaa's work. For that reason, I am raising my score to 6. Personally, I lean toward research that either studies the fundamental problem or solves the watermarking challenge rather than further exposing its well-known vulnerabilities, but I also recognize the importance of this work as a benchmark and diagnostic contribution for future robustness studies. So, kudos to the authors! Goodluck and thanks for your contribution!
>
> Note: if the authors can respond to my other concerns, and maybe run the experiments I mentioned, that would be good!

---

> > ### Author Response · Authors · 2025-11-24
> > **Official Response by Authors to Reviewer nyNq (1/2)**
> >
> > We sincerely thank you for your positive feedback and for kindly increasing the rating to 6. We also deeply appreciate and acknowledge your guidance on improving our manuscript. Below, we provide a detailed point-by-point response and hope that our clarifications can effectively address your concerns.
> >
> > > **Effectiveness comparison with Diass's method:**
> >
> > We thank the reviewer for the helpful suggestion. Following your recommendation, we directly compare RLCracker against Diaa et al.’s DPO-based paraphraser [1] using their released LoRA checkpoints. The results are presented in the tables below. Despite being trained on only 7,000 samples and using smaller base models (e.g., Qwen3-0.6B and Qwen2.5-3B), RLCracker consistently matches or outperforms the DPO model across both 500-token and 1500-token settings on the KGW and KGW_self watermarks.
> >
> > For the Unigram watermark, where the DPO paraphraser underperforms substantially, RLCracker achieves much higher ESR and stronger P-SP fidelity, demonstrating robust generalization capability. These results clearly show that RLCracker not only adapts effectively across different watermark types but also provides comparable or superior performance to the DPO paraphraser, while operating under significantly lighter training requirements.
> >
> > We hope these additional experiments directly address your request and provide strong empirical evidence supporting the effectiveness of RLCracker.
> >
> > | Method                      | KGW (500) ESR | KGW (500) P-SP | KGW (1500) ESR | KGW (1500) P-SP |
> > |-----------------------------|---------------|----------------|----------------|-----------------|
> > | DPO (Llama3.1-8B)           | 67.5          | 0.71           | 47.5           | 0.76            |
> > | RLCracker (Qwen3-0.6B)      | 80.8          | 0.89           | 48.5           | 0.88            |
> > | RLCracker (Qwen2.5-3B)      | 97.5          | 0.86           | 57.3           | 0.83            |
> > | RLCracker (Qwen3-8B)        | 91.5          | 0.90            | 64.5           | 0.78            |
> >
> > | Method                      | KGW_self (500) ESR | KGW_self (500) P-SP | KGW_self (1500) ESR | KGW_self (1500) P-SP |
> > |-----------------------------|--------------------|---------------------|---------------------|----------------------|
> > | DPO (Llama3.1-8B)           | 84.5               | 0.86                | 74.0                  | 0.84                 |
> > | RLCracker (Qwen3-0.6B)      | 85.2               | 0.87                | 66.8                | 0.84                 |
> > | RLCracker (Qwen2.5-3B)      | 89.8               | 0.86                | 78.0                 | 0.83                 |
> > | RLCracker (Qwen3-8B)        | 90.8               | 0.90                 | 84.3                | 0.83                 |
> >
> > | Method                      | Unigram (500) ESR | Unigram (500) P-SP | Unigram (1500) ESR | Unigram (1500) P-SP |
> > |-----------------------------|-------------------|--------------------|--------------------|---------------------|
> > | DPO (Llama3.2-3B)           | 58.5              | 0.70                | 15.5               | 0.66                |
> > | DPO (Qwen2.5-3B)            | 61.3              | 0.71               | 16.8               | 0.66                |
> > | RLCracker (Qwen3-0.6B)      | 67.0                | 0.86               | 90.2               | 0.93                |
> > | RLCracker (Qwen2.5-3B)      | 78.5              | 0.81               | 98.5               | 0.92                |
> > | RLCracker (Qwen3-8B)        | 80.5              | 0.80                | 81.8               | 0.84                |

---

> > > ### Author Response · Authors · 2025-11-24
> > > **Official Response by Authors to Reviewer nyNq (2/2)**
> > >
> > > > **The evasion attacks on watermarking explored in this paper are becoming increasingly incremental.**
> > >
> > > We thank the reviewer for the thoughtful comment and strongly agree with the broader perspective. Our goal is not to compete with prior attacks, but to highlight a deeper limitation of current watermarking schemes. Our KL-based robustness radius formalizes the idea that watermark removal becomes possible as long as the watermark distribution deviates from the human distribution.
> > >
> > > This provides an alternative view of the fundamental fragility emphasized in works such as *Watermarks in the Sand* [2] that **any non-zero distributional gap can be systematically learned and exploited by an adaptive paraphraser**. In this sense, our findings complement prior results by showing that robustness failure is not tied to the sophistication of a particular attack, but arises from an inherent mismatch between watermarked and natural human text distributions. We hope this perspective helps guide the design and stress-testing of future watermarking schemes under more adaptive threat models.
> > >
> > > > **Why the proposed KL-based radius provides a measurable or predictive notion of robustness? And the evaluation scope is still limited.**
> > >
> > > We thank the reviewer for the insightful comments. Our KL-based robustness radius is not proposed as a heuristic proxy, but as a formally derived lower bound on the adaptive robustness radius under a natural sub-Gaussian concentration assumption (Assumption 1). As shown in Section 4.1, the KL radius arises directly from a geometric relaxation of the worst-case paraphrasing problem and yields an explicit and provable robustness certificate (Theorem 1). This certificate provides a necessary condition for watermark resilience: the watermark remains detectable only if every adversarial paraphrasing distribution remains within a bounded KL neighborhood of the local watermark distribution.
> > >
> > > The goal of the KL radius is not to predict exact detection scores, but to give a principled, distribution-level measure of how much semantic drift a watermark can tolerate. This perspective complements empirical evaluations by showing that robustness fundamentally depends on how tightly the watermark distribution concentrates and how large an adversarial search space the attacker controls. The empirical results then illustrate that, in practice, relatively small distributional shifts are already sufficient for an adaptive paraphraser to breach this certified region, consistent with the theory.
> > >
> > > Regarding evaluation scope, as noted in our response to Q3, existing adaptive paraphrasers such as [1,3] operate under substantially stronger threat models that are incompatible with ours, limiting direct comparisons. Nevertheless, as the reviewer also observed, our experiments already demonstrate that RLCracker is effective in erasing a range of watermarks while preserving semantic fidelity. We hope that the KL-based robustness framework and our empirical findings together provide a useful diagnostic tool for stress-testing future watermarking schemes under adaptive threat models.
> > >
> > > Once again, we thank the reviewer for your thoughtful comments and effort in improving our manuscript.
> > >
> > > [1] Abdulrahman Diaa, Toluwani Aremu, and Nils Lukas. Optimizing Adaptive Attacks against Watermarks for Language Models. arXiv preprint arXiv:2410.02440, 2025.
> > >
> > >
> > > [2] Zhang, Hanlin, et al. "Watermarks in the sand: Impossibility of strong watermarking for generative models." arXiv preprint arXiv:2311.04378 (2023).
> > >
> > > [3] Nikola Jovanović, Robin Staab, and Martin Vechev. Watermark Stealing in Large Language Models. arXiv preprint arXiv:2402.19361, 2024.

---

> > > > ### Comment · Reviewer_nyNq · 2025-11-25
> > > >
> > > > Thank you for providing extra experiments and explanations/answers to my other concerns. I see this work as a go-to for future robustness studies. You should add the experiments to the appendix if you can. Kudos and goodluck!!!

---

> > > > > ### Author Response · Authors · 2025-11-28
> > > > > **Official Response by Authors to Reviewer nyNq**
> > > > >
> > > > > We are genuinely grateful for your thoughtful feedback and are encouraged that we have been able to address your concerns. We have included the additional experiments comparing RLCracker with the DPO-based attack [1] and the WS attack [2] in Appendix E.11. We have also provided further clarification regarding the public release of our code in our general response. We truly appreciate the time you have taken to review our work and the constructive suggestions you have offered, which have been invaluable in enhancing our manuscript. If you have any further concerns, we would be grateful for any additional suggestions that could help us further strengthen the final version.
> > > > >
> > > > > [1] Abdulrahman Diaa, Toluwani Aremu, and Nils Lukas. Optimizing Adaptive Attacks against Watermarks for Language Models. arXiv preprint arXiv:2410.02440, 2025.
> > > > >
> > > > > [3] Nikola Jovanović, Robin Staab, and Martin Vechev. Watermark Stealing in Large Language Models. arXiv preprint arXiv:2402.19361, 2024.

---

### Official Review · Reviewer_gbCX · 2025-10-26

**Soundness:** 3
**Presentation:** 3
**Contribution:** 3
**Rating:** 8
**Confidence:** 3

**Summary:**

In this paper, the authors first introduce the concept of an adaptive robustness radius to quantify the resilience of LLM watermarks against adaptive adversaries. This analysis demonstrates that LLM watermarks can be vulnerable when adversaries adaptively optimize their attack prompts and model parameters. The authors then propose a powerful RL-based watermark attack, RLCracker. With limited training data, RLCracker learns an effective adversarial paraphraser that can erase watermarks while preserving the original text meaning. Extensive experiments show that RLCracker achieves state-of-the-art performance.

**Strengths:**

- The paper is well-written, and it’s clear that the authors put significant effort into making it high-quality. I really enjoyed reading it.
- The paper provides many interesting and valuable insights that are highly beneficial to the community, especially those discussed in Section 5.2.
- The proposed method is interesting and achieves strong performance. Moreover, it is transferable across different models, which makes it more flexible. In addition, the method requires only a very small dataset, making it practical and realistic.
- The experimental section is comprehensive and well-executed.

**Weaknesses:**

- All attacker models are from the Qwen family. I wonder whether other attacker models, such as Llama, could also achieve similarly impressive performance.
- To my understanding, the current RL training pipeline is not fully black-box, as the training objective relies on the watermark-induced distribution, which in turn requires access to the watermark model’s logits. This could be a limitation of the work, since such access is typically unavailable for commercial-level models.

**Questions:**

- I’d like to double-check my concern mentioned above: for the token-wise KL reward, does the method require access to the logits of the watermarked model? I would assume that a more realistic threat model is one where the attacker only has access to a detector — that is, by providing a (paraphrased) text to the detector and receiving a binary watermark signal. I believe the proposed method could be generalized to such a scenario, though the performance might degrade due to the sparsity of the reward signal.
- How does the proposed method perform across different watermarking methods? For example, if the model is trained on one watermark, can it effectively transfer to another?
- Have the authors evaluated the attack on any semantic watermarking approaches, such as https://arxiv.org/abs/2311.08721?

---

> ### Author Response · Authors · 2025-11-18
> **Official Response by Authors to Reviewer gbCX (1/2)**
>
> We sincerely thank the reviewer for the thoughtful and constructive feedback. Below, we provide a detailed point-by-point response and hope that our clarifications and revisions effectively address your concerns.
>
> > **Q1: RLCracker can generalize to other attacker models from different model families?**
>
> **A:** We sincerely thank the reviewer for the insightful question about model choices. We used Llama3.2-3B-Instruct as the attacker model on EWD, PF, and SWEET watermarks. As shown in the table, RLCracker outperforms both Base and SysP methods with significantly higher ESR, demonstrating its superior performance and versatility across different model families.
>
> | Length       | Method   | EWD ESR | EWD P-SP | SWEET ESR | SWEET P-SP | PF ESR  | PF P-SP |
> |--------------|----------|---------|----------|-----------|------------|---------|---------|
> | 500 tokens   | Base     | 26.8    | 0.91     | 42.3      | 0.92       | 36.5    | 0.85    |
> |              | SysP.    | 66.5    | 0.89     | 76.3      | 0.89       | 67.8    | 0.83    |
> |              | RLCracker| 89.5    | 0.84     | 90.8      | 0.84       | 79.3    | 0.80    |
> | 1500 tokens  | Base     | 6       | 0.82     | 10        | 0.81       | 18.3    | 0.79    |
> |              | SysP.    | 27      | 0.77     | 38.5      | 0.76       | 43.8    | 0.79    |
> |              | RLCracker| 60      | 0.72     | 61.8      | 0.71       | 71      | 0.75    |
>
>
>
> > **Q2: (1) Does the method require access to the logits of the watermarked model to approximate $P_{wm}$? (2) Allowing access to the detector is more realistic.**
>
> **A: (1)** We thank the reviewer for raising this important clarification. RLCracker does ***not*** require access to the watermarked model’s logits. The entire RL pipeline operates in a fully black-box setting with respect to both the watermark generator and the detector.
>
> For the token-wise KL reward, $P_{wm}$ is ***not*** taken from the target model. As clarified in Sec. 4.2 (Line 313-316), we approximate $P_{wm}$ by weakly rewriting the watermarked instance $wr$ using a weak paraphraser $\pi_{\text{ref}}$. Prior watermarking work [1] shows that such shallow paraphrasing reliably preserves watermark-induced structure, so $\pi_{\text{ref}}(wr)$ provides a practical black-box proxy for the watermark distribution.  Table 1, 8, 15-25 empirically supports that this approximation behaves like a true watermark reference, indicated by low ESRs and Removal rates of the Base methods of smaller models.
>
> Similarly, $P_{h}$ is obtained by querying the *same* reference model with the original question $q$, motivated by the fact that modern LLMs closely track natural language distributions. Thus, neither $P_{wm}$ nor $P_{h}$ requires logits or internal scores from the watermarked model.
>
> **(2)** We also appreciate the reviewer’s suggestion of considering stricter threat models such as detector-only feedback. RLCracker can be extended to such settings, though sparse rewards may reduce efficiency. This aligns with recent findings in detector-based attacks, which typically require thousands of preference pairs and surrogate verifiers.
>
> Motivated by this suggestion, we additionally evaluate RLCracker against a state-of-the-art detector-based baseline. We implement the DPO attack following [2], training Qwen2.5-3B-Instruct with both 100 preference pairs (matching RLCracker’s data budget) and 7000 preference pairs (their original configuration). As shown below, **RLCracker consistently outperforms both DPO variants across all watermarking schemes**, with the advantage becoming especially pronounced on long (1500-token) sequences.
>
> | Length       | Method       | Unigram ESR | Unigram P-SP | KGW_self ESR | KGW_self P-SP | PF ESR  | PF P-SP |
> |--------------|--------------|-------------|--------------|--------------|---------------|---------|---------|
> | 500 tokens   | DPO(100)     | 45.2        | 0.81         | 78.8         | 0.89          | 69.3    | 0.85    |
> |              | DPO(7000)    | 69.3        | 0.71         | 81           | 0.82          | 71.8    | 0.79    |
> |              | RLCracker    | 78.5        | 0.81         | 89.8         | 0.86          | 81.8    | 0.84    |
> | 1500 tokens  | DPO(100)     | 7.25        | 0.69         | 65.8         | 0.87          | 41.3    | 0.83    |
> |              | DPO(7000)    | 20.8        | 0.67         | 74           | 0.84          | 47      | 0.75    |
> |              | RLCracker    | 98.5        | 0.92         | 78           | 0.83          | 77.8    | 0.81    |
>
> These results reinforce the practicality of RLCracker’s fully black-box formulation and its robustness relative to methods that rely on verifier feedback or large-scale preference datasets.

---

> > ### Author Response · Authors · 2025-11-18
> > **Official Response by Authors to Reviewer gbCX (2/2)**
> >
> > > **Q3: Can a model trained on one watermarking method effectively generalize to and remove watermarks from other watermarking methods?**
> >
> > **A:** We appreciate the reviewer for the insightful question. To test watermark removal transferability, we evaluated the Qwen3-4B model trained on EWD data against other watermarking schemes, with ESRs reported in the table below. As shown in the table, its strong removal ability on EWD does not generalize and can even underperform the base model on other watermarks. This is because the attacker learns only the EWD-specific watermark pattern, and shifting the distribution away from EWD can unintentionally move it closer to the distributions induced by other watermarking algorithms, resulting in weak transferability.
> >
> > | Model        | EWD   | PF    | SWEET | Unigram | KGW_self |
> > |--------------|-------|-------|-------|---------|----------|
> > | BaseModel    | 54.3  | 63    | 65.8  | 43.8    | 79.3     |
> > | EWD-Trained  | 93.3  | 36.75 | 41.75 | 13.3    | 41       |
> >
> >
> >
> > > **Q4: Can RLCracker be applied to remove semantic-based watermarks?**
> >
> > **A:** We thank the reviewer for the insightful question. To investigate this, we apply RLCracker using Qwen3-4B on 500-token texts watermarked by semantic-based SemMark[3]. As shown in the table below, RLCracker outperforms all other baseline methods with 45% higher ESRs on SemMark while effectively preserving semantic similarity. We hypothesize this stems from RLCracker’s distributional view: by shifting outputs toward a broader human-like manifold, the RL agent naturally suppresses the subtle semantic biases introduced by SemMark, without needing access to its mechanism.
> >
> > | Method    | ESR   | P-SP  |
> > |-----------|-------|-------|
> > | Base      | 6.5   | 0.97  |
> > | SysP      | 10    | 0.93  |
> > | Think     | 41.3  | 0.84  |
> > | RLCracker | 86.5  | 0.78  |
> >
> >
> >
> > [1] Kirchenbauer, John, et al. "A watermark for large language models." ICML 23.
> >
> > [2] Abdulrahman Diaa, Toluwani Aremu, and Nils Lukas. Optimizing Adaptive Attacks against Watermarks for Language Models. arXiv preprint arXiv:2410.02440, 2025.
> >
> > [3] Ren, Jie, et al. "A robust semantics-based watermark for large language model against paraphrasing." Findings of NAACL 2024.

---

> ### Author Response · Authors · 2025-11-22
> **Looking Forward to Constructive Feedback and Further Discussion**
>
> Dear Reviewer gbCX,
>
> Thank you very much for your thoughtful review. We greatly appreciate the time and effort you've invested in assessing our manuscript. We have carefully revised our response and would be grateful if you could take a moment to review our updated general response, as well as our specific response to your comments. We hope that these revisions address your concerns, but if there are still aspects that may warrant further discussion, we would greatly appreciate your guidance.
>
> Once again, we deeply appreciate your time and valuable feedback.
>
> Best regards,
>  The Authors

---

> ### Author Response · Authors · 2025-11-24
> **Looking Forward to Constructive Feedback and Further Discussion**
>
> Dear Reviewer gbCX,
>
> Thank you very much for your thoughtful and detailed review. We have made additional revisions in Lines 423-425, 428-429, 530-532 and 534-537 to our manuscript based on Reviewer pW6u’s comments. Specifically, we have emphasized that our RLCracker tool can serve as a practical diagnostic resource for watermark designers to assess whether their watermarking schemes meet the required standards before deployment. We hope that the revisions, along with our specific responses to your comments, address your concerns effectively. If there are still any points that require further clarification or discussion, we would greatly appreciate your guidance.
>
> Again, we deeply value your time and constructive feedback, which has significantly contributed to improving our manuscript.
>
> Best regards,
>
> The Authors

---

> > ### Comment · Reviewer_gbCX · 2025-11-25
> >
> > Thank the authors for addressing my questions. I appreciate the additional results, especially those concerning semantic watermarks, which meaningfully strengthen the paper.
> >
> > At the same time, it would be helpful to include some analysis quantifying how different the estimated $P_h$ is from the true $P_h$. From Table 8, RLCracker appears to achieve a low P-SP, which suggests that the gap might be substantial. It would also be interesting to examine whether querying a much stronger model—rather than the same unwatermarked paraphraser—for estimating $P_h$ could improve the estimation quality.
> >
> > Overall, this is a solid paper, and I am maintaining my high score.

---

> ### Author Response · Authors · 2025-11-28
> **Official Response by Authors to Reviewer gbCX (1/2)**
>
> We sincerely thank you for the valuable feedback and greatly appreciate your thoughtful guidance. Below, we provide a detailed point-by-point response and hope that our clarifications and revisions effectively address your concerns.
>
> > **Q1. Analysis of quantifying how different the estimated $P_h$ is from the true $P_h$.**
>
> **A1:** We sincerely thank the reviewer for the insightful question. Since we cannot directly obtain the model logits corresponding to the human-written distribution, we are unable to directly calculate the distance between our estimated $P_h$ and the true human-written distribution. Therefore, we assess the effectiveness of this approximation by comparing the detection score distributions of model-generated text and natural text across various watermark detection algorithms. Specifically, we randomly selected 1,000 samples from the C4 dataset, from which we generated both natural text and model-generated text. We then calculated the detection scores for both types using the SWEET, KGW, and EWD watermark detectors. As illustrated in Figures 8,9,10 in Appendix E.12, we observe that as the model size increases, the detection score distribution of model-generated text shifts closer to that of natural text. This suggests that the model's ability to approximate the human-written distribution improves as its size grows. However, we also note that these shifts are generally subtle, suggesting that even smaller models are capable of effectively approximating the human-written distribution $P_h$.
>
>
>
> > **Q2. The distribution gap between RLCracker output and true human-written text might be substantial.**
>
> **A2:** We thank the reviewer for the helpful observation. We agree that a distributional gap between RLCracker outputs and true human-written text is expected. This gap arises naturally from our optimization setup: $P_{wm}$ is obtained by lightly rewriting the watermarked text, whereas $P_h$ is produced by feeding the question to an unwatermarked reference model that has no knowledge of the watermark and generates stochastic human-like responses. Because $P_h$ and $P_{wm}$ originate from different conditioning contexts, they inherently occupy different regions of the distribution space.
>
> RLCracker is optimized to move away from $P_{wm}$ while moving closer to $P_h$, so its output distribution naturally lies *between* these two references (as shown in Figure 4(b)). Our current goal is to improve watermark evasion while preserving semantics, not to fully match the true human-written distribution. In future work, we plan to incorporate stronger human-likeness priors to further reduce this residual gap.
>
>
>
> > **Q3. Would querying a much stronger model for estimating $P_h$ could improve estimation quality?**
>
> We thank the reviewer for the insightful question. As discussed in Q2, a distributional gap between $P_h$ and $P_{wm}$ is inherent because they are conditioned on fundamentally different inputs: $P_{wm}$ is generated from the watermarked response $wr$, while $P_h$ is generated from the question $q$. Due to the stochasticity of LLMs and the mismatch in conditioning contexts, this gap persists even if $P_h$ were obtained from a substantially stronger model. A more powerful model may produce more fluent or detailed text, but it still has no access to the watermarking context and thus cannot collapse this structural gap.
>
> To empirically evaluate whether a stronger $P_h$ improves the estimation quality, we chose the Qwen3-0.6B as the attack model and tested Qwen3-0.6B and Qwen3-8B as estimators of $P_h$ across three watermarks (EWD, SWEET, PF). As shown in the table below, the ESR, removal rate, and P-SP scores are nearly identical across models on both 500-token and 1500-token test sets, indicating no substantial gains from using a larger model for estimating $P_h$.
>
> Furthermore, Table 8 in our manuscript shows that using a *larger paraphraser* leads to noticeably stronger watermark removal performance. This suggests that RLCracker's performance is primarily constrained by the expressive power of the paraphraser rather than by the accuracy of the $P_h$ estimator. Improving the paraphraser architecture is therefore a more impactful direction for enhancing attack effectiveness.
>
> | Approximater | Length | EWD ESR | EWD Rem. | EWD P-SP |
> |--------------|--------|---------|----------|----------|
> | Qwen3-0.6B   | 500    | 91.5    | 98.5     | 0.83     |
> | Qwen3-8B     | 500    | 91.8    | 98.3     | 0.83     |
> | Qwen3-0.6B   | 1500   | 63.2    | 93.8     | 0.72     |
> | Qwen3-8B     | 1500   | 61.8    | 92.5     | 0.72     |
>
> | Approximater | Length | SWEET ESR | SWEET Rem. | SWEET P-SP |
> |--------------|--------|-----------|------------|------------|
> | Qwen3-0.6B   | 500    | 92.5      | 99         | 0.82       |
> | Qwen3-8B     | 500    | 91.3      | 95.8       | 0.85       |
> | Qwen3-0.6B   | 1500   | 63.2      | 96.8       | 0.71       |
> | Qwen3-8B     | 1500   | 61.5      | 95.3       | 0.71       |

---

> > ### Author Response · Authors · 2025-11-28
> > **Official Response by Authors to Reviewer gbCX (2/2)**
> >
> > | Approximater | Length | PF ESR | PF Rem. | PF P-SP |
> > |--------------|--------|--------|---------|---------|
> > | Qwen3-0.6B   | 500    | 76.8   | 94.8    | 0.78    |
> > | Qwen3-8B     | 500    | 77.5   | 94      | 0.79    |
> > | Qwen3-0.6B   | 1500   | 62     | 96.5    | 0.72    |
> > | Qwen3-8B     | 1500   | 62     | 95.5    | 0.72    |

---

### Official Review · Reviewer_pW6u · 2025-10-29

**Soundness:** 3
**Presentation:** 3
**Contribution:** 3
**Rating:** 6
**Confidence:** 4

**Summary:**

This work develops a technique for scrubbing away LLM watermarks using an RL based adversarial optimization routine. They introduce a formal definition of adversarial robustness to such a scrubbing attack under a similarity constraint akin to bounds in classic adversarial optimization to motivate their specific implementable approach and help explain trends in their experiments.  Then they evaluate a range of watermarks in different data domains and using different models as the watermarker versus the attacker. Their results suggest that with limited samples and black box access to the watermarker's outputs, high attack success rates are possible suggesting the need for continued research into watermarking schemes that are more robust to _adversarial_ attacks rather than just benign corruptions.

**Strengths:**

1. Focus on a complementary adversarial dimension of existing LLM watermark robustness research is well motivated.

2. Evaluation setup concerning watermarks, datasets, and models is thorough and in line with prior work.

3. Baselines attacks considered are limited (mainly prompting) but still informative in showcasing the performance improvements of the proposed RL technique.

4. Table 1 and Fig 4b: Overall headline results of the RL approach in their evaluation setting are convincing, Fig 4b feels like a great teaser or twitter figure.

**Weaknesses:**

### Limitations on future-proofness

### 1.
The RL technique tries to push the text distribution away from P_wm towards P_h which implicitly relies on this "wm model" versus "human" gap; in the limit of extremely strong models and "non-distortionary" imperceptible watermarks, these gaps should disappear. Stated anohter way, given lack of watermark algorithm, key or detector access, the RL based approach will rely on the distortion the wm causes and a pool of samples to learn the difference between the two distributions. This is principled, but again in the limit of progress on models and watermarks, it should become infeasible. I did not write this paper, nor do I necessarily endorse its complete technical argument, but it is a relevant example of the general point (Sadasivan et al. 2023, https://arxiv.org/abs/2303.11156).

### 2.
There is a small but critical clarity issue in what samples are used in the RL loop for understanding the method. The RL approach relies on a pool of watermarked completions to the prompts, and another set of completions representing the "human" distribution. The words "human-like" are used in a few place though so it is unclear whether these are simply unwatermarked rollouts from the same model, or actual ground truth human written completions from internet datasets.
Of particular interest is the potential case that it is indeed "unwatermarked rollouts" from the same model. While suitable for research demonstrations, this is potentially an unrealistic assumption as an API model (say Gemini or Claude) would be watermarked behind the scenes and there would not necessarily be an exact copy of the same API model to query in parallel with the same weights, just with the watermark inactive.

This might matter a lot because the objective is based on increasing KL(q||wm) and decreasing KL(q||"hum"), if "hum" is actually "!wm" then in some sense this is a "single dimensional" problem along the watermark bias axis and should be extremely sample efficient to traverse. If it however is truly held out samples from some other distribution (human text) that is not the same as the watermarking model but with the watermark turned off, then the sample complexity is expected to be worse because in KL(P||Q) the differences between P and Q are a composition of watermark bias, as well as other differences between the two and that noise should slow things down.

### Issues with presentation of certain ablations

### 3.
L401 Fig 4a While an interesting point for future research, the "think time" or "reasoning depth" results are not as significant as the prose makes it seem. Most of the improvement is in the increase from zero to one think units. Relatedly, the "Think" method is also not described clearly enough in Sec 5.1 to understand what "time" means here. Is this something like tokens? Number of rewrites within the context window? Perhaps the fact that the majority of improvement is in the first unit of think time would be more apparent if it was clear what the axis referred to. (Note, I eventually looked at D.3, but core details need to be inlined in Sec 5)

### 4.
The rotating keys or multiple watermarks setting is interesting, but unconvincing in its current form; reviewer wrote a question wondering about it early in reading before they got to this section.

The experiment appears to evaluate whether the RL approach can simultaneously find a paraphrase distribution that is separated from 4 distinct keys rather than 1, but this is unrealistic. It is hard to imagine a real setting where the provider would have a use case for _just_ 4 keys. Notably, it is expected that if every sample in the pool is keyd uniquely (reasonable if keys are tied to say "users" or "domains" or something like that), then the expected performance of the method is unclear.
It is actually possible that (depending on above question about what the "hum" texts actually are) the RL attacker might just learn to generate text that is not like the text the underlying model generates, regardless of the key.
This experiment is interesting because it ablates in a way that tries to uncover the actual mechanism that is being used to avoid the watermark; if in a 100 sample pool with 100 keys ESR is still high, versus ESR is near zero, this paints very different pictures about _how_ the approach is achieving the headline Table 1 results (without detracting from them, to be clear).

**Questions:**

### Misc questions and comments

1. L368  and Table 2 results are slightly disjoint from main Table 1, for instance KGW vs KG_s and SIR versus SIR, and the numbers don't match indicating that they are distinct experimental results. Could the same point instead be made by extracting the "base" row from each of the model row groups and the 1500 token column group to present a "zoomed in" view that is still coherent with main results?

2. Table 4 could be improved by clearly labeling the two "Models" axes to be more clear what model is the wm text generator and what model is the RL'd attacker. Also, including the "diagonal" of the table that is not a shift (same generator and attacker) would improve the clarity of the result.

3. This problem feels symmetric, so, if there is a gap to be learned in evasion, the same gap can be used to learn spoofing, just an interesting followup idea.

---

> ### Author Response · Authors · 2025-11-18
> **Official Response by Authors to Reviewer pW6u (1/2)**
>
> We sincerely thank the reviewer for the thoughtful and constructive feedback. Below, we provide a detailed point-by-point response and hope that our clarifications and revisions effectively address your concerns.
>
>
>
> > **Q1: Can the RLCracker still be effective as watermarking schemes improve and the distortion gap between $P_{wm}$ and $P_h$ disappears?**
>
> **A:** We thank the reviewer for the insightful point. We agree that if a watermark were truly and perfectly distortion-free, then the $P_{wm}$–$P_{h}$ gap would vanish and any distributional attack would necessarily fail. However, as pointed out by [1], “distortion-free’’ watermarks still introduce unavoidable distributional structure in practice once a secret hash or key steers generation. This means that theoretical distortion-freeness does not guarantee practical distribution-freeness.
>
> Our experiments confirm this directly: the PF watermark[2] is theoretically described as distortion-free, yet RLCracker reliably removes it using only small models and sampled watermarked text, revealing that the scheme still leaks a measurable gap between $P_{wm}$ and $P_{h}$.
>
> We view this not as a limitation of our method, but as a characterization of a *necessary condition* for watermark robustness: only when a watermark is truly distortion-free in practice will distributional attacks cease to apply. Accordingly, RLCracker provides a practical diagnostic tool for designers to evaluate whether their watermarking schemes achieve this standard before deployment.
>
>
>
> > **Q2: Clarity issue with the human-like distribution $P_h$ and watermark distribution $P_{\text{wm}}$.**
>
> **A:** We thank the reviewer for identifying this important clarity issue. We address the three concerns directly:
>
> **(1) What samples does RLCracker use in the RL loop, and what does “human-like’’ mean?**
> RLCracker does **not** rely on any external human-written dataset or on a watermark-off version of the target model. As clarified in Sec. 4.2 (blue text in line 313 to 317), we instantiate  $P_{wm}$ by weakly paraphrasing the watermarked instance $wr$ using a small model that typically preserves watermark patterns, and  $P_{h}$ by querying the same *unwatermarked reference model* $\pi_{\text{ref}}$ (e.g., Qwen3-0.6B) with the same question $q$.  Modern LLMs approximate natural human text distributions, so $\pi_{\text{ref}}(q)$ provides a practical, accessible proxy for $P_{h}$ in the strict no-box threat model. We have updated Sec. 4.2 (Line 313-316) and Appendix D.3 to clarify this construction.
>
> **(2) Does RLCracker unrealistically assume access to a “watermark-off’’ version of the same API model?** No. Our threat model does *not* assume access to any internal version of the watermarked API model. API systems such as Claude or Gemini do not reveal such variants, and our method does not require them. The attacker only needs:  the collected watermarked samples $(q, wr)$, and  any publicly available unwatermarked model to approximate $P_{h}$.  Thus, RLCracker remains realistic under strict API-only constraints and does not rely on privileged access to the provider model.
>
> We thank the reviewer again for prompting this clarification. Our updated manuscript now makes the construction of $P_{wm}$ and $P_{h}$ explicit and resolves the ambiguity around the term “human-like.’’
>
> > **Q3: It is unclear what the "think time" means; core details need to be inlined in Sec 5.**
>
> **A:** We thank the reviewer for the valuable question. In this context, "think time" refers to the number of times the model performs reasoning during the rephrase task. We have revised Section 5.2 (Line 453-456) of our manuscript to further clarify the procedure, and we have updated the x-axis label of Fig. 4(a) to "reasoning steps" for better clarity.

---

> > ### Author Response · Authors · 2025-11-18
> > **Official Response by Authors to Reviewer pW6u (2/2)**
> >
> > > **Q4: Using only 4 rotating keys is unrealistic; a more practical setting is one unique key per sample.**
> >
> > **A:**  We appreciate the reviewer’s valuable suggestion for the supplementary experiment. We generated 100 additional training samples, each with a unique secret hash key (*one key per user*), and trained RLCracker on these samples, testing it on a single-key test set. As shown in the table below, even with the one sample per key setup, RLCracker consistently improves watermark removal, with only minimal performance degradation compared to the single-key setting. The superior performance of RLCracker stems from its use of a reference model to convert discrete watermark data into a watermark distribution, enabling the model to effectively learn the general watermark pattern and successfully remove the watermark. We have also revised Section 5.3 (Line 518-519, 524) in the manuscript based on your advise.
> >
> > | Model      | Key   | EWD ESR | EWD P-SP | SWEET ESR | SWEET P-SP | PF ESR | PF P-SP |
> > |------------|-------|---------|----------|-----------|------------|--------|---------|
> > | Qwen3-0.6B | Single| 91.5    | 0.83     | 92.5      | 0.82       | 76.8   | 0.78    |
> > |            | Mixed | 88.8    | 0.87     | 90.5      | 0.88       | 75.5   | 0.81    |
> > | Qwen3-1.7B | Single| 91.8    | 0.83     | 91        | 0.86       | 79.5   | 0.80    |
> > |            | Mixed | 90.8    | 0.83     | 90        | 0.86       | 78.3   | 0.81    |
> > | Qwen3-4B   | Single| 93.3    | 0.87     | 95.8      | 0.85       | 82.3   | 0.82    |
> > |            | Mixed | 92.3    | 0.86     | 94.3      | 0.87       | 80.5   | 0.85    |
> >
> >
> >
> > > **Q5: The Table 2 results may be slightly disjoint from the main Table 1.**
> >
> > **A:** We thank the reviewer for the helpful observation. The discrepancies stem from the fact that the watermarks in Table 1 and Table 2 are not the same: KGW vs. KGW\_self and SIR vs. XSIR are distinct watermarking schemes with different detection rules and robustness characteristics, so their ESRs are not expected to match.
> >
> > Table 2 serves a different purpose: it highlights that even the strongest long-text watermarks become increasingly vulnerable as model capability scales, ***even under the simple Base attack***. This complements Table 1 by providing a focused view of long-text robustness across a broader set of watermark families. We have revised the caption of Table 1 to explicitly point readers to the corresponding long-text results in Table 2. We believe this improves coherence while avoiding redundant repetition of numbers across tables.
> >
> >
> >
> > > **Q6: If there is no distribution shift between the watermark generator and the attacker model, will RLCracker's performance be affected? The presentation of Table 4. could be improved.**
> >
> > **A:** We sincerely appreciate the reviewer’s valuable question. We conducted cross-model experiments using the Qwen3 family models: 0.6B, 1.7B, and 4B, which as presented in Appendix E.10. As shown in Table 14, even when the watermark generator and attacker model are the same, RLCracker can effectively direct the model to generate paraphrased text that diverges from the watermarked distribution. This highlights that RLCracker's effectiveness does not rely on internal output distribution discrepancies between the watermark generator and attacker model. We have also improved the presentation of Table 4 by clearly labeling the two 'Models' axes to distinguish between the watermarked text generator and the RL-based attacker.
> >
> >
> >
> > > **Q7: The gap between $P_{wm}$ and$P_h$ may also be used to learn spoofing attack.**
> >
> > **A:** We fully agree with the reviewer and are deeply encouraged that our work has inspired ideas for implementing similar spoofing attacks. In our future work, we will delve deeper into optimizing our RLCracker and exploring the effectiveness of similar methods in spoofing attack applications.
> >
> >
> >
> >
> >
> > [1] Liu, Aiwei, et al. "Can Watermarked LLMs be Identified by Users via Crafted Prompts?." ICLR 25 spotlight.
> >
> > [2] Zhao, Xuandong, Lei Li, and Yu-Xiang Wang. "Permute-and-Flip: An optimally stable and watermarkable decoder for LLMs." ICLR25.

---

> ### Author Response · Authors · 2025-11-22
> **Looking Forward to Constructive Feedback and Further Discussion**
>
> Dear Reviewer pW6u,
>
> Thank you very much for your thoughtful review. We greatly appreciate the time and effort you've invested in assessing our manuscript. We have carefully revised our response and would be grateful if you could take a moment to review our updated general response, as well as our specific response to your comments. We hope that these revisions address your concerns, but if there are still aspects that may warrant further discussion, we would greatly appreciate your guidance.
>
> Once again, we deeply appreciate your time and valuable feedback.
>
> Best regards,
>  The Authors

---

> > ### Comment · Reviewer_pW6u · 2025-11-22
> >
> > I appreciate the authors' thorough response to my review and the effort invested in running some additional experiments and making a number of updates to their manuscript. Given that my main questions are well addressed and that their proposed additions and edits help strengthen both the clarity and completeness of the work, I am happy to update my score to more fully recommend acceptance.
> >
> > Lingering comments and requests:
> >
> > - I am of course biased but I think that some of the topics in our review discussion are worth mentioning in the camera ready. "RLCracker provides a practical diagnostic tool for designers to evaluate whether their watermarking schemes achieve this standard before deployment." is a great statement for the discussion/conclusion section of the paper. It is up to you, but I would add a comment about how the RLCracker method implicitly relies upon the distinguishability (in practice) of watermarked and non-watermarked text, and efficiently learns this gap. Your work suggest an additional use-case for the algorithm as a stress testing method for checking the indistinguishability claims of proposals for future low-distortion watermarks.
> >
> > - Please check the Table 1 caption, I didn't see the link to Table 2 that was mentioned in the response. I actually suggest just adding a sentence in the paragraph accompanying Table 2 (so around L419-L429) that clarifies that these are different wm methods and/or that it is a standalone ablation.
> >
> > - I appreciate the improved setup for the multi-key ablation. Please add a clause somewhere in L513-L526 clarifying whether the test key is one of the 100 unique keys, or a "101st" held out key used to test the trained attack model.
> >
> > Overall, great work! I hope to see the paper presented at the conference.

---

> > > ### Author Response · Authors · 2025-11-24
> > >
> > > We sincerely thank you for the positive feedback and for kindly increasing your rating. We greatly appreciate your thoughtful guidance, which has helped us strengthen the clarity and presentation of our manuscript. In response, we have made the following targeted revisions:
> > >
> > >
> > >
> > > > **Position RLCracker as a watermark indistinguishability stress-test.**
> > >
> > > Following your recommendation, we have added a dedicated paragraph in the Conclusion and Broader Impact section that highlights RLCracker’s additional role as a diagnostic and stress-testing tool for watermark designers. The new paragraph explains how RLCracker implicitly relies on and efficiently learns the empirical distribution gap between watermarked and unwatermarked text, and how this property can be used to evaluate indistinguishability claims of future low-distortion watermarks prior to deployment. These additions appear in Lines 530-532 and 534-537.
> > >
> > > > **Clarify Table 2 is a standalone long-text ablation.**
> > >
> > > Following your suggestion, we added an explicit sentence in the paragraph accompanying Table 2 (Lines 423-425 and 428-429) stating that Table 2 presents a standalone long-text ablation and uses a different subset of watermarking schemes than Table 1. This removes any ambiguity about whether the two tables report directly comparable results.
> > >
> > > > **Clearer Multi-Key Ablation Setting.**
> > >
> > > We have clarified the generated hash key used in our test data is distinct from the key used in the training data in Lines 521-522.
> > >
> > > Once again, we thank the reviewer for the thoughtful comments and would be grateful for any further suggestions that could help us strengthen the final version.

---

### Official Review · Reviewer_aaTd · 2025-10-31

**Soundness:** 3
**Presentation:** 2
**Contribution:** 3
**Rating:** 6
**Confidence:** 4

**Summary:**

This paper studies watermark removal attacks, i.e., paraphrasing watermarked text such that the semantic meaning is preserved but the detectable watermark is removed. The paper proposes adaptive robustness radius, a formal metric which quantifies watermark robustness to adversarial paraphrasing. An attacker can optimize paraphraser parameters and context to try to minimize this radius.

The paper proposes RLCracker, an RL-based watermark removal attack that does not need access to the watermark detector and needs relatively few watermarked examples. The paper runs extensive experiments across many watermarking schemes and paraphraser model sizes, finding that RLCracker outperforms baseline methods.

**Strengths:**

* The paper takes a principled approach to propose the adaptive robustness radius.
* The core idea behind the RLCracker attack is simple and intuitive, which is based on moving away from the watermark distribution and moving closer to the human distribution.
* The method addresses limitations from previous methods by generalizing better to longer generations, only requiring few watermarked examples, and not requiring access to the watermark detector.
* The experiments are thorough, covering many watermarking schemes and multiple model sizes, providing evidence that the method works across various settings.
* Implementation details and additional experiments are provided in the appendix, which enhances the transparency and reproducibility.

**Weaknesses:**

* The adaptive robustness radius and associated theoretical results do not seem very related/necessary to the RLCracker method. The RLCracker method is simply just to increase KL divergence from the watermark distribution and decrease KL divergence from the human distribution. This does not seem to rely on the formal metrics introduced earlier.
* The abstract and introduction state that *“optimizing the attack context and model parameters can substantially reduce this radius.”* However, the radius is defined in terms of holding for all $\\theta \\in \\Theta, c \\in \\mathcal{C}$ or for all distributions $Q$, so isn’t the radius a constant with respect to the space of parameters/distributions?
* The method uses $\\pi\_{\\text{ref}}(o\_{i,t} \\mid wr, o\_{i, \<t})$ to approximate $P\_{wm}$. However, it seems difficult for the non-watermarked $\\pi\_{\\text{ref}}$ to approximate the watermark distribution, especially for more complex watermark signals. The paper does not empirically show how good this approximation is.
* Another potentially important metric is text quality, which is not evaluated in this paper. A paraphrase could be semantically similar to the original, but still worse in other aspects, such as fluency, clarity, style, etc. This could happen if the paraphraser model is much smaller/less capable than the original model.
* There are some ambiguities and imprecisions in parts of the mathematical notation, which are listed in the questions section below.
* The paper doesn’t include empirical or theoretical justifications for why Assumption 1 (Sub-Gaussian Detector Score) should generally hold for watermarking schemes.
* Some results appear to be deferred to the appendix, but there is no pointer to these sections of the appendix in the main paper (see questions for details).

**Questions:**

1. I think that the abstract should state that the P-SP is a semantic similarity score, since readers may be unfamiliar with it.
2. Line 40 cites [He at al. (2024)](https://arxiv.org/abs/2402.14007) to claim that existing watermarking schemes are robust against translation, but the cited paper finds that translation is an effective attack to remove watermarks.
3. Since paraphraser models are usually not deterministic, is $\\mathbf{X’}$ a random variable?
   * If it is a random variable, then it would be clearer to have $\\mathbf{X’} \\sim \\pi\_\\theta$ instead of $\\mathbf{X’} \= \\pi\_\\theta$.
4. The function $r(\\mathbf{X}; s, c, \\theta)$ in line 173 is never explicitly defined.
5. It is confusing to use $\\theta$ for both the language model and paraphraser parameters.
   * For example, in Definition 1, why does the “paraphraser parameters $\\theta$” appear in the watermarked output distribution $P\_{s,c,\\theta}$?
   * Same thing for using $c$ for context for both original language model and paraphraser.
6. Is it generally true for most or all watermarking schemes that the detector score is sub-Gaussian around its mean?
7. In Theorem 1, line 213, does the $(x)\_{+}$ operation come before the squaring? It is ambiguous from the notation.
8. In Theorem 1, line 215, $\\mathcal{C}$, $\\Theta$, and $R\_{\\text{KL}}$ are not explicitly defined. I’m assuming $R\_\\text{KL}$ is the infimum of $r\_{\\text{KL}}$ over $c \\in \\mathcal{C}$ and $\\theta \\in \\Theta$?
9. Line 273: why is $\\pi\_\\theta$ conditioned on $wr$ instead of $q$ in the KL divergence estimator?
10. Line 314: what does using “dynamic $w\_1$” mean?
11. The experiment setup lists Qwen3-8B as an attacker model, but it does not appear in Table 1\. It appears in Table 8 in the appendix, but without RLCracker.
12. KGW, UPV, and SynthID are listed in the setup, but do not appear in Table 1\. They appear in Tables 14 and 15 in the appendix, but there is no reference to these tables in the main paper.
13. How does the ESR change as the P-SP score threshold varies? It would be interesting to see a plot of this.
14. I am having trouble seeing why using Lemma 1 leads to line 731 in the appendix. The form does not seem to match up.
15. Some additional potential related works are [Zhang et al. (2023)](https://arxiv.org/abs/2311.04378), [Chang et al. (2024)](https://arxiv.org/abs/2407.14206), and [Rastogi et al. (2024)](https://arxiv.org/abs/2411.05277).

### Minor issues

1. Line 116 typo: “fail to” \-\> “failing to”
2. Line 174: “defender’s guaranteed to zero”, is this missing a noun after guaranteed?
3. Definition 1, line 196: I think it would make more sense for the “s.t.” to be a comma instead.
4. Line 235: this is the first time the EWD watermark is mentioned, so it should have a citation here.
5. Line 431 typo: “ERS” \-\> “ESR”
6. Can you add a citation for Lemma 1 (Donsker-Varadhan Inequality)?
7. Krishna et al. (2023a) and (2023b) are duplicated references of the same paper.
8. Line 470 typo: “is around 2% occurs” \-\> “is around 2%, occurring”
9. Line 482 typo: “broader imapct” \-\> “broader impact”
10. Typo: “Eq. equation 2” appears in the appendix a few times.

---

> ### Author Response · Authors · 2025-11-18
> **Official Response by Authors to Reviewer aaTd (1/3)**
>
> We sincerely thank the reviewer for the thoughtful and constructive feedback. Below, we provide a detailed point-by-point response and hope that our clarifications and revisions effectively address your concerns.
>
> > **Q1: There is a gap between RLCracker and the theoretical results.**
>
> **A:** We thank the reviewer for the valuable question. The distributional robustness lens introduced in Sec. 4.1 is not auxiliary, it is what enables RLCracker. This definition provides two key benefits: (1) it is ***more informative***, as modeling a local watermark distribution reveals the watermark’s approximate behavior ***without requiring any prior knowledge of the underlying watermarking mechanism***; and (2) it offers a ***smooth KL geometry*** that supports principled, gradient-guided optimization rather than discrete instance-level search. These properties make KL divergence the natural measure for characterizing watermark robustness.
>
> Theorem 1 then states a ***necessary condition*** for a successful attack: the adversarial paraphrasing distribution must move beyond the certified KL neighborhood of the watermark distribution. This provides the theoretical justification and target for RLCracker’s objective: its KL-based reward is expressly designed to increase $\mathrm{KL}(Q\|P_{wm})$ and shift the distribution toward $P_h$ in accordance with this condition. We have revised Sec. 4.1 to highlight this connection and make the theory-to-method transition more prominent.
>
>
>
> > **Q2: Is the adaptive robustness radius a constant?**
>
> **A:**  We thank the reviewer for the insightful question. The true adaptive robustness radius $r^*(X)$ is indeed a *global* quantity: by definition, it must hold for all paraphraser parameters $\theta$, contexts $c$, and attack distributions $Q$, so it is constant for a given watermark instance. However, computing this global radius is intractable in practice due to the combinatorial nature of the discrete paraphrase space.
>
> To obtain a tractable surrogate, our methodology restricts attention to a paraphrasing family $\pi_\theta(\cdot\,;c)$. Under *fixed*  model parameters $\theta$ and attack context $c$, this induces a local watermark distribution and yields the *KL adaptive radius*, which approximates the worst-case robustness within this attack family. Because this surrogate conditions on $(\theta,c)$, it becomes *optimizable*: an attacker can reduce the KL radius by tuning $\theta$ and $c$. To avoid confusion between the global radius $r^*(X)$ and this context-dependent surrogate, we have revised Lines~17--18 of the abstract for clarity.
>
> > **Q3: Whether a non-watermarked model can reliably approximate $P_{wm}$, especially for complex watermarks, is not empirically shown.**
>
> **A:** We appreciate the reviewer for highlighting this point. Our use of $\pi_{\text{ref}}(o_{i,t}\mid wr, o_{i,<t})$ to approximate $P_{wm}$ is grounded in a key observation from prior watermarking work [1]: simple paraphrasing does ***not*** remove the watermark when both the paraphraser and the context are weak. In other words, feeding a watermarked text $wr$ into an unwatermarked paraphraser still produces outputs that retain the watermark signal, making $\pi_{\text{ref}}$ a practical and effective proxy for the watermark-induced distribution.
>
> This behavior is also reflected in our experiments: Table 1 shows that weaker models yield low ESRs under simple paraphrasing, indicating that the watermark persists in the rewritten outputs. We have revised Sec. 4.2 to explain this more clearly and to emphasize why this approximation faithfully captures $P_{wm}$ in the black-box setting.
>
> > **Q4: Lack of evaluation for paraphrased text quality.**
>
> **A:** We sincerely thank the reviewer for highlighting the text quality evaluation. We followed prior work [2] which evaluates the fluency using (1) perplexity (PPL) on Llama3.1-8B-Instruct, and (2) GPT as a judge (GPTS.). We also calculate additional Sentence-BERT scores (sBert) for semantic similarity. As shown in the table below, the RLCracker-optimized Qwen3-4B model generates high-quality rephrased text, preserving semantics and achieving high GPT scores and comparable perplexity to the original watermarked text. This improvement is due to our training objective, min $D_{KL}(Q||P_h)$, which aligns the model output closer to human language distribution, resulting in comparable text quality. We provide more results in Appendix E.1, Table 15, 17, 19 ,21, 23, 25.
>
> | Watermark | P-SP | sBert | GPTS. | PPL (rephrased/original) |
> |-----------|------|-------|-------|--------------------------|
> | EWD       | 0.87 | 0.9   | 8.81  | 4.47 / 3.58              |
> | PF        | 0.82 | 0.88  | 8.61  | 5.03 / 10.6              |
> | SWEET     | 0.85 | 0.89  | 8.73  | 4.35 / 3.53              |

---

> ### Author Response · Authors · 2025-11-18
> **Official Response by Authors to Reviewer aaTd (2/3)**
>
> > **Q5: Are there empirical or theoretical justifications for why the Sub-Gaussian Detector Score Assumption holds for most watermarking schemes?**
>
> **A:** We sincerely thank the reviewer for the insightful question. Figure 4(b) shows that the detection scores for 400 EWD and SWEET samples exhibit tightly concentrated, light-tailed behavior consistent with a sub-Gaussian distribution. In addition, Appendix C.1 provides a formal justification: under mild and widely satisfied conditions, the normalized green-list detector score can be expressed as a bounded-sum martingale, yielding provable sub-Gaussian tails via Azuma–Hoeffding. Together, these empirical observations and theoretical arguments support the validity of the Sub-Gaussian Detector Score Assumption for common watermarking schemes.
>
>
>
> > **Q6: Cited paper may not be suitable.**
>
> **A:** We thank the reviewer for pointing this out. The cited work [3] indeed demonstrates that a few standard watermarking schemes can be vulnerable to translation and introduces a strengthened design to address this weakness. Our intention was to reference this line of research rather than suggest that existing methods are robust to translation. We have revised the sentence in the introduction to accurately reflect this and to prevent any misinterpretation.
>
>
>
> > **Q7: Does the $(x)_+$ operation come before the squaring?**
>
> **A:** We thank the reviewer for identifying this ambiguity. Yes, the $(x)_+$ operation is applied ***before*** squaring. The certificate uses $((\mu(X,s,c,\theta)-\delta)_+)^2$ to ensure that only positive deviations above the detection threshold contribute to the radius. We have revised the statement of Theorem 1 to make this ordering explicit.
>
>
>
> > **Q8: Why the denominator of unbiased KL divergence estimator is conditioned on input q.**
>
> **A:** We thank the reviewer for the question. In our setting, the KL term is defined over the *adversarial paraphrasing distribution* $Q$, whose input is the *watermarked text* $wr$. The attacker does not paraphrase the question $q$—it paraphrases the watermarked output itself. Therefore, the unbiased KL estimator conditions the denominator on $\pi_\theta(o_{i,t}\mid wr, o_{i,<t})$, which correctly reflects the distribution that RLCracker is learning to manipulate. Conditioning on $q$ would correspond to a different task (question rewriting), which is not the attack considered here.
>
>
>
> > **Q9: What does using “dynamic $w_1$” mean?**
>
> **A:** We thank the reviewer for the insightful question. RLCracker uses a dynamic $w_1$ to adjust the semantic reward weight based on its magnitude. This allows the model to focus on semantic alignment when generated text has low similarity to the watermarked text, reducing the KL reward's impact and steering training toward better semantic similarity. As semantic fidelity improves, $w_1$ reduces the semantic reward influence, encouraging broader distribution exploration. We provide more details in Appendix D.3 (Line 1150-1155).
>
>
>
> > **Q10: Missing RLCracker results for Qwen3-8B.**
>
> **A:** We sincerely thank the reviewer for pointing out the missing results. At the time of submission, the Qwen3-8B experiments were still running and had not yet been completed, so we were unable to include them in Table 1. We have now finished these experiments and updated the manuscript accordingly, improving the presentation of Table 1. As shown, the Qwen3-8B model achieves consistently higher ESRs across all watermarking schemes. We appreciate your valuable feedback.
>
>
>
> > **Q11: How does the ESR change as the P-SP score threshold varies? It would be interesting to see a plot of this.**
>
> **A:** We sincerely thank the reviewer and fully agree with the valuable comment. We plotted the ESR changes for four attack methods on the Qwen3-4B model as the P-SP score threshold varies, and have updated Appendix D.8 with the results. As shown, when the P-SP threshold is above 0.85, the ESR for all four methods is similar. However, when the threshold drops below 0.8, RLCracker's ESR rises sharply, surpassing all other baselines and ultimately converging above 90%.
>
>
>
> > **Q12: The form in proposition 3 does not seem to match up.**
>
> **A:** We sincerely thank the reviewer for the valuable comments. After re-examining the theoretical flow, we realized that Proposition 3 was not sufficiently connected to the core development of our framework. Including it without a stronger contextual foundation risked creating confusion. To maintain clarity and coherence, we have removed the entire proposition from the paper and revised Section 4.2 (Lines 265–270) accordingly to ensure a smoother and more focused presentation of our main results.

---

> > ### Author Response · Authors · 2025-11-18
> > **Official Response by Authors to Reviewer aaTd (3/3)**
> >
> > > **Q13: There are several issues in the paper, including unclear notation, ambiguous references, formal errors, missing definitions, and typographical mistakes.**
> >
> > **A:** We sincerely thank the reviewer for raising these insightful questions and pointing out the typographical errors. In response, we have made the following revisions:
> >
> > 1. **Clarification of the P-SP score in the abstract**: We have modified the abstract to clarify the meaning of the P-SP score. Specifically, we have replaced "P-SP" with "with minimal semantic shift" to improve reader understanding.
> >
> > 2. **Clarification of $ X' $ as a random variable**: $ X' $ is not a random variable; it refers to the paraphrased output generated by the model given the input.
> >
> > 3. **Explicit definition of the function $ r(X; s, c, \theta) $ in line 173**: To eliminate unnecessary notation and provide clarity, we have revised the adversarial optimization objective in Section 3 to:$
> >    \min_{c \in \mathcal{C},\, \theta \in \Theta} r(\mathbf{X}),
> >    $and have provided a clear definition of this function (Line 175).
> >
> > 4. **Inconsistent use of $\theta$ for both the language model and paraphraser parameters**: We have streamlined the notation throughout the paper to ensure consistent and clear use of parameters (Lines 118–208).
> >
> > 5. **Explicit definitions of $ C $, $ \Theta $, and $ R_{KL} $**: We have now explicitly defined $ C $, $ \Theta $, and the function $R_{\mathrm{KL}}(\mathbf{X}, \mathcal{C}, \Theta) = \min_{c \in \mathcal{C}, \, \theta \in \Theta} r_{\mathrm{KL}}(\mathbf{X}, c, \theta)$ in Line 252.
> >
> > 6. **Lack of references to tables in the main text**: We have improved the presentation of our theoretical results and methods, enhanced the layout of the tables (Lines 378-381), and added additional results on text quality (Tables 1, 8, 15–25).
> >
> > 7. **Addition of related works**: We have expanded the related works section in Appendix B, incorporating the references [4, 5, 6] suggested by the reviewer.
> >
> > 8. **Typographical errors**: All identified typographical errors have been corrected.
> >
> >
> >
> > [1] Kirchenbauer, John, et al. "A watermark for large language models." ICML 23.
> >
> > [2] Cheng, Yixin, et al. "Revealing Weaknesses in Text Watermarking Through Self-Information Rewrite Attacks." ICML'25.
> >
> > [3]He, Zhiwei, et al. "Can watermarks survive translation? on the cross-lingual consistency of text watermark for large language models." arXiv preprint arXiv:2402.14007 (2024).
> >
> > [4] Zhang, Hanlin, et al. "Watermarks in the sand: Impossibility of strong watermarking for generative models." arXiv preprint arXiv:2311.04378 (2023).
> >
> > [5] Chang, Hongyan, Hamed Hassani, and Reza Shokri. "Watermark smoothing attacks against language models." arXiv preprint arXiv:2407.14206 (2024).
> >
> > [6] Rastogi, Saksham, and Danish Pruthi. "Revisiting the robustness of watermarking to paraphrasing attacks." arXiv preprint arXiv:2411.05277 (2024).

---

> ### Author Response · Authors · 2025-11-22
> **Looking Forward to Constructive Feedback and Further Discussion**
>
> Dear Reviewer aaTd,
>
> Thank you very much for your thoughtful review. We greatly appreciate the time and effort you've invested in assessing our manuscript. We have carefully revised our response and would be grateful if you could take a moment to review our updated general response, as well as our specific response to your comments. We hope that these revisions address your concerns, but if there are still aspects that may warrant further discussion, we would greatly appreciate your guidance.
>
> Once again, we deeply appreciate your time and valuable feedback.
>
> Best regards,
>  The Authors

---

> > ### Comment · Reviewer_aaTd · 2025-11-22
> >
> > Has the revised PDF been uploaded? I think I still can only see the original PDF.

---

> > > ### Author Response · Authors · 2025-11-22
> > >
> > > Dear Reviewer aaTd,
> > >
> > > We do have uploaded our revised manuscript, and the PDF we see on our side is indeed the updated version, with part of the content highlighted in blue. We are not sure why this issue has occurred. If you are still experiencing the same problem, we will kindly seek assistance from the AC.

---

> > > ### Author Response · Authors · 2025-11-24
> > > **Looking Forward to Constructive Feedback and Further Discussion**
> > >
> > > Dear Reviewer aaTd,
> > >
> > > Thank you very much for your thoughtful and detailed review. We have made additional revisions in Lines 423-425, 428-429, 530-532 and 534-537 to our manuscript based on Reviewer pW6u’s comments. Specifically, we have emphasized that our RLCracker tool can serve as a practical diagnostic resource for watermark designers to assess whether their watermarking schemes meet the required standards before deployment.
> > >
> > > We kindly ask if you would be able to review our updated manuscript. We hope that the revisions, along with our specific responses to your comments, address your concerns effectively. If there are still any points that require further clarification or discussion, we would greatly appreciate your guidance.
> > >
> > > If you are unable to access the revised manuscript, we will kindly seek assistance from the AC. We deeply value your time and constructive feedback, which has significantly contributed to improving our manuscript.
> > >
> > > Best regards,
> > >
> > > The Authors

---

> > > > ### Comment · Reviewer_aaTd · 2025-11-25
> > > >
> > > > I am able to see the updated PDF now, I think it was previously an issue with my browser.
> > > >
> > > > I appreciate the authors taking the significant amount of time and effort to revise the paper and thoroughly respond to my and other reviewers’ questions and concerns.
> > > >
> > > > I have a few remaining/additional questions:
> > > >
> > > > 1. I think that there is still some ambiguity in the manuscript about what exactly $r(X)$ is. What is the formal definition of $r(X)$. What is the relationship between $r(X)$ and $r^\\star(X)$? Does $r(X)$ involve the constraint that $f(\\pi\_\\theta(X; c), s) \\leq \\delta$?
> > > > 2. If $X’$ is not a random variable, does this mean that the paraphrase is generated from the paraphrase model with temperature 0?
> > > > 3. You said that $X’$ is not a random variable, but $Q$ is defined as “an attacker’s distribution over meaning-preserving rewrites.” Could you clarify the difference here?
> > > > 4. Can there be a pointer in the main body to the expanded related works section in Appendix B?
> > > > 5. Line 250: based on the non-strict inequality in Proposition 1, the smallest attainable KL adaptive radius does not necessarily decrease even when non-zero, it just cannot increase (it can remain the same).
> > > >    1. Same issue with “strictly reduces” in line 260\.
> > > > 6. Line 320: in the KL estimator, what are we computing the probability of under $P\_\\ast$? $P\_\\ast$ itself is a distribution, not a number, so I’m not sure what it means to directly have it in the equation like that.
> > > >    1. Are we averaging over the outputs?
> > > > 7. In Table 7, $w’\_1$ for Qwen2.5-3B-Instruct appears as 0, which is different from all the other models. Is this correct?
> > > >
> > > > ## Incorrect proof
> > > >
> > > > 8. There are errors in the proof of Proposition 2 in Appendix C.1, making it incorrect.
> > > >    1. Line 925 claims that $$f(X’, s) \- \\mu \= \\frac{S\_n}{\\sqrt{V}}.$$ However, $$f(X’, s) \- \\mu \= \\frac{1}{\\sqrt{V}} \\sum\_{i} (Y\_i \- p\_i) \- 0 \= \\frac{1}{\\sqrt{V}} \\sum\_{i}(Y\_i \- m\_i) \+ (m\_i \- p\_i)\] \= \\frac{S\_n}{\\sqrt{V}} \+ \\frac{1}{\\sqrt{V}}\\sum\_{i}(m\_i \- p\_i),$$ so the equality is only true if $$\\sum\_i m\_i \= \\sum\_i p\_i,$$ i.e., $$\\sum\_i \\mathbb{E}\[Y\_i \\mid \\mathcal{F}\_{i-1}\] \= \\sum\_i \\mathbb{E}\[Y\_i\],$$ which is not generally true given “no independence or Markov assumptions.”
> > > >    2. Line 930: since $V \\leq n$, $$2 \\exp\\left( \- \\frac{t^2 V}{2n}\\right) \\geq 2 \\exp\\left(-\\frac{t^2}{2}\\right)$$ since the exponent is negative, so the inequality claimed in the paper is flipped in the wrong direction.
> > > > 9. Proposition 2 as stated appears to be false.
> > > >    1. Counterexample: with “no independence or Markov assumptions”, then we can let $Z \\sim \\text{Bernoulli}(1/2)$ and set $Y\_i := Z$ for all $i$. Then, $f(X’, s)i$ is either equal to $-\\sqrt{n}$ or $\\sqrt{n}$ with probability $1/2$ each, which is clearly not sub-Gaussian.
> > > >
> > > > ## Typos
> > > >
> > > > 10. Beginning of abstract: “Large Language Models (LLMs) watermarking” \-\> “Large language model (LLM) watermarking”
> > > > 11. Line 41: Liu et al. (2023b) should be a parenthetical citation.
> > > > 12. Line 66: “effectively removes” \-\> “effectively removing”
> > > > 13. Line 204: “For a given watermark instance $X$”: should this be “watermarked input $X$”?
> > > > 14. Line 528: there should be a space after “Conclusion.”
> > > > 15. Line 837: should “Theorem 2” be labeled “Theorem 1”?

---

> > > > > ### Author Response · Authors · 2025-11-28
> > > > > **Official Response by Authors to Reviewer aaTd (1/2)**
> > > > >
> > > > > We sincerely thank you for the valuable feedback and greatly appreciate your thoughtful guidance. Below, we provide a detailed point-by-point response and hope that our clarifications and revisions effectively address your concerns.
> > > > >
> > > > > > **Q1. What is the formal definition of $r(X)$? What is the relationship between $r(X)$ and $r^\star(X)$.**
> > > > >
> > > > > **A1:** We sincerely thank the reviewer for the insightful question. The optimization objective of the adversaries is to directly minimize the semantic deviation needed to break the watermark, by tuning the model parameters $\theta$ and attack context $c$. Specifically, the adversary's objective is: $\inf_{\theta \in \Theta,\; c \in \mathcal{C},\; \mathbf X'} \; d(\mathbf X, \mathbf X') \quad \text{s.t.}\quad \mathbf X' \in \mathrm{supp}(\pi_\theta(\cdot\mid \mathbf X,c)),\;f(\mathbf X', s) \le \delta.$ We recognize the definition of $r(\mathbf{X})$ is indeed ambiguous, and since it was only introduced once on Line 178 without further usage, we have chosen to remove it for clarity. We have also revised Line 178 to more directly highlight the adversary's objective, aiming to improve the clarity and readability of the manuscript.
> > > > >
> > > > >
> > > > >
> > > > > > **Q2 & 3. Is $X'$ a random variable?**
> > > > >
> > > > > **A2&3:** We thank the reviewer for raising this helpful clarification request. In the revised manuscript, we now make a clear notational distinction between *deterministic paraphrase instances* and *random paraphrases drawn from a distribution*.
> > > > >
> > > > > * **Section 3:** We use $\mathbf{X}'$ to denote a *single paraphrased sequence*, i.e., a concrete element of $\mathcal{V}^*$. The instance-level robustness radius  $r^\star(\mathbf{X})$ certifies correctness for *all* such paraphrase  instances, so $\mathbf{X}'$ is treated deterministically rather than as a random variable.
> > > > > * **Section 4:** We introduce a distributional view of the attacker. To avoid ambiguity, we now use a distinct notation $  \mathbf{X}'' \sim Q$ to denote a *random paraphrase* sampled from the adversarial paraphrasing distribution $Q$. This makes explicit the difference between the deterministic paraphrase $\mathbf{X}'$ and the stochastic paraphrase $\mathbf{X}''$, and cleanly separates the instance-level and distributional formulations.
> > > > >
> > > > >  We have revised Lines 128-131, 161, 168, and 178 of our manuscript for better clarity and readability.
> > > > >
> > > > > > **Q4Missing pointer to Appendix B in the Related Works section**
> > > > >
> > > > > **A4:** We sincerely appreciate the reviewer’s efforts in enhancing the readability of our manuscript. We have added a pointer to Appendix B at the end of the Related Works section, with the pointer in Line 119 marked in blue.
> > > > >
> > > > > > **Q5. Based on the non-strict inequality in Proposition 1, the smallest attainable KL adaptive radius does not necessarily decrease even when non-zero.**
> > > > >
> > > > > **A5:** We sincerely thank the reviewer for pointing out the inappropriate wording. We fully agree that the phrases “must be reduced’’ and “strictly reduces’’ were too strong and could be misinterpreted as making a universal monotonicity claim. Our intention was only to convey that enlarging the candidate set of contexts
> > > > > $c$ and model parameters $\theta$ *can* reduce the KL radius, a trend that we observe empirically in Appendix E.9 (Table 13), where richer attack contexts and more capable paraphrasing models consistently lead to smaller KL radii and higher watermark removal success. To avoid overstating the guarantee, we have revised Lines 247, 250, and 260, replacing “must’’ and “strictly’’ with “can’’ to accurately reflect what is supported by our analysis and empirical results.
> > > > >
> > > > > > **Q6. What are we computing the probability of under $P_\* $ ? $P_\* $ itself is a distribution, not a number.**
> > > > >
> > > > > We thank the reviewer for raising this question. The notation $P_{*}$ is used to represent either the human-like distribution $P_{h}$ or the watermarked distribution $P_{wm}$. In practice, both are approximated using the unwatermarked reference model $\pi_{\mathrm{ref}}$, which provides computable token probabilities. Concretely, we approximate:
> > > > >
> > > > > $$P_{h}(o_{i,t}) \approx \pi_{\mathrm{ref}}(o_{i,t}\mid q, o_{i,<t}),\qquad
> > > > > P_{wm}(o_{i,t}) \approx \pi_{\mathrm{ref}}(o_{i,t}\mid wr, o_{i,<t}),$$
> > > > >
> > > > > where $(q, wr)$ is the question–watermarked-response pair observable to the attacker. We have revised the manuscript (Lines~317–323) to explicitly define $P_{*}$ and clarify how it is computed.

---

> ### Author Response · Authors · 2025-11-28
> **Official Response by Authors to Reviewer aaTd (2/2)**
>
> > **Q7. Is it correct for $w_1^{'}$ of Qwen2.5-3B-Instruct appears as 0?**
>
> **A7:** We sincerely thank you for your insightful question. Due to the model's internal characteristics, we found that for the non-reasoning model Qwen2.5-3B-Instruct, directly setting $ w_1^{'} $ to 0 yields better performance, which is equivalent to setting the weight for our semantic similarity reward $\hat{A}_i$ to the constant 1. Similarly, in response to reviewer gbCX's question about RLCracker's applicability to other model families, we observed that setting $ w_1^{'} $ to 0 also yields better performance for Llama 3.2-3B-Instruct, compared to setting $ w_1^{'} $ to 9, as done for reasoning models like the Qwen 3 family.
>
> > **Q8. There are errors in the proof of Proposition 2**
>
> **A8:** Thank you for the careful reading and for identifying the issues in the original proof of Proposition 2. We agree with your observations regarding (i) the missing drift term, (ii) the sign error in the tail bound, and (iii) the fact that a dimension-free sub-Gaussian parameter cannot hold without additional structural assumptions. Your Bernoulli construction is a helpful illustration of this limitation.
>
> In the revised manuscript, we have *restated Proposition 2* and *fully replaced the proof*. The new argument still does not rely on independence or Markov properties. Instead, it uses the simple fact that each term $(Y_i - p_i)$ lies in $[-1,1]$, so their sum is a bounded random variable and therefore sub-Gaussian with variance proxy $n^{2}/4$. Normalizing by $\sqrt{V}$ yields the corrected bound
>
> $$\Pr\left(\big|f(\mathbf X',s)-\mathbb E f(\mathbf X',s)\big|\ge t\right)
> \le 2\exp\left(-\frac{2V t^{2}}{n^{2}}\right).$$
>
> To make the intuition transparent, we have added two concise remarks after the proof:
> (i) one explaining that the result follows directly from the fact that any linear combination of bounded variables is sub-Gaussian, even under arbitrary dependence; and
> (ii) one illustrating how your Bernoulli example fits the revised proposition, since it remains sub-Gaussian for each fixed $n$ with variance proxy $\Theta(n)$, exactly as captured by our corrected statement.
>
> We appreciate your thoughtful feedback; it strengthened both the clarity and correctness of this result.
>
> > **Q9. There are several typos in the manuscript.**
>
> **A9:** We thank the reviewer for pointing out the typographical errors. We have now corrected all identified errors, which are highlighted in blue. We have also double-checked the grammar throughout the paper and corrected the errors.

---

### Author Response · Authors · 2025-11-18
**General Response**

We sincerely thank all the reviewers for their time, thoughtful feedback, and constructive evaluations. Reviewers aaTd and nyNq acknowledged our contribution in proposing the adaptive robustness radius. Specifically, Reviewer aaTd appreciated the **principled** nature of our approach, while Reviewer nyNq recognized that the proposed radius provides a **formal foundation** for evaluating worst-case watermark robustness. Reviewer aaTd also noted the **intuitiveness** of the RLCracker attack, highlighting that our method **addresses limitations** of prior approaches. Reviewer pW6u commended our work is **well-motivated** and the convincing evaluation setting for RLCracker. Reviewer gbCX noted that our work offers **many valuable insights** that are **highly beneficial** to the community. Both Reviewer aaTd and Reviewer gbCX also acknowledged the extensiveness of our experimental evaluation. Furthermore, all four reviewers praised the strong performance of RLCracker.

We also appreciate the constructive feedback regarding the notations and typographical errors. In response, we have made the following updates, with changes in the main text marked in blue:

1. We’ve clarified the KL-adaptive radius definition and explanation in Section 4.1 to improve understanding of both our theory and algorithm. We’ve streamlined the notation for clarity and added more details on the principles behind our method. Additionally, we justify the sub-Gaussian assumption in Appendix C.1.

2. We’ve strengthened the connection between our theory and Theorem 1 and clarified the methods for approximating $P_h$ and $P_{wm}$ in Section 4.2. Specifically, $P_{wm}$ is instantiated by passing the watermarked instance $wr$ through a lightweight paraphraser, while $P_h$ is derived by querying the same unwatermarked paraphraser with the question $q$ used to generate the watermark (Lines 313–316).

3. We’ve refined the result tables in Section 5 and Appendix E, added RLCracker results for Qwen3-8B, and included a new analysis of the text quality across models in Tables 8 and 15–25.

4. We have expanded the related works section in Appendix B to provide a more comprehensive overview.

5. We explored the ESR variation with different P-SP score thresholds in Appendix E.8, and provided an empirical analysis of the correlation between the adaptive robustness radius and watermark evasion success in Appendix E.9. Appendix E.10 shows that RLCracker’s effectiveness hinges on the gap between $P_{wm}$ and $P_h$, not on internal distribution discrepancies between the watermark generator and the attack model.

6. All typographical errors have been corrected based on the reviewer's comments.

We also thank Reviewer nyNq for raising ethical concerns. We would like to clarify that our goal is to expose vulnerabilities in existing watermarking schemes and highlight the limitations of current evaluation methods, not to harm existing technologies. We hope our work encourages the development of more robust watermarking techniques.

Again, we thank the reviewers for their valuable feedback, which has significantly improved our manuscript.

---

> ### Author Response · Authors · 2025-11-28
> **General Response to Public release of the code**
>
> We sincerely thank all the reviewers for their valuable feedback and thoughtful discussion. We fully agree with reviewers aaTd and nyNq that the potential for real-world harm from our algorithm is likely limited. However, we will defer to the ethics review chairs, who are better positioned to provide expert guidance on this matter. If the ethics review chairs agree that our algorithm does not pose significant risks, we would be happy to release the algorithm’s code, as stated in our Reproducibility Statement, to serve as a watermark robustness evaluation tool, helping LLM watermark designers to create more resilient watermarking techniques.
>
> On the other hand, if the chairs determine that our approach could potentially cause harm, we will respectfully refrain from releasing the trained models or the full attack pipeline. In such cases, only non-sensitive components will be made publicly available for basic reproducibility. Access to the core code will follow a controlled-release process (similar to Meta’s), in which researchers must request access individually, agree to responsible-use terms, and provide an academic or institutional affiliation. This approach aims to support legitimate research while minimizing the risk of broad misuse.

---

### Author Response · Authors · 2025-11-29
**Summary of Discussion**

We sincerely thank all reviewers for their time and valuable feedback, which significantly improved our manuscript. Due to a system error, we regret that our scores were reverted to their initial values. Given the length of our discussion, we have provided a summary of the discussion for the convenience of the ACs, to facilitate a clear and efficient understanding of our work and the key points raised. Speicifically,

- **Reviewer nyNq** initially expressed concerns regarding **(Q1)** the similarity of our work to previous studies, and **(Q2)** the lack of comparison with methods assuming stronger threat models. We addressed these concerns by **(R1)** clarifying the differences between our approach and prior work in Section 4.1, and **(R2)** providing additional experiments in Appendix E.9 and E.11. Ultimately, the reviewer acknowledged the contributions of our work, stating that *“see this work as a go-to for future robustness studies”*, leading to an increase in the score **from 2 to 6** on **November 22**.
- **Reviewer pW6u** raised concerns regarding **(Q1)** unclear details in our algorithm and experiments, particularly in how we approximate the watermarked and human-written distributions, and **(Q2)** the need for refinement in our multi-key ablation setup, suggesting that the best configuration would be *“one-sample-one-key”*. To address these, we **(R1)** revised the Section 4.2 of our manuscript to clarify the details of our algorithm, and **(R2)** provided additional experiments in Section 5.3, Appendix E.10 and E.12. The reviewer appreciated the revisions, stating that *“my main questions are well addressed”*, and highlighted that "*RLCracker provides a practical diagnostic tool for designers to assess whether their watermarking schemes meet the required standards before deployment*." The reviewer concluded with: “*Overall, **great work! I hope to see the paper presented at the conference***”. As a result, the score was increased **from 6 to 8** (stronger recommendation) **on November 23**.
- **Reviewer gbCX** raised similar concerns regarding **(Q1)** clarity in our algorithm, particularly in how we approximate the watermarked and human-written distributions, and **(Q2)** requested additional experiments using other model families and on semantic watermarks. To address these, we have **(R1)** made the necessary revisions in Section 4.2 of our manuscript, and **(R2)** added validation experiments. The reviewer appreciated the additional results, stating *“this is a **solid** paper, and I am maintaining my high score”*, and maintained a **score of 8 on November 25**. Our final response to the new suggestions raised by the reviewer on November 25 was posted on November 28.
- **Reviewer aaTd** provided a thoughtful analysis of our manuscript, highlighting ambiguities and typographical errors. The main concerns were as follows: **(Q1)** unclear notation definitions, algorithm details; **(Q2)** lack of justification for key assumptions and issues with the theoretical derivations; **(Q3)** missing evaluations metrics for text quality. To address these, we have: **(R1)** revised the notation system and algorithm details in Sections 3 and 4; **(R2)** added the justification for Assumption 1 in Appendix C.1 and corrected the theoretical derivations in Appendix C; **(R3)** included additional experiments and updated the relevant tables. The reviewer acknowledged the **principles** of our algorithm and the extensiveness of our evaluations, stating that *“The method addresses limitations from previous methods by generalizing better to longer generations”*, and has **maintained the score of 6 to date**. Our final response to the questions raised by the reviewer on November 25 was posted on November 28. We believe our responses have fully addressed the raised concerns.

We are genuinely encouraged that after the discussion, all reviewers have reached a consensus, and our scores have now risen to **8** (pW6u) **8** (gbCX) **6** (nyNq) **6** (aaTd) with confidence **4** (pW6u) **3** (gbCX) **4** (nyNq) **4** (aaTd). We also believe that our RLCracker can serve as an *effective and lightweight* LLM watermark robustness evaluation tool, helping designers create more robust LLM watermarks.

---

### Note · Program_Chairs · 2026-01-17
**Submission Desk Rejected by Program Chairs**

The following references in this submission do not refer to real documents and/or have major errors in bibliographic information:

 Yizheng Zhu, Hongxin Zhang, and Pin-Yu Chen. Certified robustness to adversarial word substitutions. In EMNLP, 2021